



# How well can satellite altimetry and firn models resolve Antarctic firn thickness variations?

Maria T. Kappelsberger[1], Martin Horwath[1], Eric Buchta[1], Matthias O. Willen[1,a], Ludwig Schröder[1,b], Sanne B.M. Veldhuijsen[2], Peter Kuipers Munneke[2], and Michiel R. van den Broeke[2]

[1]Institut für Planetare Geodäsie, Technische Universität Dresden, Dresden, Germany
[2]Institute for Marine and Atmospheric Research Utrecht (IMAU), Utrecht University, Utrecht, The Netherlands
[a]now at: Department of Geoscience and Remote Sensing, Delft University of Technology, Delft, The Netherlands
[b]now at: Bundesamt für Kartographie und Geodäsie, Leipzig, Germany

**Correspondence:** Maria T. Kappelsberger (maria.kappelsberger@tu-dresden.de)

**Abstract.** Elevation changes of the Antarctic Ice Sheet (AIS) related to surface mass balance (SMB) and firn processes vary strongly in space and time. Their short-term natural variability is large and hampers the detection of long-term climate trends. Firn models or satellite altimetry observations are typically used to investigate such firn thickness changes. However, there is a large spread among firn models. Further, they do not fully explain observed firn thickness changes, especially on smaller

temporal and spatial scales. Reconciled firn thickness variations will facilitate the detection of long-term trends from satellite altimetry, the resolution of the spatial patterns of such trends and, hence, their attribution to the underlying mechanisms. This study has two objectives: First, we quantify interannual Antarctic firn thickness variations on a $10\,\mathrm{km}$ grid scale. Second, we characterise errors in both the altimetry products and firn models. To achieve this, we jointly analyse satellite altimetry and firn modelling results in time and space. We use the timing of firn thickness variations from firn models and the satellite-

observed amplitude of these variations to generate a combined product ('adjusted firn thickness variations') over the AIS for 1992–2017. The combined product characterises spatially resolved variations better than either firn models alone or altimetry alone. We detect highest absolute differences between the adjusted and modelled variations at lower elevations near the AIS margins, probably influenced by the lower resolution, more blurred spatial distribution of the modelled variations. In a relative sense, the largest mismatch between the adjusted and modelled variations is found in the dry interior of the East Antarctic Ice

Sheet (EAIS), in particular across large megadune fields. Here, the low signal-to-noise ratio poses a challenge for both models and altimetry to resolve firn thickness variations. The altimetric residuals still contain a large part of the altimetry variance and include firn model errors, such as firn signals not captured by the models, and altimetry errors. Apart from time-variable penetration effects of radar altimetry signals, the residuals disclose patterns indicating uncertainties in intermission calibration.

## 1  Introduction

The global mean sea level rose by $3.05 \pm 0.24\,\mathrm{mm\,yr^{-1}}$ during the period 1993–2016 (Horwath et al., 2022). Ice-mass loss from Antarctica contributed $\sim 6\%$ to this rise, and is likely to continue (Horwath et al., 2022; IPCC, 2021). The evolution of the Antarctic Ice Sheet (AIS) is of critical concern because the AIS contains the world's largest reservoir of frozen freshwater,





the equivalent of $\sim 58\,\mathrm{m}$ in global mean sea level (Fretwell et al., 2013), and projections of Antarctica's future contribution to the sea-level rise exhibit a large spread (Schlegel et al., 2018). Relative to 1995–2014, by 2100, Antarctica is expected to

contribute $0.03$ to $0.27\,\mathrm{m}$ and $0.03$ to $0.34\,\mathrm{m}$ (likely ranges) to the global mean sea level rise under the low and very high greenhouse gas emissions scenario, respectively (Fox-Kemper et al., 2021). In order to narrow the range of future sea-level rise projections, we need to better understand the ice-sheet processes and, for this, improve models and observational constraints to quantify the associated volume and mass changes with higher accuracy.

## 1.1 Antarctic mass balance and the role of SMB variations

The mass balance of a grounded ice sheet is commonly separated into three processes: surface mass balance (SMB), basal mass balance, and ice discharge. SMB comprises total precipitation (snowfall, rainfall), total sublimation (from surface and drifting snow), drifting snow erosion and meltwater runoff (van den Broeke et al., 2016; van Wessem et al., 2018). It refers to processes occurring on the surface of the ice sheet in the snow and firn layer. Ice discharge is the ice flow across the grounding line and is linked to processes occurring in the ice layer (Willen et al., 2021). Basal mass balance is thought to be small (Otosaka et al.,

2023a), and not considered here.

The current overall mass balance of the AIS is dominated by an increase in mass loss through ice discharge resulting from an acceleration of glacier flow, primarily from outlet glaciers of the West Antarctic Ice Sheet (WAIS) (Velicogna et al., 2020; Rignot et al., 2019). However, variations in SMB (dominated by precipitation) control the variability of the Antarctic mass balance on interannual to decadal timescales (Rignot et al., 2019; Davison et al., 2023). The amplitudes of SMB variations, as

well as the SMB itself, vary strongly over space. They are influenced by ice sheet topography and atmospheric and oceanic conditions (Lenaerts et al., 2019). Antarctic SMB variability is associated with large-scale atmospheric circulation, such as the Amundsen Sea Low, the Southern Annular Mode and the El Niño Southern Oscillation (e.g. Cullather et al., 1996; Lenaerts et al., 2019; Noble et al., 2020; Kaitheri et al., 2021). The strong interannual Antarctic SMB variability hampers the detection of statistically significant trends in the Antarctic (surface) mass balance. To separate long-term trends from short-term variability,

the time period considered is essential (Wouters et al., 2013). Ice cores indicate an increase in SMB, in particular in West Antarctica, over the twentieth century (Thomas et al., 2017; Wang et al., 2019; Medley and Thomas, 2019). Over the shorter satellite period, on a decadal and multidecadal scale, possible trends are masked by the large short-term variability (Mottram et al., 2021; Gutiérrez et al., 2021). An improved quantification of interannual SMB variations in space and time is required in order to robustly resolve long-term SMB trends (King and Watson, 2020). This is currently lacking (e.g. Mottram et al., 2021).

## 1.2 Modelling and observing SMB and firn thickness changes

To date, the SMB for the entire ice sheet is commonly simulated using regional climate models (RCMs) that are thoroughly evaluated against hundreds of in situ observations of SMB. Earth system models have recently caught up in this regard (Lenaerts et al., 2019). RCMs specialise in the physics of polar ice sheets (van Wessem et al., 2018; Agosta et al., 2019). They are forced by atmospheric reanalysis products which typically provide data from 1979 onwards (Gossart et al., 2019). Mottram et al.

(2021) compared Antarctic SMB simulations from an ensemble of five different RCMs all forced by ERA-Interim (Dee et al.,



2011). Model differences comprise e.g. the topography model, horizontal resolution and complexity in (sub)surface, snow and firn schemes. Mottram et al. (2021) find that different RCMs provide similar outputs for annual to decadal SMB variations on a continental scale, as long as they are driven by the same reanalysis product. However, spatial variations in SMB show a poorer agreement. On a basin scale, the largest deviations are found for the Antarctic Peninsula and the basin that includes the Transantarctic Mountains and part of the interior of the East Antarctic Ice Sheet (EAIS) (basin 8 in this study; Fig. 3). For this basin, the ensemble standard deviation (std) of $40\,\mathrm{Gt\,yr^{-1}}$ amounts to $37\,\%$ of the ensemble mean. Moreover, even when models provide similar basin-wide SMB estimates, their spatial patterns differ substantially on a regional and local scale. The largest deviations between the models are mainly at the coastal margin of the entire grounded AIS.

Results from RCMs are used to force firn models, that simulate the temporal evolution of the Antarctic firn due to SMB and firn processes such as densification (Ligtenberg et al., 2011; Lundin et al., 2017). Firn elevation changes, or firn thickness changes, are an output of firn models. Verjans et al. (2021) examined differences in linear trends of firn thickness changes between a range of $54$ different firn model setups for the EAIS. On a basin scale, the ensemble stds range from $0.2$ to $1.0\,\mathrm{cm\,yr^{-1}}$ and amount to $15$ to $300\,\%$ of the ensemble mean trends of their respective basins. Over the entire EAIS, the choice of climate forcing (RCM), firn compaction and surface snow density contribute to the ensemble spread by $72\,\%$, $20\,\%$ and $4\,\%$, respectively, which highlights the importance and need for more precise RCMs.

Besides modelling tools, satellite measurements are the only possibility to infer ice-sheet-wide changes in SMB. Observations from the satellite gravimetry missions GRACE and GRACE-FO are widely used to estimate Antarctic ice mass changes (e.g. Horwath and Dietrich, 2009; Velicogna and Wahr, 2013; Barletta et al., 2013; Groh et al., 2019). Comparisons between gravimetric ice-mass balance estimates and modelled SMB results (with additional consideration of ice-dynamics changes) were made for the entire AIS, its main regions, drainage basins, or glacier catchments (Mohajerani et al., 2018; Velicogna et al., 2020; Groh and Horwath, 2021). However, gravimetric mass-balance estimates have to be corrected for superimposed signals such as glacial isostatic adjustment, involving large uncertainties (Shepherd et al., 2018; Whitehouse et al., 2019; Willen et al., 2020; Groh and Horwath, 2021). Moreover, GRACE/GRACE-FO cannot resolve mass changes on smaller spatial scales and their observations are restricted to the period after 2002.

By contrast, observations from satellite altimetry provide a higher spatial resolution of several kilometres and go back to the year 1992 for covering most of the AIS (Wingham et al., 1998). They allow the derivation of temporal changes of the ice sheet's surface elevation and are therefore sensitive to volume changes of the AIS and to the deformation of the solid Earth, with the latter negligible compared to the former (Willen et al., 2021). Most of the altimetry missions utilise(d) radar waves (e.g. Envisat, CryoSat-2). Since 2003 laser altimeters are also used (e.g. ICESat-2). While laser altimeters rely on good atmospheric conditions (no thick clouds or blowing snow) radar altimetry is independent of weather conditions (Otosaka et al., 2023a). On the other hand, laser signals are reflected at or near the ice sheet surface, independently of its properties, while radar signals penetrate into the upper snow/firn layers. Radar altimetry results can thus be biased by the time-variable dielectric properties of the ice sheet surface (Davis and Ferguson, 2004; Rémy et al., 2012). If elevation changes due to changing ice flow can either be neglected or subtracted, altimetric elevation changes can be compared to modelled elevation changes due to SMB and firn processes provided by firn models (Kuipers Munneke et al., 2015; Medley et al., 2022a).



While the rates of modelled and observed elevation changes agree well when averaged over large drainage basins and over 25 years, the correlation diminishes significantly on a grid scale and over 5 years (supplement to Shepherd et al., 2019). Recently, Veldhuijsen et al. (2023) reported that agreement between altimetry and firn modelling results improved when an updated firn model was employed. Nevertheless, discrepancies still remain. (See Section 2.3 for further details on comparisons between altimetry and firn models.) Inconsistencies between models and altimetry also affect the derivation of altimetric ice mass changes, as this depends on modelling results. Using the models in a rigorous, deterministic manner (Kuipers Munneke et al., 2015) resulted in altimetric mass changes characterised by widespread signals of dynamic imbalance that are not deemed fully realistic (supplement to McMillan et al., 2016; Shepherd et al., 2019; Kappelsberger et al., 2021). The reason likely lies in errors in the involved altimetry and modelling results. Therefore, a simplified approach using a (steady-state) density model is commonly applied (e.g. Sørensen et al., 2011; McMillan et al., 2016; Schröder et al., 2019a; Shepherd et al., 2019; Kappelsberger et al., 2021).

### 1.3 Previous work

Using SMB and firn modelling outputs alone to quantify interannual variations in SMB and firn thickness introduces large uncertainties: inter-model spread is large, and the model outputs also differ from observational data (Section 1.2). Likewise, interannual variations analysed using only data from gravimetry and altimetry are strongly affected by their errors (Horwath et al., 2012; Mémin et al., 2015; Su et al., 2018; Shi et al., 2022). Moreover, it is difficult to relate the variations derived from observations alone to their physical causes. Therefore, the studies of Sasgen et al. (2010), Bodart and Bingham (2019), Kim et al. (2020), Kaitheri et al. (2021) and Zhang et al. (2021) compared or combined observational and modelling/meteorological data. However, their derived interannual variations are spatially coarsely resolved (at about $400\,\mathrm{km}$) and mainly limited to the GRACE/GRACE-FO period.

### 1.4 Purpose

This study focuses on the interannual variations in firn thickness on a regional to local scale. Knowledge of interannual variations is required to isolate long-term trends in ice volume or mass changes (Section 1.1). To identify the underlying glaciological processes and separate SMB and firn signals from ice dynamics, the spatial patterns of interannual variations and long-term trends need to be resolved. As the analysis of basin integrals is not sufficient for this purpose, we work at $10\,\mathrm{km}$ grid-scale level. We characterise and quantify firn thickness variations in space and time by combining results from satellite altimetry and firn modelling. By combining both data sets, we expect to reduce uncertainties and errors compared with the variations derived from altimetry or models alone. For the first time, the entire spatial information present in both the altimetry products and modelling outputs, together with the high (monthly) temporal resolution of gridded altimetry products, is jointly exploited. Apart from determining firn thickness variations empirically, our analysis provides information on the error characteristics of both the altimetry products and the model outputs.



## 2   Data

### 2.1   Altimetry

We use the altimetry product from Technische Universität Dresden (TUD) (Schröder et al., 2019a), referred to as TUD altime-
try. As an alternative data set, we use the product from Jet Propulsion Laboratory (JPL) (Nilsson et al., 2022), referred to as
JPL altimetry. Schröder et al. (2019a) and Nilsson et al. (2022) derived monthly resolved elevation changes of the grounded
AIS from a multi-mission satellite altimetry analysis. The elevation changes represent elevation anomalies, as they refer to the
difference between the elevation at time $t$ and the elevation at a reference epoch $t_0$. We use elevation changes over the time
period May 1992 to December 2017 containing data from pulse-limited radar altimetry ERS-1, ERS-2, Envisat and CryoSat-2
low resolution mode (LRM), from radar altimetry CryoSat-2 in synthetic aperture radar interferometric (SARIn) mode and
from laser altimetry ICESat. As each altimetry mission differs in its orbit configuration, its maximum southern latitude differs.
Thus, the lower time limit May 1992 is set to ensure spatial data coverage up to $81.5°$ S. Grid cells with large gaps in the al-
timetry time series, such as the area south of $81.5°$ S and the Antarctic Peninsula are excluded. The upper time limit December
2017 is set to ensure an overlapping period of TUD and JPL altimetry. In the following, the main altimetry processing steps
are summarised and differences between TUD and JPL pointed out.

Schröder et al. (2019a) and Nilsson et al. (2022) corrected the measurements from pulse-limited radar altimetry for sloping
terrain with the relocation method (Roemer et al., 2007; Nilsson et al., 2016) using different digital elevation models (Helm
et al. (2014) versus Fretwell et al. (2013)). Both studies applied a threshold retracker for the 'offset center of gravity' amplitude
(Wingham et al., 1986) to the radar return signal (waveform). While the TUD product adopts a very low threshold at $10\%$
to reduce the sensitivity to variations in firn pack properties (Schröder et al., 2019a), the JPL product is based on a $30\%$
threshold for ERS-1, ERS-2 and Envisat data. CryoSat-2 LRM data were treated similarly for both products (using a $10\%$
threshold). Data from the CryoSat-2 SARIn mode was processed by Helm et al. (2014) and Nilsson et al. (2016) for TUD
and JPL, respectively. The height measurements were analysed using repeat-track altimetry on a polar-stereographic grid to
derive elevation time series. For this analysis, Schröder et al. (2019a) and Nilsson et al. (2022) used different grid spacing and
different search radii (constant versus varying/mission-dependent). Further differences refer to the removal of time-invariant
topography (bilinear surface versus varying models/mean, bilinear or biquadratic surface) and the correction for time-variable
radar signal penetration and scattering effects (backscatter correction versus backscatter, leading edge width and trailing edge
slope correction). While Schröder et al. (2019a) performed these two steps in one least-squares fit, Nilsson et al. (2022)
implemented two separate fits for this purpose.

To derive a continuous time series of elevation changes, intermission/intermode calibration offsets must be solved. This is
a major difference between both altimetry products: TUD is based on overlapping epochs or subtracting a technique-specific
reference elevation and JPL used a least-squares adjustment based on all altimetric measurements. In general, also the weighting
between measurements from different missions, in particular the weighting ratio between Envisat and ICESat (Table 1 Schröder
et al. (2019a) versus Table 1 Nilsson et al. (2021)) differs. Moreover, Nilsson et al. (2022) scaled the seasonal amplitudes of the
time series of ERS-1, ERS-2 and Envisat to the seasonal amplitudes derived from CryoSat-2. Finally, Schröder et al. (2019a)



smoothed the processed data by a three-month moving average and a $10\,\mathrm{km}$ $\sigma$ Gaussian weighting function. This reduced the spatial grid resolution to $10\,\mathrm{km}$ x $10\,\mathrm{km}$. Nilsson et al. (2022) interpolated the processed data with collocation (max. search radius of $50\,\mathrm{km}$, correlation length of $20\,\mathrm{km}$) on a spatial grid with a formal resolution of $1920\,\mathrm{m}$ x $1920\,\mathrm{m}$. We interpolate the JPL product to the spatial grid of TUD by averaging the data over $10\,\mathrm{km}$ x $10\,\mathrm{km}$. We smooth the JPL time series by a three-month moving average in order to conform to the TUD product. The use of the JPL altimetry data is further restricted to those points in time and space where TUD altimetry data is available.

In addition to TUD and JPL, Shepherd et al. (2019) published a long-term, multi-mission altimetry product. However, we do not use their product because it is not resolved on a monthly basis (consecutive 5-year intervals are provided).

## 2.2 Firn models

We use the firn model IMAU-FDM v1.2A of Veldhuijsen et al. (2023), referred to as IMAU (Institute for Marine and Atmospheric Research Utrecht) firn model, which is an update of v1.1 (Ligtenberg et al., 2011). As an alternative data set we involve the GSFC-FDM v1.2.1 of Medley et al. (2022a), referred to as GSFC (Goddard Space Flight Center) firn model. It uses the Community Firn Model framework of Stevens et al. (2020, 2021). Here, one output of the models is used, the firn thickness changes. Firn thickness changes represent firn thickness anomalies, as they refer to the difference between firn thickness at time $t$ and the mean firn thickness over a certain reference period (see below). The IMAU model outputs are given every ten days and on a regular grid with a spacing of $27\,\mathrm{km}$ from 1979 to 2020. The GSFC model outputs are given every five days and on a regular grid with a spacing of $12.5\,\mathrm{km}$ from 1980 to 2021. In accordance with the altimetry data, we involve firn thickness changes from the grounded AIS excluding the Antarctic Peninsula and the period May 1992 to December 2017. We adapt the temporal resolution to that of the altimetry product by calculating monthly means and applying a three-month moving average smoothing. In the following, the main firn model set ups are summarised and differences between IMAU and GSFC pointed out.

The IMAU firn model is forced with 3-hourly fields of surface temperature, $10\,\mathrm{m}$ wind speed and SMB components (snowfall, rainfall, sublimation, snowdrift erosion, snowmelt) from the ERA5 atmospheric reanalysis data (Hersbach et al., 2020) dynamically downscaled with RACMO2.3p2 (van Wessem et al., 2018) to a spatial resolution of $27\,\mathrm{km}$ x $27\,\mathrm{km}$. The GSFC firn model is forced with hourly fields of snowfall, total precipitation, evaporation, $2\,\mathrm{m}$ air temperature, skin temperature and runoff from the MERRA-2 atmospheric reanalysis data (Gelaro et al., 2017) downscaled to a spatial resolution of $12.5\,\mathrm{km}$ x $12.5\,\mathrm{km}$. The firn layer was initialised by looping over the forcing data of the reference period 1979–2020 (for the IMAU model) and 1980–2019 (for the GSFC model) until the firn column was refreshed at least once. This implies the assumption that the reference period represents stable climatic conditions and the current firn layer is in equilibrium. However, Veldhuijsen et al. (2023) noted that the assumption of a steady-state firn layer "may be a poor assumption" in regions where precipitation has increased over the last centuries, such as the Antarctic Peninsula and Ellsworth Land.

Both firn models use the same semi-empirical equation of Arthern et al. (2010) to model dry-snow densification but their procedure for deriving the empirical correction terms differs. IMAU derives this empirical correction from observations in Antarctica, while GSFC employs observations from both Antarctica and Greenland. Furthermore, the two firn models use a





different parameterisation for surface snow density. Veldhuijsen et al. (2023) use the formulation of Lenaerts et al. (2012), which depends on instantaneous surface temperature and $10\,\mathrm{m}$ wind speed, but with updated constants derived from their own calibration. Medley et al. (2022a) built a new model depending on snow accumulation, air temperature, total wind speed, and specific humidity. In general, they follow the approach from Helsen et al. (2008), which incorporates mean annual parameters. Both firn models take into account the processes of meltwater percolation and refreezing.

## 2.3 Previous comparisons between altimetry and firn models

The data sets used in this study have been compared mainly on the basis of multi-year to decadal rates and seasonal amplitudes. The results of these comparisons are briefly summarised below.

Nilsson et al. (2022) reported that elevation change rates of TUD, JPL and Shepherd et al. (2019) are generally in good agreement and within their uncertainties (over 1992–2016). By excluding regions of dynamic imbalance, they also compared

rates between altimetric elevation changes and modelled firn thickness changes derived from the IMAU-FDM v1.1 (Ligtenberg et al., 2011) forced by ERA-Interim reanalysis data. In Dronning Maud Land and Enderby Land the thickening patterns of modelled and observed rates are in good overall agreement, except that the observed rates show stronger magnitudes than the modelled rates of FDM v1.1. However, in the region of Wilhelm II Land and Wilkes Land the differences between observed and modelled rates are larger. Rates are of opposite sign (altimetry: positive rates; FDM v1.1: negative rates). The three altimetry

products agree in magnitude and sign (Nilsson et al., 2022). Recently, the update from FDM v1.1 to v1.2A (forced by ERA5 reanalysis data; Section 2.2) lead to an improved agreement with the observed rates (evaluated by the TUD product over 2003–2015). With the update, the modelled rates were found to be more positive in Dronning Maud Land and Enderby Land and less negative in Wilhelm II Land and Wilkes Land. However, discrepancies between altimetric and modelled rates remain, in particular for the Antarctic Peninsula and Ellsworth Land (Veldhuijsen et al., 2023).

Estimates of the average ice sheet seasonal amplitude in firn thickness give discrepant results for different altimetry products, firn models and time periods (with the latter also involving different spatial coverage): 5.1, 2.7 and $2.9\,\mathrm{cm}$ for TUD altimetry, JPL altimetry and the IMAU-FDM v1.1, respectively, over 1992–2016 (Nilsson et al., 2022), and, 5.2, 3.1 and $3.0\,\mathrm{cm}$ for TUD altimetry, the IMAU firn model (v1.2A) and the GSFC firn model, respectively, over 2003–2015 (Veldhuijsen et al., 2023).

## 2.4 Illustration of data sets

In this section, we illustrate and compare the original data sets through basin-mean time series and root mean square (rms) maps. This recalls typical ways of previous comparisons between altimetry and firn modelling results and serves as a reference for our subsequent exploration of a wider range of spatio-temporal scales.

Fig. 1 (dash-dotted lines) shows the basin-mean time series for the original elevation changes from TUD altimetry, $h^{\mathrm{A1}}$, and the IMAU firn model, $f^{\mathrm{Ma}}$. (For the JPL altimetry, $h^{\mathrm{A2}}$, and the GSFC firn model, $f^{\mathrm{Mb}}$, similar time series are shown in

Fig. S1.) Agreement between $h^{\mathrm{A1}}$ and $f^{\mathrm{Ma}}$ is generally good on interannual scales. Differences appear in the long-term trends. The trend differences are greatest for basin 10 (Amundsen Sea Embayment region). This is due to the effect of changing ice flow (Mouginot et al., 2014; Gardner et al., 2018; Diener et al., 2021), reflected in $h^{\mathrm{A1}}$, while this effect is purposely not





**Figure 1.** Basin-mean time series of the original elevation changes from the IMAU firn model (Ma), $f^{\mathrm{Ma}}$, (dash-dotted, black line) and from TUD altimetry (A1), $h^{\mathrm{A1}}$, (dash-dotted, cyan line). Basin-mean time series of modelled firn thickness variations from Ma, $fv^{\mathrm{Ma}}$, (solid, black line) and of adjusted firn thickness variations based on A1a, $fv^{\mathrm{A1a}}$, (solid, cyan line).





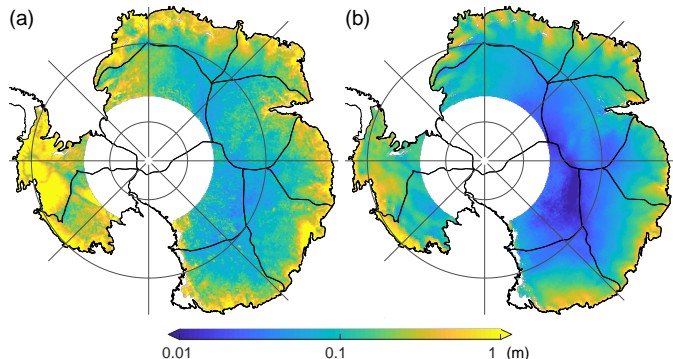

**Figure 2.** Root mean square (rms) of the original time series of elevation change over the entire period for (a) TUD altimetry, $h^{A1}$, and (b) the IMAU firn model, $f^{Ma}$. The color scale is logarithmic.

considered by $f^{Ma}$. A further difference is that, prior to 2003 the seasonal amplitudes of altimetry exceed those of the firn model (Ligtenberg et al., 2012; Nilsson et al., 2022).

Fig. 2 shows maps of rms values for $h^{A1}$ and $f^{Ma}$ over the entire period 1993-2017. The rms values include the effect of a linear component which dominates e.g. $h^{A1}$ in the Amundsen Sea Embayment region. Besides long-term influences, the overall spatial patterns of $h^{A1}$ and $f^{Ma}$ are related to the spatial variability of the SMB, with values increasing from the AIS interior to the margin (Van Wessem et al., 2014; Lenaerts et al., 2019). The rms values of $h^{A1}$ are generally larger than those of $f^{Ma}$. This is mainly due to the higher noise level in altimetry measurements before 2003 (Schröder et al., 2019a; Nilsson et al., 2022).

For the period after 2003, the rms values of altimetry and the firn model are in better agreement (Fig. S2). (Fig. S2 also shows the rms for all data sets used (A1, A2, Ma, Mb) separately for the periods before and after 2003.)

## 3    Methods

### 3.1    Regression approach

We jointly analyse satellite altimetry and firn modelling results while focusing on interannual to decadal time scales. The new

combination approach is a regression of altimetric elevation changes against several signals explained in the following. For





each $10\,\mathrm{km}$ x $10\,\mathrm{km}$ grid cell, we describe the time series of monthly elevation changes from altimetry $h^{\mathrm{A}}$ by

$$
\begin{aligned}
h^{\mathrm{A}}(t) = {} & a + bt + c\,(0.5\,t^2) \\
& + \mathrm{H}_1(t)\left[d_1\cos(\omega t) + d_2\sin(\omega t) + d_3\cos(2\omega t) + d_4\sin(2\omega t)\right] \\
& + \mathrm{H}_2(t)\left[d_5\cos(\omega t) + d_6\sin(\omega t) + d_7\cos(2\omega t) + d_8\sin(2\omega t)\right] \\
& + \sum_{n=1}^{N} e_n^{\mathrm{A}}\, PC_n^{\mathrm{M}}(t) \\
& + r^{\mathrm{A}}(t)
\end{aligned}
\tag{1}
$$

with $\mathrm{H}_1(t) = \begin{cases} 1, & \text{if } t < 2003 \\ 0, & \text{if } t > 2003 \end{cases}$ and $\mathrm{H}_2(t) = \begin{cases} 0, & \text{if } t < 2003 \\ 1, & \text{if } t > 2003. \end{cases}$

The regression parameters $a$ (offset), $b$ (linear trend), $c$ (acceleration), $d_{1,\dots,8}$ (amplitudes of annual and semi-annual harmonic signals, with $\omega = 2\pi/1\,\mathrm{yr}$) and $e_{1,\dots,N}^{\mathrm{A}}$ (scaling factors for dominant temporal patterns in modelled firn thickness variations) are estimated by least squares adjustment. They are adjusted w.r.t. the reference epoch, $t_0$, September 2010 no matter of data coverage. The definition of $N$ dominant temporal patterns in modelled firn thickness variations $PC_1^{\mathrm{M}}, \dots, PC_N^{\mathrm{M}}$ depends on the drainage basin to which the considered location belongs. It is explained in Section 3.1.1. The residuals $r^{\mathrm{A}}$ are the difference between the elevation changes $h^{\mathrm{A}}$ and the fitted model.

Seasonal signals are modelled by annual and semi-annual cosine and sine functions. By applying the masks $\mathrm{H}_1$ and $\mathrm{H}_2$, we fit different seasonal amplitudes for the time periods before and after 2003. In this way we account for the inconsistency in the seasonal amplitudes between the older pulse-limited radar altimetry missions (ERS-1, ERS-2) and the newer missions of different techniques (Envisat, ICESat, CryoSat-2) (Nilsson et al., 2022). The corrections for the influence of the ice sheet surface dielectric properties on the radar return signal (Section 2.1) are only partly able to reduce artificially large seasonal amplitudes in particular for the older missions (Ligtenberg et al., 2012).

We subtract the adjusted offset, linear, quadratic and seasonal signals from $h^{\mathrm{A}}$ to derive elevation changes on interannual time scales from altimetry according to

$$
\begin{aligned}
hv^{\mathrm{A}}(t) = {} & h^{\mathrm{A}}(t) - \\
& \{a + bt + c\,(0.5\,t^2) \\
& + \mathrm{H}_1(t)\left[d_1\cos(\omega t) + d_2\sin(\omega t) + d_3\cos(2\omega t) + d_4\sin(2\omega t)\right] \\
& + \mathrm{H}_2(t)\left[d_5\cos(\omega t) + d_6\sin(\omega t) + d_7\cos(2\omega t) + d_8\sin(2\omega t)\right]\}
\end{aligned}
\tag{2}
$$

The interannual elevation changes are termed altimetric variations, $hv^{\mathrm{A}}$.

We perform a weighted regression. Observations $h^{\mathrm{A}}$ after 2003 are weighted by 1, while observations prior to 2003 are given a different (usually lower) weight, according to the finding of a generally higher noise level of the results from the older altimetry missions (Schröder et al., 2019a; Nilsson et al., 2022). The weight prior to 2003 is defined (individually for every





grid point) by the ratio of the noise variance of $h^{\mathrm{A}}$ prior to 2003 and after 2003. The noise variance ratio is assessed empirically from the variance of the high-pass filtered time series (cf. Groh et al., 2019). The high-pass filtering (performed separately for the period prior to 2003 and after 2003) consists in removing linear and seasonal signals and subsequently removing a low-pass filtered version of the time series, where the low-pass filter is a Gaussian filter with a $6\sigma = 12$ months filter width.

### 3.1.1 Adjusted firn thickness variations

Our regression approach relies on the ability of firn models to capture the timing of dominant variations in SMB and firn processes across basins. However, the amplitudes and spatial patterns of the variations are adjusted to satellite altimetry results. We trust the temporal more than the spatial patterns of the firn model for the following reasons. Mottram et al. (2021) as well as Lenaerts et al. (2019) and Gutiérrez et al. (2021) have pointed out that the spatial patterns of RCMs, which force firn models, show a large spread between models but not their temporal patterns (Section 1.2). While spatially resolved differences (between

models, between observations and between models and observations) are substantial, the differences have been shown to be reduced when basin averages are used (Agosta et al., 2019; Shepherd et al., 2019; Willen et al., 2021). Also, the overall good agreement of basin-mean time series on interannual scales (between the data sets used here) has also been noted in Section 2.4, Fig. S1.

For each grid cell, the adjusted firn thickness variations $fv^{\mathrm{A}}$ are determined by the linear combination in Eq. 1:

$$fv^{\mathrm{A}}(t) = \sum_{n=1}^{N} e_n^{\mathrm{A}} PC_n^{\mathrm{M}}(t). \tag{3}$$

The dominant temporal patterns in firn thickness variations, $PC_n^{\mathrm{M}}$, are identified by principal component analysis (PCA) of the firn modelling data. PCA, also called empirical orthogonal function (EOF) analysis, is applied to identify dominant modes of variability, represented by pairs of a principal component (PC) and an EOF, where EOFs and the corresponding, uncorrelated PCs represent the spatial and temporal patterns, respectively. Comprehensive and general references for PCA are Preisendorfer

(1988) and Jolliffe (2002), while e.g. Forootan and Kusche (2012) or Boergens et al. (2014) apply PCA and extensions of PCA to geodetic data.

Prior to applying PCA to the firn modelling data, we remove offset, linear, quadratic and seasonal signals from the modelled firn thickness changes $f^{\mathrm{M}}$ according to

$$f^{\mathrm{M}}(t) = a + bt + c\,(0.5\,t^2)$$

$$+ d_1 \cos(\omega t) + d_2 \sin(\omega t) + d_3 \cos(2\omega t) + d_4 \sin(2\omega t)$$

$$+ fv^{\mathrm{M}}(t), \tag{4}$$

where $a, b, c, d_{1,\dots,4}$ are estimated by an ordinary least-squares adjustment. The residuals, $fv^{\mathrm{M}}$, are referred to as firn thickness variations. The PCA is performed on these firn thickness variations after their standardisation. We standardise the time series of $fv^{\mathrm{M}}$ for each grid cell, i.e. we shift and scale it that it has zero mean and a std of one, because we aim to equally represent

the patterns of temporal evolution regardless of location or absolute amplitudes. Otherwise, PCA results would mainly reflect


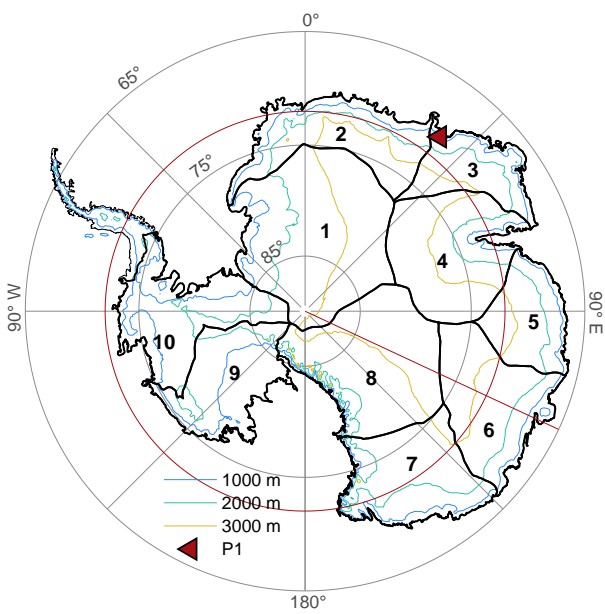

**Figure 3.** Drainage basins of the EAIS and WAIS used in this study (thick black lines) following Rignot et al. (2011a, b). The outline of Antarctic Peninsula is indicated by a thin black line. Contour lines of the ice sheet surface are shown at $1000\,\mathrm{m}$, $2000\,\mathrm{m}$ and $3000\,\mathrm{m}$. Highlighted in red are the circle at constant latitude of $72°\,\mathrm{S}$ (profile 1) and the line at constant longitude at $115°\,\mathrm{E}$ (profile 2). Grid point P1 is located at $\mathrm{lon} = 37.7°\,\mathrm{E}$, $\mathrm{lat} = 70.2°\,\mathrm{S}$. We use the polar stereographic projection EPSG:3031 (WGS84, latitude of true scale: $71°\,\mathrm{S}$, central meridian: $0°$). All further maps are displayed in the same projection and with the same spacing of longitude grid lines (every $45°$) and latitude grid lines (every $10°$).

patterns that are dominant at the margins as SMB and firn thickness variations exhibit much larger amplitudes at the margins than in the interior (van Wessem et al., 2018; Lenaerts et al., 2019).

PCA is applied individually to $fv^{\mathrm{M}}$ for 10 selected regions (Fig. 3). To define the regions, we make use of the drainage basin definition by Rignot et al. (2011a, b). We aggregate basins smaller than $\sim 600,000\,\mathrm{km}^2$ with those neighbouring basins where we find strongest correlation between their first three PCs. This step reduces the original number of 15 drainage basins for the EAIS and WAIS to 10. For each (aggregated) basin, we choose the first $N$ modes that contain at least $90\,\%$ of the total variance of the (standardised) data. In addition, North's rule of thumb (North et al., 1982) is applied to test whether the eigenvalues of these $N$ patterns are well separated with respect to their errors. The first $N$ dominant temporal patterns $PC_n^{\mathrm{M}}$ enter Eq. 1 in normalised form.

### 3.1.2 Goodness of fit

To examine how well a regression fits the observations, we calculate the coefficient of determination, $R^2$ ('R squared'), as

$$R^2 = 1 - \frac{SS(r)}{SS(h_{\mathrm{tot}})}. \tag{5}$$





**Table 1.** Names of different versions of adjusted firn thickness variations, $fv^A$, derived by applying the regression approach Eq. 1 with different data sets. Differences to A1a are indicated in bold.

| Name | $h^A$ from | $PC^M$ from |
|------|-----------|-------------|
| A1a | TUD altimetry (A1) | IMAU firn model (Ma)* |
| A**2**a | **JPL** altimetry (A**2**) | IMAU firn model (Ma)* |
| A1**b** | TUD altimetry (A1) | **GSFC** firn model (**Mb**)* |
| A**2b** | **JPL** altimetry (A**2**) | **GSFC** firn model (**Mb**)* |

* standardised $fv^M$ (Section 3.1.1)

$SS(r)$ and $SS(h_{\text{tot}})$ are the residual and total sum of squares, respectively. $SS(r)/SS(h_{\text{tot}})$ describes the proportion of unexplained variance. Here, we calculate $R^2$ for every grid cell individually and exclude the adjusted linear, seasonal and quadratic signals in $h_{\text{tot}}$. Thus, Eq. 5 specifies to

$$R_A^2 = 1 - \frac{SS(r^A)}{SS(hv^A)} = 1 - \frac{SS(r^A)}{SS(fv^A + r^A)}. \tag{5a}$$

### 3.2 Different versions of adjusted firn thickness variations

We derive two different sets of $PC^M$ depending on the firn model incorporated. In our annotation, we distinguish the firn models by superscripts 'Ma' and 'Mb' for the IMAU and GSFC model, respectively. The regression approach (Eq. 1) is applied with each set of $PC^M$ and equally to each of the two altimetry products $h^A$ from TUD and JPL, which we distinguish by superscripts 'A1' and 'A2', respectively. Thus, depending on the combination of data sets used, we obtain four versions of adjusted firn thickness variations ($fv^{A1a}$, $fv^{A2a}$, $fv^{A1b}$ and $fv^{A2b}$). This also results in four versions of altimetric residuals ($r^{A1a}$, $r^{A2a}$, $r^{A1b}$ and $r^{A2b}$) and of associated coefficients of determination ($R_{A1a}^2$, $R_{A2a}^2$, $R_{A1b}^2$ and $R_{A2b}^2$). Table 1 gives an overview of the applications of the regression approach.

We additionally fit a regression similar to Eq. 1 to the firn model data, $f^M$, after their interpolation to the altimetric grid of $10\,\text{km}$ spacing. The same deterministic model Eq. 1 is used, but no weighting is applied. In this way, the regression parameters $a, b, c, d_{1,...,8}$ and in particular the scaling factors adjusted to altimetry, $e_n^A$, can be directly compared to the scaling factors derived from the firn models, $e_n^M$. Replacing $e_n^A$ by $e_n^M$ in Eq. 3 would then lead to a variant of modelled firn thickness variations, restricted to the dominant temporal modes found in the PCA. We refer to this variant as truncated modelled firn thickness variations, $fv_{90}^M$. The suffix '90' indicates that the dominant patterns were chosen such that they cover at least $90\,\%$ of the variance of standardised time series within the specific basin.

In Appendix A1, we additionally assess three alternative ways of defining 'adjusted' firn thickness variations. These alternatives are: (E1) Accept the modelled firn thickness variations, $fv^M$, without any adjustment to altimetry. (E2) Instead of using PCA-based dominant temporal patterns use the modelled time series of firn thickness variations at every grid cell and scale it to fit the altimetry. These alternative variations are called scaled firn thickness variations. We refer to them by $fv^{E2}$. (E3) Identify the dominant temporal patterns of modelled firn thickness variations by a PCA without prior standardisation of the time series.





These alternative variations are called modified adjusted firn thickness variations. We refer to them by $fv^{\mathrm{E3}}$. See Table B1 for an overview of the defined symbols and their terminology.

### 3.3 Assessment methods

#### 3.3.1 Uncertainty of adjusted firn thickness variations

To assess the impact of the choice of data sets and thus the influence of different errors on $fv^{\mathrm{A}}$, differences between time series of the various versions of firn thickness variations, $fv$, (Section 3.2) are used. In general, from each time series of differences we can calculate the temporal root mean square (rms). This procedure is applied to time series differences evaluated for every grid cell and also for differences in basin-mean time series.

To assess the uncertainty of the adjusted firn thickness variations, $fv^{\mathrm{A}}$, we consider the maximum deviation within the different versions of $fv^{\mathrm{A}}$ (Table 1). For this purpose, we form all possible combinations of differences between the four versions of $fv^{\mathrm{A}}$. It results in six combinations of time series differences and thus, six (temporal) rms values, where we choose the one that is maximum.

#### 3.3.2 Robustness of adjusted firn thickness variations

The adjusted firn thickness variations, $fv^{\mathrm{A}}$, can be considered an improved representation of firn thickness variations compared with the modelled variations, $fv^{\mathrm{M}}$, if we can statistically demonstrate that the differences within different versions of $fv^{\mathrm{A}}$ are significantly smaller than the differences to $fv^{\mathrm{M}}$. To investigate this, we perform statistical tests comparing distributions of temporal rms of differences within $fv^{\mathrm{A}}$ to differences $fv^{\mathrm{A}} - fv^{\mathrm{M}}$.

    We work with a two-sample, one-sided Kolmogorov-Smirnov test which is a non-parametric hypothesis test as the dif-
ferences in $fv$ do not follow a normal distribution. The Kolmogorov-Smirnov test uses the empirical cumulative distribution function (cdf), which is the integral of the probability density function (pdf), to compare the distributions of two samples. The null hypothesis (H0) reads: both samples, the data of both differences to be compared, are from the same continuous distribution. Thus, the alternative hypothesis (H1) reads: the empirical cdf of sample one (the differences within $fv^{\mathrm{A}}$), is larger than the empirical cdf of sample two (the differences between $fv^{\mathrm{A}}$ and $fv^{\mathrm{M}}$), that is the differences within $fv^{\mathrm{A}}$ tend to be smaller than
the differences between $fv^{\mathrm{A}}$ and $fv^{\mathrm{M}}$.

#### 3.3.3 Spectral analysis of regression results

The altimetric residuals, $r^{\mathrm{A}}$, and the adjusted firn thickness variations, $fv^{\mathrm{A}}$, are analysed in the spectral domain to characterise their stochastic properties. We calculate the power spectral density (psd) and the spectral indices, $\kappa$, of the underlying time series of $r^{\mathrm{A}}$ and $fv^{\mathrm{A}}$. We use the software HECTOR v1.7.2 (Bos et al., 2012) to estimate $\kappa$. As $r^{\mathrm{A}}$ and $fv^{\mathrm{A}}$ do not yield a
white noise behaviour we use the formulation of power-law noise to approximate their stochastic properties. (For example, power-law with $\kappa = -1$ and $\kappa = -2$ represents flicker and random walk noise, respectively.)





### 3.3.4 Principal component analysis of altimetric residuals

The four versions of altimetric residuals (Section 3.2) are further analysed in the spatio-temporal domain. First, we perform PCA on the altimetric residuals themselves to further identify dominant signals related to ice sheet processes not considered or incorrectly represented by the firn models. (Note that the residuals may additionally contain signals related to variations in ice flow dynamics or subglacial hydrology.) Second, we perform PCA on the residual differences to further detect and investigate prevailing uncertainties in altimetry analysis. Only data after 2003 is used because of the higher noise level in the altimetry measurements of the older satellite missions. Inclusion of pre-2003 data would result in more noisy dominant patterns and therefore could distort detected dominant modes. We standardise the time series of residuals and residual differences, as we did previously when identifying dominant patterns in modelled firn thickness variations (Section 3.1.1).

The first PCA is applied to four versions of standardised residuals ($r^{\mathrm{A1a}}$, $r^{\mathrm{A1b}}$, $r^{\mathrm{A2a}}$ and $r^{\mathrm{A2b}}$). The second PCA is applied to two versions of standardised residual differences ($r^{\mathrm{A1a}} - r^{\mathrm{A2a}}$ and $r^{\mathrm{A1b}} - r^{\mathrm{A2b}}$). For each PCA, we set up one aggregated 'super data matrix' in which we arrange the time series of residuals/residual differences for all pixels and for the different versions into a single set of time series. PCA is conducted to identify the dominant temporal patterns, which are shared by all versions, together with their space-dependent and version-dependent amplitudes, i.e. their spatial patterns.

## 4 Results

### 4.1 Dominant patterns in modelled firn thickness variations

Fig. 4 shows the PCA results for the example of basin 3 and the IMAU firn model input data, $fv^{\mathrm{Ma}}$. The figure shows the dominant spatial patterns (EOFs) and temporal patterns (PCs) together with their share of the total variance. We recall that PCA is performed individually for each basin and that $fv^{\mathrm{Ma}}$ are standardised prior to PCA. A comprehensive presentation of results for all basins and for the two alternative input firn models IMAU and GSFC is given by Fig. S3 –S7. Depending on the basin, different numbers of modes (i.e. PC–EOF pairs) are required to explain at least $90\,\%$ of the total variance: two modes for basin 5, three modes for basins 1, 3 and 6, four modes for basins 2, 4 and 8 and five modes for basins 7, 9 and 10 (based on Ma). The first, second, third, fourth and fifth modes describe 58 to $74\,\%$, 11 to $21\,\%$, 4 to $12\,\%$, 3 to $5\,\%$ and 3 to $4\,\%$ of the data variance, respectively (based on Ma).

The PCs and EOFs reveal a typical hierarchy of modes of an autocorrelated geophysical signal. The first temporal patterns, $PC_1^{\mathrm{M}}$, show a longer wavelength signal than the following PCs. The first EOFs show an approximately uniform distribution, while the following EOFs are more complex and change sign. For basin 3, the first three EOFs exhibit a uniform behaviour, a north-south gradient and an east-west gradient, respectively (Fig. 4). The first mode of basins 2 and 3 (the region of Dronning Maud Land and Enderby Land) capture the accumulation events in 2009 and 2011 (Boening et al., 2012; Lenaerts et al., 2013). Their temporal patterns, $PC_1^{\mathrm{M}}$, show a characteristic increase during these years (Fig. 4). All subsequent modes are more difficult to interpret as a geophysical signal because of the fact that their determination is governed by the mathematical orthogonality property of PCs.





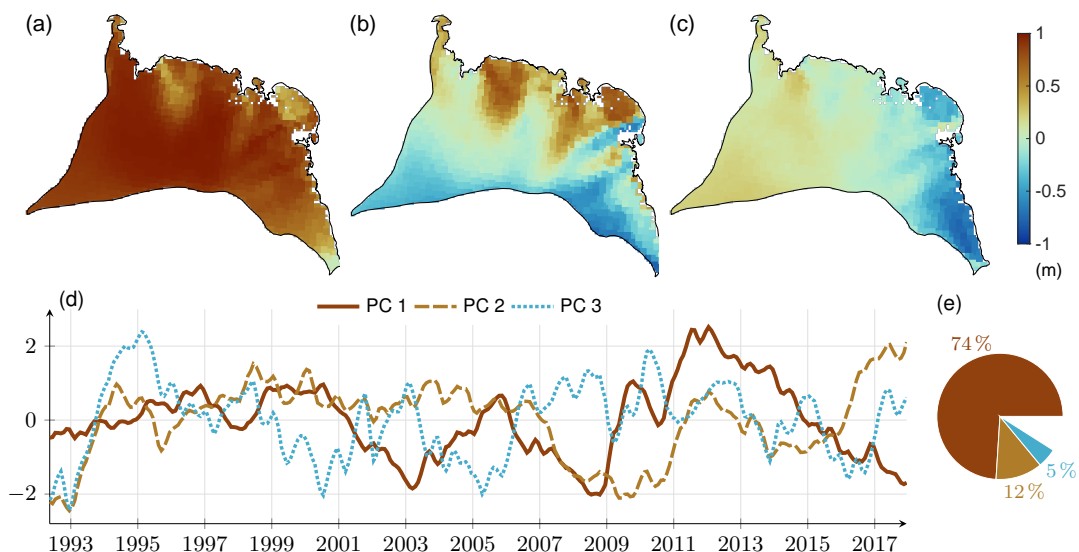

**Figure 4.** PCA results of basin 3: dominant patterns in firn thickness variations identified from standardised firn modelling data (Ma). (a, b, c) First, second and third spatial pattern (EOF). (d) First three temporal patterns (PCs). (e) Associated percentages of the basin's total data variance. We define the PCs as standardised time series (mean of zero, std of 1) and without a unit while the EOFs have the unit of metre.

## 4.2 Regression results

### 4.2.1 Time series for a selected grid point

Fig. 5 exemplifies the derivation of adjusted firn thickness variations for a selected grid point, P1, and based on the regression A1a (Table 1). P1 ($37.7°$ E, $70.2°$ S) is located in basin 3 close to the ice sheet margin at $\sim 1080\,\mathrm{m}$ height (Fig. 3). There, the adjusted and modelled firn thickness variations, $fv^{\mathrm{A1a}}$ and $fv^{\mathrm{Ma}}$, have a standard deviation (std) of $41.0$ and $51.5\,\mathrm{cm}$, respectively (Fig. 5b). In addition to $fv^{\mathrm{Ma}}$, we illustrate the time series of truncated modelled firn thickness variations, $fv^{\mathrm{Ma}}_{90}$ (Section 3.2), which has a std of $49.1\,\mathrm{cm}$. The difference between $fv^{\mathrm{Ma}}$ and $fv^{\mathrm{Ma}}_{90}$ equals $r^{\mathrm{Ma}}$ and is shown in Fig. 5f.

By construction, the scaling factors $e_{1,\ldots,3}$ equal the std of the respective scaled dominant temporal patterns. (In the case of data gaps in the altimetry time series, this equality holds approximately.). Both $fv^{\mathrm{A1a}}$ and $fv^{\mathrm{Ma}}_{90}$ are dominated by $PC^{\mathrm{M}}_1$ of basin 3, as this pattern is scaled by $e^{\mathrm{A1a}}_1 = 39.6\,\mathrm{cm}$ (altimetry) and $e^{\mathrm{Ma}}_1 = 48.4\,\mathrm{cm}$ (firn model). For $e_2$, altimetry and the firn model have opposite signs, yet small values, so that they contribute little to $fv$.

The std of altimetric residuals $r^{\mathrm{A1a}}$ is $31.0\,\mathrm{cm}$, less than the std of $fv^{\mathrm{A1a}}$. The coefficient of determination, $R^2_{\mathrm{A1a}}$ (Eq. 5a) is $0.61$. When calculated separately for the time before and after 2003, $R^2_{\mathrm{A1a}}$ equals $-0.06$ and $0.84$, respectively. Thus, the adjusted firn thickness variations, $fv^{\mathrm{A1a}}$, describe less of the variance of altimetric variations before 2003 while after 2003 they explain $84\,\%$. Distinguishing the time before and after 2003 is reasonable as we include different weights for the altimetry observations before and after 2003 (Section 3.1).



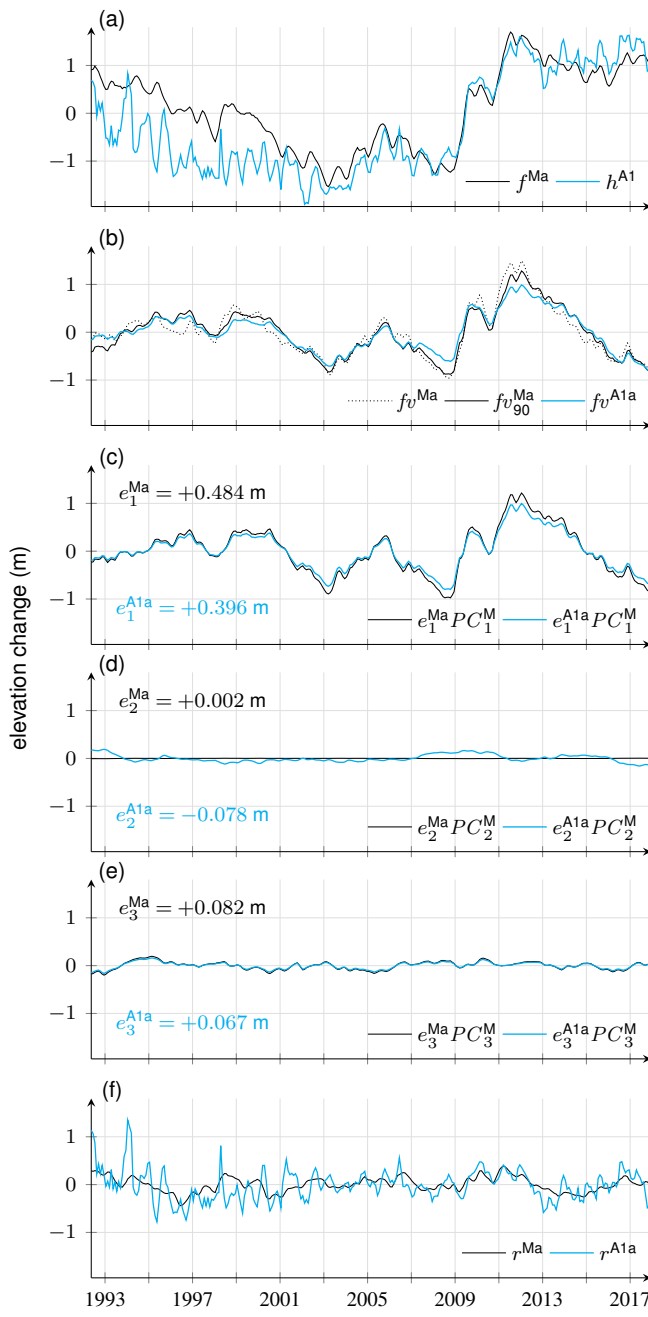

**Figure 5.** Illustration of the generation of adjusted firn thickness variations $fv^{\mathrm{A}}$ for the grid point P1 (Fig. 3). Cyan and black curves show regression results from the adjustment to TUD altimetry (A1a) and, for a direct comparison, to the IMAU firn model (Ma), respectively. (a) Original time series, $h^{\mathrm{A1}}$ and $f^{\mathrm{Ma}}$. (b) Modelled firn thickness variations, $fv^{\mathrm{Ma}}$ (dashed, black), truncated modelled firn thickness variations, $fv_{90}^{\mathrm{Ma}}$ (solid, black), and adjusted firn thickness variations, $fv^{\mathrm{A1a}}$ (solid, cyan). (c), (d), (e) Scaled first, second, and third dominant temporal patterns in $fv^{\mathrm{Ma}}$. Hence, the solid black/cyan curve in (b) is the sum of the black/cyan curves in (c–e). (f) Time series of the regression residuals. The black curve ($r^{\mathrm{Ma}}$) equals $fv^{\mathrm{M}} - fv_{90}^{\mathrm{Ma}}$.





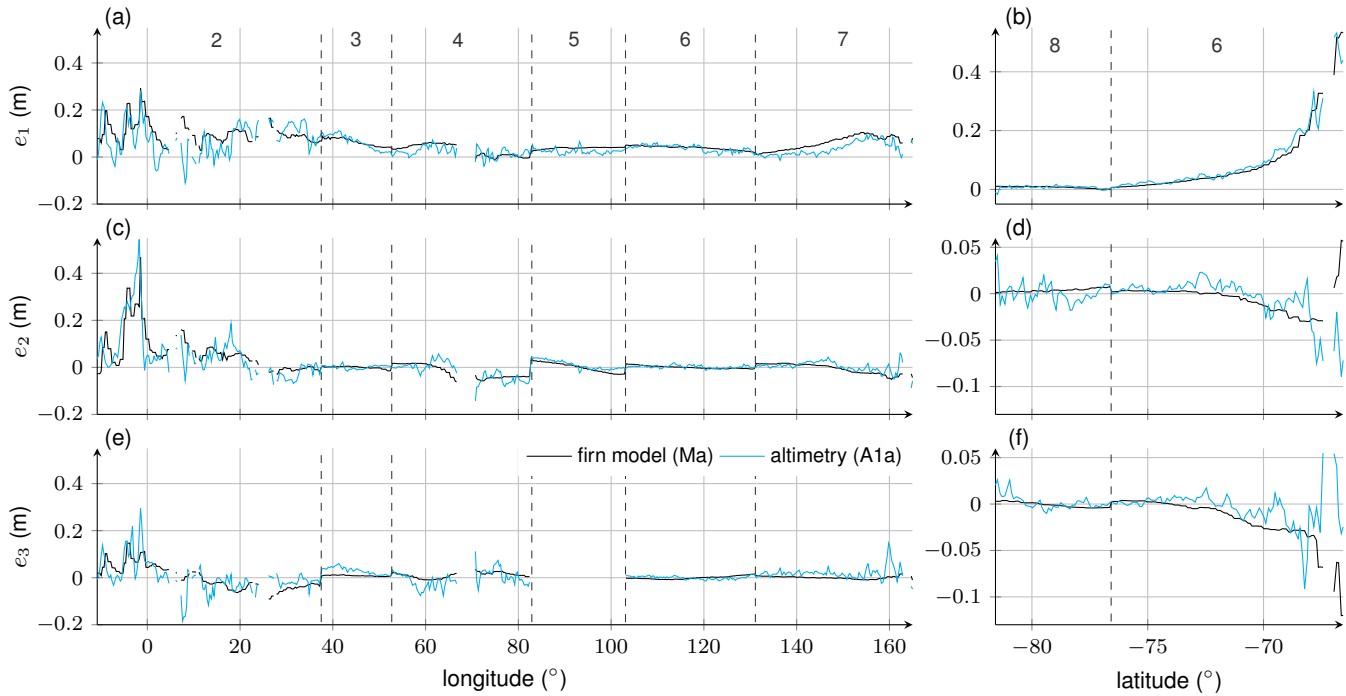

**Figure 6.** Adjusted scaling factors along profile 1 (left) and profile 2 (right). (a, b) $e_1$, (c, d) $e_2$ and (e, f) $e_3$. Cyan and black curves show the scaling factors adjusted to TUD altimetry (A1a) and to the IMAU firn model (Ma), respectively. Note the different scaling of the y-axes of profile 2.

For a larger subset of selected grid points (Fig. S8), time series of the original elevation changes $h$ and the regression results are shown in Fig. S9–S12. While the following Section 4.2.2 focuses on the adjusted scaling factors $e_{1,...,N}$, maps of the other regression parameters (adjusted linear, quadratic and seasonal terms) are presented in Fig. S13–S15.

### 4.2.2   Scaling factors $e$

Fig. 6 shows the spatial variation of the scaling factors $e_{1...3}$ along two selected profiles marked in Fig. 3. Profile 1 is along
the circle of latitude at $72°$ S. Profile 2 is along the meridian at $115°$ E. The absolute magnitude of both scaling factors (from A1a and Ma) is largest at the ice sheet margin. This applies for profile 1 across basin 2, in the middle part of basin 4 and at the end part of basin 7 as well as for profile 2 at the end part of basin 6. Observed factors, $e_{1...3}^{\text{A1a}}$, reveal stronger variations along both profiles than modelled factors, $e_{1...3}^{\text{Ma}}$. Discontinuities across basin borders arise because the scaling factors refer to basin-specific patterns.

The scaling factors $e_{1...3}^{\text{A1a}}$ and $e_{1...3}^{\text{Ma}}$ per grid cell are mapped for the example of basin 3 in Fig. 7. The patterns of the factors, like the EOFs (Fig. 4), follow a typical hierarchy already discussed in Section 4.1. Overall, the patterns of $e_{1...3}^{\text{Ma}}$ are in a good agreement with $e_{1...3}^{\text{A1a}}$. However, the first spatial pattern from the model extends further towards the ice sheet interior than the pattern from altimetry. In general, scaling factors from the model show a smoother and more blurred pattern than the



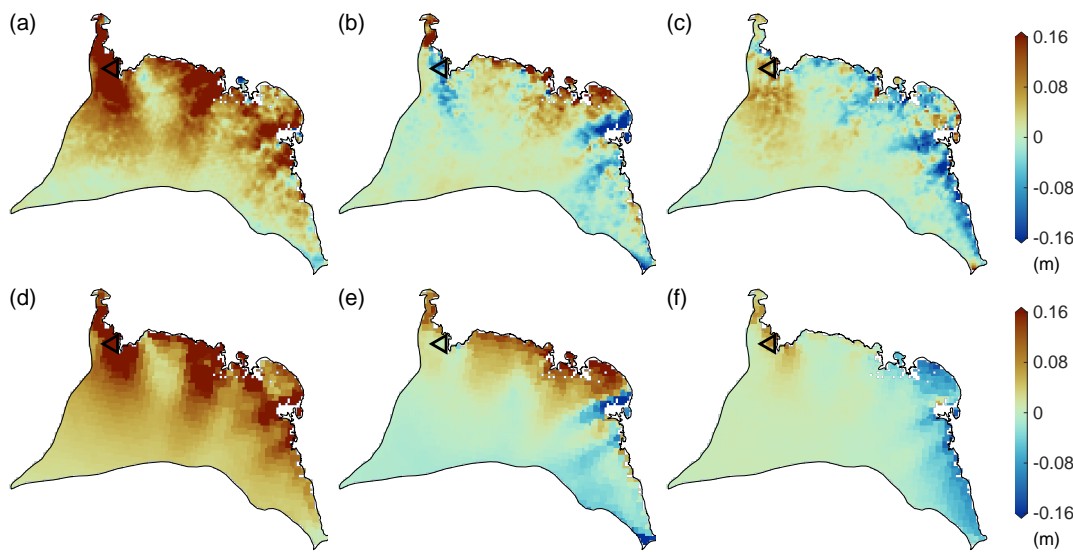

**Figure 7.** Adjusted scaling factors for basin 3. (a–c) $e_{1...3}^{\mathrm{A1a}}$, first three factors adjusted to TUD altimetry. (d–f) $e_{1...3}^{\mathrm{Ma}}$, first three factors adjusted to the IMAU firn model. The location of P1 is shown by the black triangle.

ones adjusted to altimetry. Patterns from altimetry reveal a higher level of detail and a more localised spatial distribution. At

certain regions the spatial distributions also differs. In the area at and around P1 (marked as a triangle), altimetry observes the second temporal pattern with a negative amplitude $e_2$, while the firn model suggests an amplitude near zero. A comprehensive presentation of scaling factors for all basins and for different choices of input data are given by Fig. S3 and S16.

### 4.2.3 Firn thickness variations and their sensitivity to the choice of data sets

We calculate the rms of the time series of firn thickness variations, $fv$, for each grid cell. Fig. 8a and 8b show the rms of adjusted

firn thickness variations based on A1a, $fv^{\mathrm{A1a}}$ (Table 1), and the rms of modelled firn thickness variations based on Ma, $fv^{\mathrm{Ma}}$, respectively. (The rms of all versions of $fv^{\mathrm{A}}$ and $fv^{\mathrm{M}}$ is illustrated in Fig. S17a–d and Fig. S18a, b, respectively.) In general, the spatial patterns of $fv^{\mathrm{A}}$ and $fv^{\mathrm{M}}$ are similar. Rms values are largest at the ice sheet margin and smallest over the plateau of the EAIS. For grid cells in the elevation ranges (1) below $1000\,\mathrm{m}$, (2) $1000$ to $2000\,\mathrm{m}$, (3) $2000$ to $3000\,\mathrm{m}$ and (4) above $3000\,\mathrm{m}$, median rms values are in the range of (1) $13.2$ to $16.4$, (2) $8.7$ to $10.9$, (3) $3.7$ to $5.1$ and (4) $2.2$ to $2.4\,\mathrm{cm}$, respectively. Fig. 8c

and 8d show the rms of the differences $fv^{\mathrm{A1a}} - fv^{\mathrm{Ma}}$ in an absolute and relative way, respectively. Differences between adjusted and modelled variations reveal highest absolute rms values at lower elevations, near the AIS margins (median rms differences in the range of $11.5$ to $12.7\,\mathrm{cm}$ below $1000\,\mathrm{m}$). In a relative sense, largest mismatch is found in the interior of the EAIS but also at some locations at the ice sheet margin.

To evaluate the sensitivity of $fv$ to the choice of data sets, we calculate the difference between various versions of $fv$

(Section 3.3.1). Fig. 9 shows the distributions of the rms of differences between various versions of $fv$. (Corresponding rms maps of differences are displayed in Fig. S17–S19). In addition to the distributions their median values are presented in



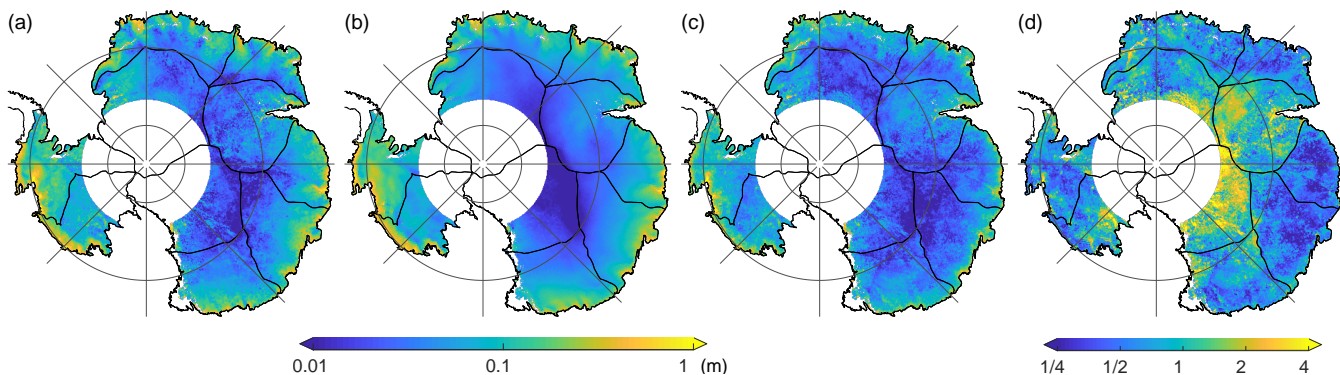

**Figure 8.** Root mean square (rms) of the times series of (a) adjusted firn thickness variations $fv^{\text{A1a}}$ and (b) modelled firn thickness variations $fv^{\text{Ma}}$. (c) Rms of the time series of the differences $fv^{\text{A1a}} - fv^{\text{Ma}}$. (d) Rms of the time series of the differences $fv^{\text{A1a}} - fv^{\text{Ma}}$ divided by the rms of $fv^{\text{Ma}}$.

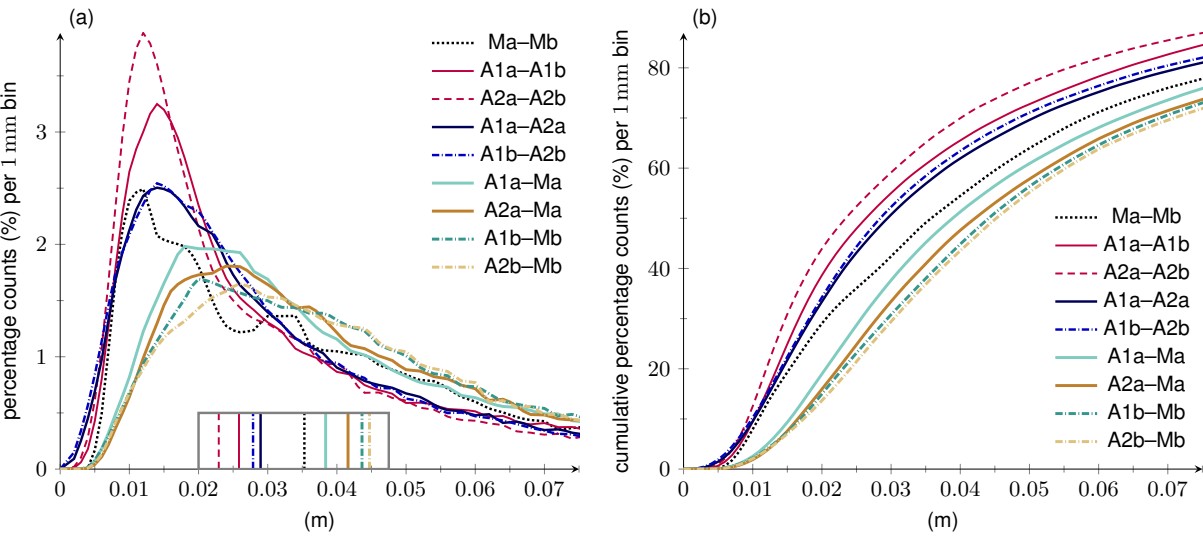

**Figure 9.** Histograms of the temporal rms, assessed at each grid cell, of differences between various versions of firn thickness variations. (a) Histograms. Vertical lines in the box indicate median values. (b) Cumulative histograms.





**Table 2.** Overview of the comparison between various versions of firn thickness variations, as detailed in Fig. 9. Column 1 indicates the addressed comparison: between versions of adjusted firn thickness variations $fv^A$ (row 1–4), between modelled firn thickness variations $fv^M$ (row 5), and between $fv^A$ and $fv^M$ (row 6–9). For each comparison, column 2 gives the median (over all grid cells) of the rms (over time) of differences between the two time series evaluated at each grid cell. The table is ordered by the median values (from small to large). Column 3 also gives the median of the rms of differences but as a relative measure. For each grid cell, the rms of differences are divided by the rms of $fv^{Ma}$. Then, the median over all grid cells is calculated. Column 4 gives a short description or possible causes.

| Difference | Median | | Description/Cause |
| --- | --- | --- | --- |
| | absolute | relative | |
| A2a−A2b | 2.3 cm | 0.46 | influence of different firn model setups based on A2 |
| A1a−A1b | 2.6 cm | 0.51 | influence of different firn model setups based on A1 |
| A1b−A2b | 2.8 cm | 0.55 | different altimetry analysis based on Mb |
| A1a−A2a | 2.9 cm | 0.58 | different altimetry analysis based on Ma |
| Ma−Mb | 3.5 cm | 0.65 | different firn model setups |
| A1a−Ma | 3.8 cm | 0.73 | Adjustment over Ma through A1* |
| A2a−Ma | 4.2 cm | 0.82 | Adjustment over Ma through A2* |
| A1b−Mb | 4.4 cm | 0.83 | Adjustment over Mb through A1* |
| A2b−Mb | 4.5 cm | 0.87 | Adjustment over Mb through A2* |

\* due to firn signals not correctly represented by the models (firn model errors) and/or due to errors in the altimetry products

Fig. 9 and listed in Table 2. In total, differences within $fv^A$ are smallest, followed by differences within $fv^M$ while differences between $fv^A$ and $fv^M$ are largest. Differences within $fv^A$ indicate a smaller influence by different firn model data than by different altimetry data. Differences between $fv^A$ and $fv^M$ are smallest for A1a (adjustment over the IMAU firn model through 445 TUD altimetry) and largest for A2b (adjustment over the GSFC firn model through JPL altimetry). The differences between the various versions of $fv$ reflect errors in the firn models and in the altimetry products. These are further discussed in Sections 5.3 and 5.4.

#### 4.2.4 Goodness of fit

The rms of the altimetric residual time series is presented in Fig. 10a (estimated per grid cell over the full period). The altimetric 450 residuals are used to calculate the goodness of fit (Section 3.1.2). Here, we distinguish between the periods before and after 2003. As mentioned in Section 4.2.1, this is useful due to the different noise levels and weighting of altimetry observations during the two periods (Section 3.1). The rms of the residuals after 2003 (Fig. 10b) are generally smaller than from the ones over the full period.





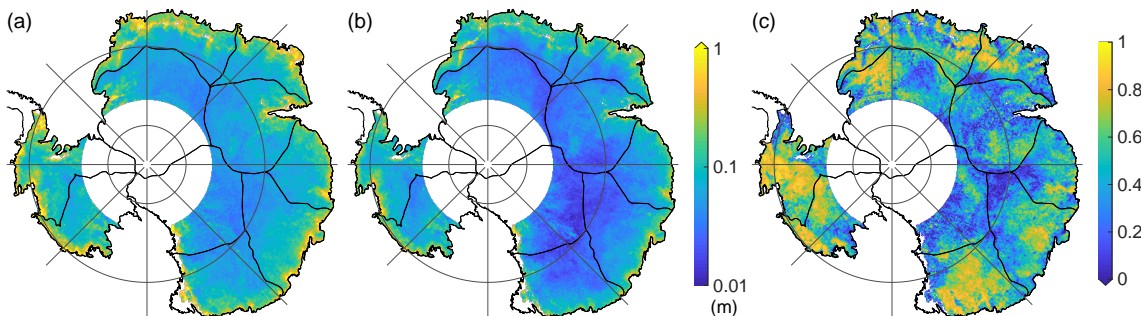

**Figure 10.** (a, b) Rms of the residual altimetric time series, $r^{\text{A1a}}$, for (a) the full period and (b) the period after 2003. (c) Coefficients of determination for the regression A1a, $R^2_{\text{A1a}}$, considering the period after 2003.

**Table 3.** Explained variance or coefficients of determination, $R^2$, for each basin and each version of regression (Table 1) over the period after 2003. Apart from the last column $\overline{\text{A1a}}$, $R^2$ is first calculated for each grid cell according to Eq. 5a and then averaged over each basin. Values of $\overline{\text{A1a}}$ are calculated by first averaging the regression results over each basin and then applying Eq. 5a.

| Basin | A1a | A2a | A1b | A2b | $\overline{\text{A1a}}$ |
|-------|------|------|------|------|------|
| 01 | 0.46 | 0.43 | 0.41 | 0.36 | 0.79 |
| 02 | 0.53 | 0.48 | 0.49 | 0.42 | 0.94 |
| 03 | 0.48 | 0.48 | 0.48 | 0.46 | 0.94 |
| 04 | 0.36 | 0.41 | 0.27 | 0.33 | 0.51 |
| 05 | 0.40 | 0.39 | 0.36 | 0.39 | 0.79 |
| 06 | 0.42 | 0.29 | 0.36 | 0.30 | 0.82 |
| 07 | 0.57 | 0.47 | 0.51 | 0.41 | 0.94 |
| 08 | 0.30 | 0.37 | 0.30 | 0.37 | 0.66 |
| 09 | 0.57 | 0.50 | 0.53 | 0.47 | 0.97 |
| 10 | 0.62 | 0.56 | 0.56 | 0.48 | 0.97 |
| 01–10* | 0.46 | 0.43 | 0.42 | 0.39 | 0.83 |

\* refers to the entire area (considered as a single basin)

The spatial distribution of the coefficients of determination based on the regression A1a, $R^2_{\text{A1a}}$, and for the period after 2003 is displayed by Fig. 10c. After the individual calculation of $R_s$ for each grid cell, basin-mean values are derived and listed in Table 3 for all versions of regression. (Fig. S20 and Fig. S21 further shows maps of the residuals rms and of $R^2$ for different versions of regression and both time periods. Table S1 lists basin averages of $R^2$ for the period before 2003.) Averaged over the entire area, $R^2_{\text{A1a}}$ is 0.46 after 2003 (Table 3). This means that on average $46\,\%$ of the variance of altimetric variations is captured by the regression model, i.e. by $fv^{\text{A1a}}$. Depending on the basin, $fv^{\text{A1a}}$ capture $30\,\%$ (basin 8) to $62\,\%$ (basin 10) of the data variance. In general, the goodness of fit decreases slightly when using JPL altimetry instead of TUD altimetry (column





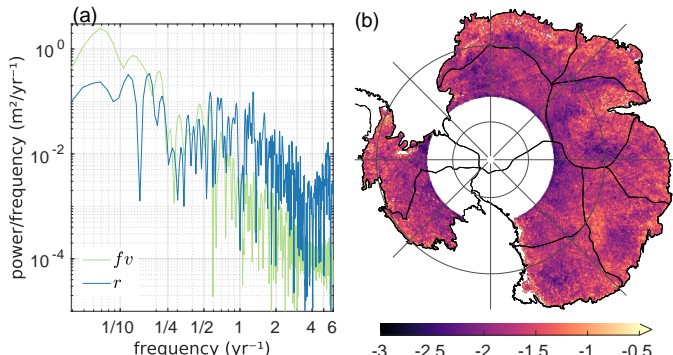

**Figure 11.** (a) Lomb-Scargle power spectral density (psd) of altimetric residuals $r^{\mathrm{A1a}}$ (blue) and adjusted firn thickness variations $fv^{\mathrm{A1a}}$ (green) for grid point P1. (b) Spectral index $\kappa$ for power-law noise adjusted to the residuals $r^{\mathrm{A1a}}$ of every grid cell.

A1a versus A2a and column A1b versus A2b of Table 3) or when incorporating the GSFC firn model instead of the IMAU firn model (column A1a versus A1b and column A2a versus A2b).

The impact of methodological changes to the regression approach (E1, E2 and E3 as summarised in Section 3.2) is presented in Appendix A2. There, Fig. A1 compares $R^2$ values of the modified approaches to $R^2_{\mathrm{A1a}}$ for each grid cell and Table A1 lists basin-averaged $R^2$ values of the modified approaches. The methodological changes result in smaller average $R^2$ values, so less of the data variance could be explained. For this reason, the modified approaches are not preferable to the chosen regression approach presented in Section 3.1.

By now, the presented $R^2$ values are based on calculations per grid cell in accordance with the regression approach Eq. 1. For basin average time series, $R^2$ become larger. Fig. 1 shows the basin-averages of adjusted firn thickness variations, which we may compare to the basin-averages of the altimetric variations through the coefficients of determination given in Table 3, last column. Indeed, $fv^{\mathrm{A1a}}$ could capture $51\,\%$ (basin 4) to $97\,\%$ (basins 9 and 10) of the variance of basin average altimetry variations. (Basin-mean time series of all regression results and versions are presented in Fig. S22–S24.) However, on the level of individual grid cells the altimetric residuals, $r^{\mathrm{A}}$, still contain a large proportion of the variance of altimetric variations. For example, for A1a and the period after 2003, an average ratio of $54\,\%$ of the altimetric variations are unexplained. Therefore, the residuals $r^{\mathrm{A}}$ are further investigated in the following Sections 4.3 and 4.4.

### 4.3 Spectral analysis of regression results

Fig. 11a shows the power spectral density (psd) of the altimetric residuals, $r^{\mathrm{A1a}}$, and the adjusted firn thickness variations, $fv^{\mathrm{A1a}}$, for the selected grid point P1. The underlying time series are displayed by Fig. 5. (For the larger subset of selected grid points, Fig. S25 and S26 display the psd of the regression results from A1a and A2a, respectively.) The psd of both $fv^{\mathrm{A1a}}$ and $r^{\mathrm{A1}}$ generally decreases from low to high frequencies. The slope of the psd is steeper for $fv^{\mathrm{A1a}}$ than for $r^{\mathrm{A1}}$. This means that the underlying time series of $fv^{\mathrm{A1a}}$ have stronger autocorrelation than that of $r^{\mathrm{A1a}}$, or in other words, the underlying time series of $r^{\mathrm{A1a}}$ are closer to white noise behaviour than $fv^{\mathrm{A1a}}$. At low frequencies the psd of $fv^{\mathrm{A1a}}$ generally exceeds the psd of $r^{\mathrm{A1a}}$, while





above a certain frequency ($\sim 0.5\,\mathrm{yr}^{-1}$ for P1) the psd of $r^{\mathrm{A1}}$ exceeds that of $fv^{\mathrm{A1a}}$. For P1 that means on time scales shorter

than $\sim 2\,\mathrm{yr}$ $r^{\mathrm{A1a}}$ includes more power then $fv^{\mathrm{A1a}}$. At P1, the spectral indices $\kappa$ adjusted to $r^{\mathrm{A1a}}$ and $fv^{\mathrm{A1a}}$ (Section 3.3.3) are

$-1.75$ and $\leq -3$, respectively. For each grid cell, $\kappa$ of $fv^{\mathrm{A1a}}$ is calculated to be $-3$ or more negative. (HECTOR only yields

numerical stable results for $\kappa \geq -3$.) For each grid cell, $\kappa$ of $r^{\mathrm{A1a}}$ are shown in Fig. 11b. The mean value over the entire area

amounts to $-1.72$. It indicates temporally correlated residuals with characteristics close to random-walk noise.

### 4.4 Dominant patterns in altimetric residuals

Fig. 12 and 13 show results of the PCA performed on the altimetric residuals and residual differences, respectively (Sec-

tion 3.3.4). The first three modes explain together $22\,\%$ of the residual variance and $20\,\%$ of the variance of residual differences.

The first mode of the residual differences captures $10\,\%$ and its temporal pattern reveals a prominent drop between July 2010

and January 2011. Due to data standardisation prior to PCA the spatial patterns cannot be directly interpreted as amplitudes in

elevation change of the respective temporal patterns. For this reason, we rescale the spatial patterns by multiplying them with

the std of each time series (which was used beforehand to normalise the time series). Thereby, we regain interpretable magni-

tudes of the spatial patterns. Fig. 12a–f and 13a–f illustrate the version-dependent original and rescaled spatial patterns for $r^{\mathrm{A1a}}$

and $r^{\mathrm{A1a}} - r^{\mathrm{A2a}}$, respectively. (For all versions and both PCA, the original and rescaled patterns are illustrated in Fig. S27–S29).

## 5 Discussion

### 5.1 Interannual firn thickness variations

Adjusted firn thickness variations $fv^{\mathrm{A}}$ (e.g. Fig. 8a for version A1a) and modelled firn thickness variations $fv^{\mathrm{M}}$ (e.g. Fig. 8b for

Ma) share the same general spatial patterns. The largest magnitudes are found at lower elevations near the ice sheet margins

with median rms values in the range of decimetres. The smallest magnitudes are found over the plateau of the EAIS with

median rms values in the range of centimetres (Section 4.2.3). This general spatial pattern was to be expected, as it is related

to the spatial variability of SMB. Snowfall, the main driver of Antarctic SMB variability, increases from the dry, relatively flat

and homogeneous interior to the steep and complex topography of the wetter coast. High snowfall at the ice sheet margins

occurs due to orographic precipitation, influenced by the winds and topography of the AIS (Lenaerts et al., 2019).

    The power spectral density (psd) of $fv^{\mathrm{A}}$ decreases from low to high frequencies with spectral indices $\kappa \leq -3$ for power-

law noise (Section 4.3, Fig. 11a). The strong temporal autocorrelation observed in the interannual firn signals go with the

findings of King and Watson (2020). They estimated the power-law noise parameter $\kappa$ in the range of $-2.3$ to $-2.2$ and $-3.0$

to $-2.6$ based on SMB estimates from RACMO2.3p2 and ice core composites, respectively. (Unlike our analysis, they did not

co-estimate a quadratic or seasonal term.)

    In the following, we compare how much variance of altimetric variations (for the period after 2003) can be explained

according to the applied approach and the two different spatial considerations used previously: First, the percentages assessed





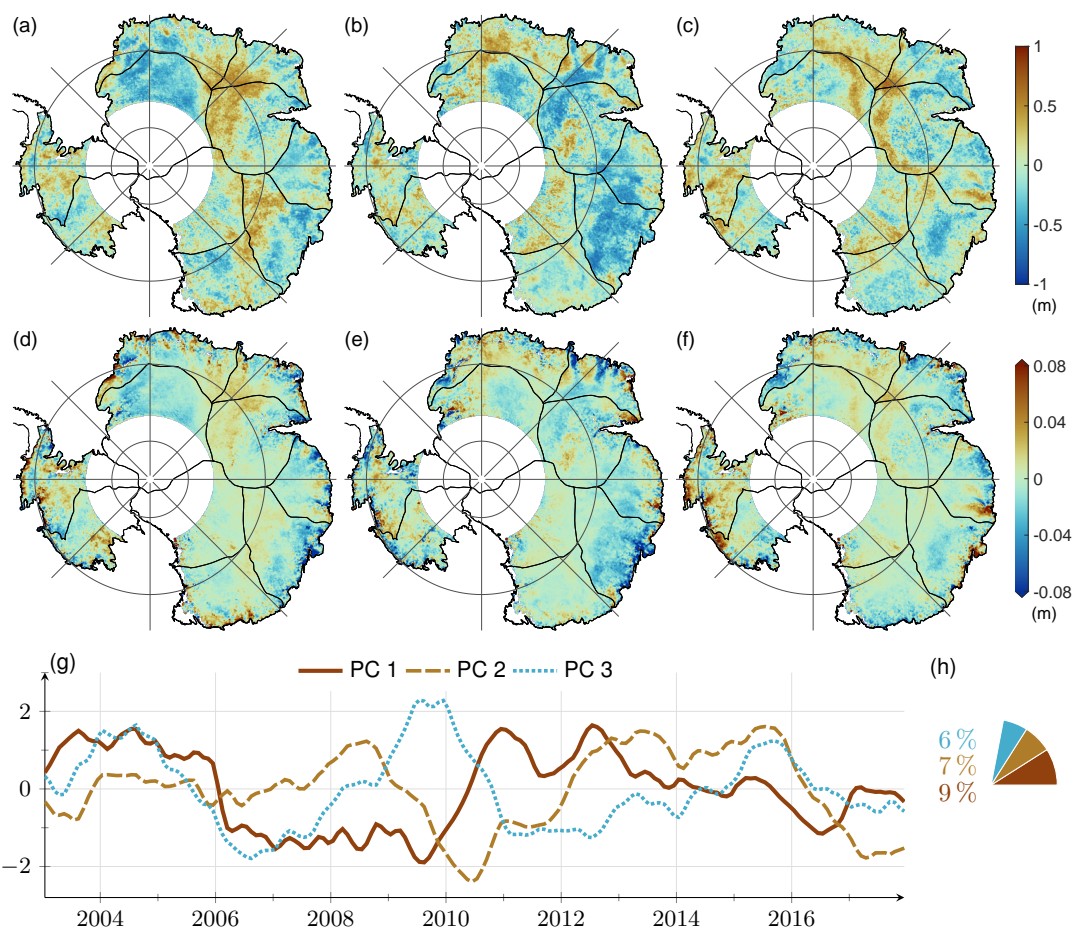

**Figure 12.** PCA results of standardised altimetric residuals for the period after 2003. (a–c) First three spatial patterns (EOFs) – version-dependent, shown here for $r^{\mathrm{A1a}}$. (d–f) Rescaled first three EOFs for $r^{\mathrm{A1a}}$. (g) First three temporal patterns (PCs) determined from the aggregated data sets of $r^{\mathrm{A1a}}$, $r^{\mathrm{A1b}}$, $r^{\mathrm{A2a}}$ and $r^{\mathrm{A2b}}$. (h) Associated percentages of the total residual variance considering the respective PC–EOF pairs. We define the PCs as standardised time series (mean of zero, std of 1) and without a unit while the EOFs have the unit of metre.



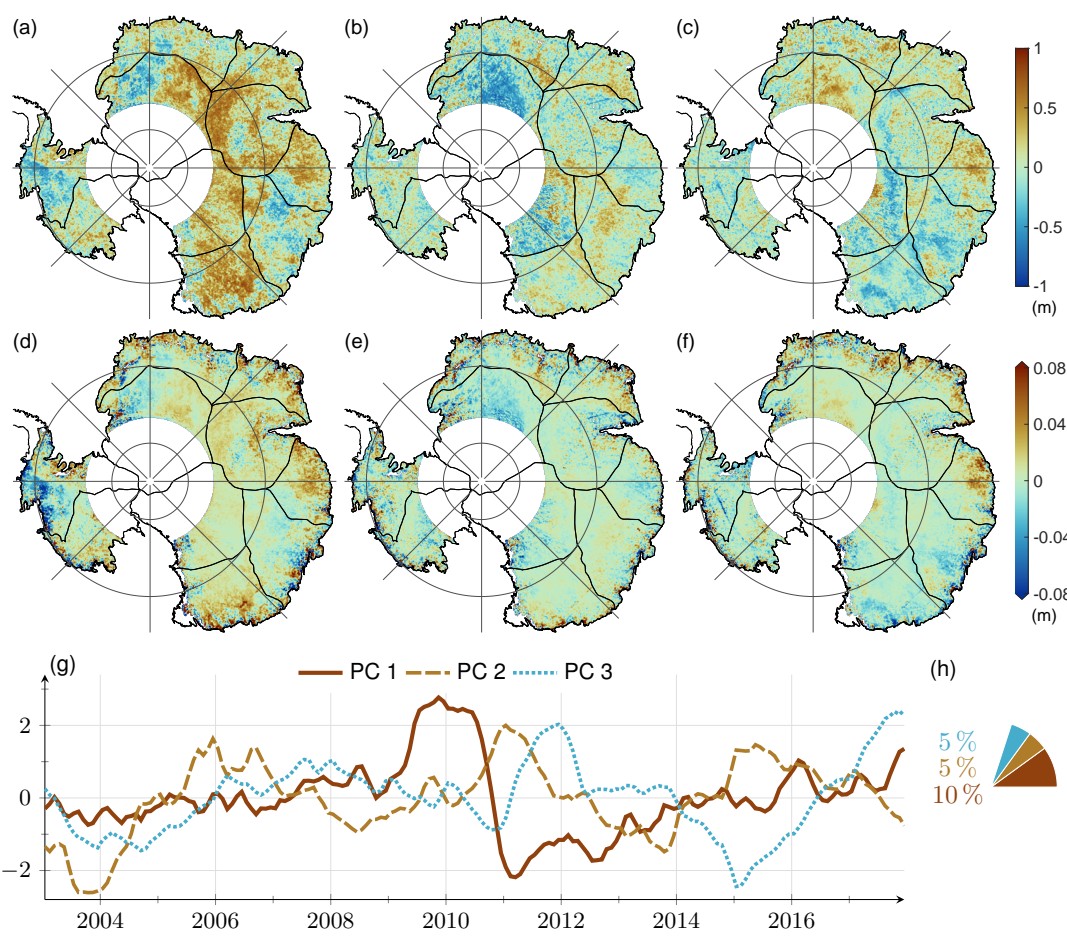

**Figure 13.** PCA results of standardised altimetric residual differences for the period after 2003. (a–c) First three spatial patterns (EOFs) – version-dependent, shown here for $r^{A1a} - r^{A2a}$. (d–f) Rescaled first three EOFs for $r^{A1a} - r^{A2a}$. (g) First three temporal patterns (PCs) – the joint basis of $r^{A1a} - r^{A2a}$ and $r^{A1b} - r^{A2b}$. (h) Associated percentages of the total variance of residual differences considering the respective PC–EOF pairs.



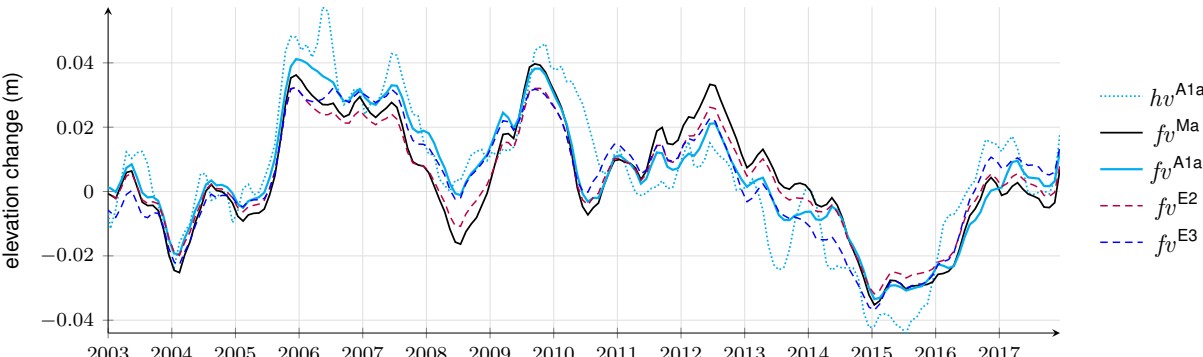

**Figure 14.** Mean Antarctic interannual elevation changes depending on the applied approach. Altimetric variations ($hv^{\text{A1a}}$), modelled firn thickness variations ($fv^{\text{Ma}}$), adjusted firn thickness variations ($fv^{\text{A1a}}$), scaled firn thickness variations ($fv^{\text{E2}}$) and modified adjusted firn thickness variations ($fv^{\text{E3}}$).

from grid cell time series and then averaged over the entire area. Second, the percentages from time series averaged over the
entire area ('mean Antarctic' time series, Fig. 14). The modelled firn thickness variations, $fv^{\text{Ma}}$, explain $11\,\%$ and $63\,\%$ for
the two spatial considerations, respectively (Table A1, columns E1 and $\overline{\text{E1}}$). The scaled firn thickness variations, $fv^{\text{E2}}$, explain
$35\,\%$ and $73\,\%$ (Table A1, columns E2 and $\overline{\text{E2}}$), respectively. The modified adjusted firn thickness variations, $fv^{\text{E3}}$, explain
$42\,\%$ and $82\,\%$ (Table A1, columns E3 and $\overline{\text{E3}}$). Finally, the adjusted firn thickness variations, $fv^{\text{A1a}}$, explain $46\,\%$ and $83\,\%$
for the two spatial considerations (Table 3, columns A1a and $\overline{\text{A1a}}$). Our regression approach (Eq. 1), which generates $fv^{\text{A1a}}$,
explains the greatest part of the variance of altimetric variations compared with the other approaches. This applies not only
for the estimates considering each grid cell equally, but also for the estimates based on time series averaged over basins or the
entire area. Furthermore, the spatial scale investigated is crucial for the results, as the estimates from the basin-mean time series
explain more of the altimetry variance than the estimates considering each grid cell equally. However, the latter are needed to
enable the investigation and further interpretation of regression results based on their spatial patterns.

**5.2   Uncertainty and robustness of adjusted firn thickness variations**

The adjusted firn thickness variations, $fv^{\text{A}}$, include the effects of firn model errors and altimetry errors. The differences $fv^{\text{A1a}} - fv^{\text{A1b}}$ (Fig. 15a) and $fv^{\text{A2a}} - fv^{\text{A2b}}$, evaluated at every grid cell, are used to assess the influence of different firn model setups
on $fv^{\text{A}}$. The median values (over all grid cells) of absolute and relative differences (A1a–A1b) are $\sim 2.6\,\text{cm}$ and $\sim 51\,\%$,
respectively (Table 2, Fig. 9). The differences $fv^{\text{A1a}} - fv^{\text{A2a}}$ (Fig. 15b) and $fv^{\text{A1b}} - fv^{\text{A2b}}$, evaluated at every grid cell, are used
to assess the influence of different altimetry analysis on $fv^{\text{A}}$. The median values (over all grid cells) of absolute and relative
differences (A1a–A2a) are $\sim 2.9\,\text{cm}$ and $\sim 58\,\%$, respectively (Table 2, Fig. 9). Both the firn model and altimetry errors are
discussed in Sections 5.3 and 5.4 separately.

To assess the combined influence of firn model and altimetry errors on $fv^{\text{A}}$, the maximum deviation within the different
versions of $fv^{\text{A}}$ is used (Section 3.3.1). Fig. 15c shows the map of the maximum rms values. The median values (over all grid



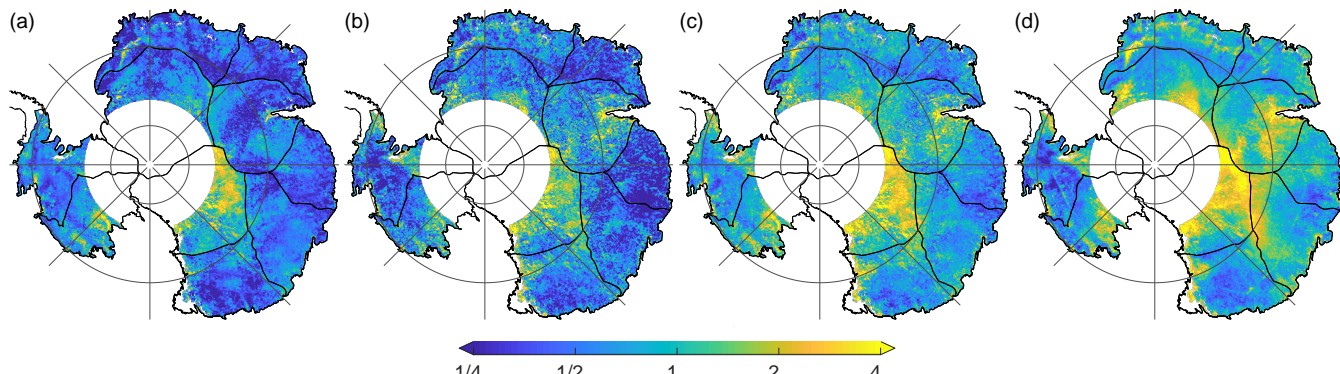

**Figure 15.** (a) Rms of (the time series of) the differences $fv^{\mathrm{A1a}} - fv^{\mathrm{A1b}}$. (b) Rms of the differences $fv^{\mathrm{A1a}} - fv^{\mathrm{A2a}}$. (c) Uncertainty estimate of $fv^{\mathrm{A}}$: Maximum rms of any combination of differences within versions of $fv^{\mathrm{A}}$. (d) Rms of the residual differences $r^{\mathrm{A1a}} - r^{\mathrm{A2a}}$ considering only the period after 2003. All rms maps (a–d) are normalised by the rms of $fv^{\mathrm{Ma}}$.

cells) of absolute and relative (maximum) differences are ∼4.3 cm and ∼82 %, respectively. In addition, median values are calculated for every basin, i.e. over all grid cells within the respective basins. The absolute and relative uncertainties range from 2.3 cm (basin 8) to 10.9 cm (basin 10) and from 59 % (basin 5) to 189 % (basin 8), respectively. We consider these estimates to be rough, but rather conservative uncertainty assessments for the adjusted firn thickness variations. In addition to the evaluation at grid cell level, the uncertainty of $fv^{\mathrm{A}}$ is assessed by time series differences of the basin means. (See Fig. S23

for the basin-mean time series of the four versions of $fv^{\mathrm{A}}$). The associated uncertainties per basin range from 1.0 cm (basin 4) to 6.7 cm (basin 10). The relative uncertainties are in the range of 21 % (basin 2) to 111 % (basin 8). For mean Antarctic $fv^{\mathrm{A}}$ an absolute and relative uncertainty of ∼1.4 cm and ∼71 %, respectively, are estimated.

To assess the robustness of $fv^{\mathrm{A}}$, statistical tests were carried out (Section 3.3.2). In particular, four tests per basin, each comparing the temporal rms of the following pair of differences in firn thickness variations are conducted: Test (1) compares

A1a−A2a to A1a−Ma, test (2) compares A1a−A2a to A2a−Ma, test (3) compares A1b−A2b to A1b−Mb and test (4) compares A1b−A2b to A2b−Mb. For all 40 tests, H0 is rejected (at the 5 % significance level) and thus, H1 is accepted. This means that the differences within $fv^{\mathrm{A}}$ are significantly smaller than the differences between $fv^{\mathrm{A}}$ and $fv^{\mathrm{M}}$. Fig. 16a exemplifies the distributions of the differences for basin 3. (The histograms and cumulative histograms for all basins are shown in Fig. S30 and S31, respectively.) The results of the statistical tests demonstrate that $fv^{\mathrm{A}}$ is relatively robust to the choice of data sets, firn

models and altimetry products. The choice of data sets does not significantly influence $fv^{\mathrm{A}}$. Consequently, the assumption that $fv^{\mathrm{A}}$ represents a significant improvement over the modelled variations is reasonable. Limitations are discussed below.

## 5.3 Firn model errors

Firn model errors arise due to firn signals not (correctly) represented by either the firn model or its input from RCMs or even reanalysis data. Differences between $fv^{\mathrm{Ma}}$ and $fv^{\mathrm{Mb}}$ (Fig. S19) as well as differences between any version of $fv^{\mathrm{A}}$ and $fv^{\mathrm{M}}$

(Fig. S18) reflect firn model uncertainties and errors. Partly, $fv^{\mathrm{A}} - fv^{\mathrm{M}}$ also include errors related to the altimetry measurements





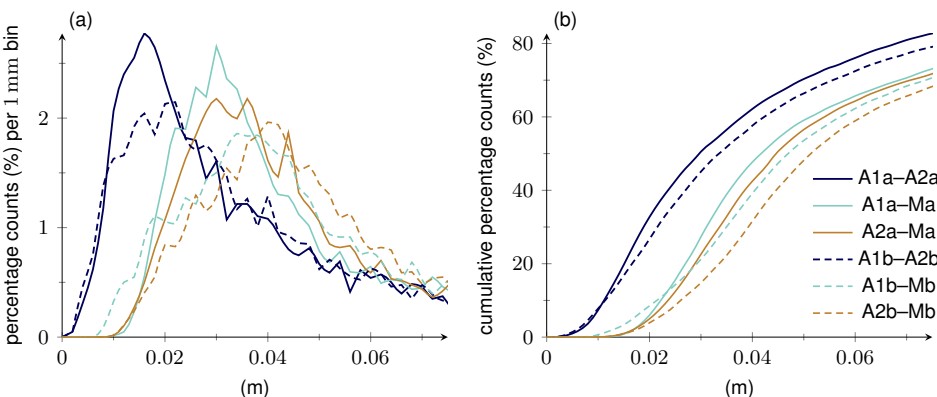

**Figure 16.** Histograms of the temporal rms of differences between various versions of firn thickness variations assessed at each grid cell of basin 3. (a) Histograms. (b) Cumulative histograms.

and analysis, as discussed in Section 5.4. (See also Table 2 for an overview of the various differences in firn thickness variations $fv$ and their description.) Firn models generally show a smoother spatial pattern than altimetry. This can be seen in the different $fv$ (Fig. 8b versus 8a) and also in the different scaling factors $e$ (Fig. 7d–f versus 7a–b). One reason for this may be the lack of small-scale, mainly wind-driven processes not resolved in the firn modelling outputs (Lenaerts et al., 2012, 2019), leading to a
blurred spatial distribution of modelled firn thickness variations.

The spatial patterns of absolute differences within $fv^{\mathrm{M}}$ and between any version of $fv^{\mathrm{A}}$ and $fv^{\mathrm{M}}$ ('the adjustments', e.g. Fig. 8c), follow the spatial pattern of the signal itself. The greatest differences occur at the margins, where the climate is wetter and temperatures and accumulation are higher than inland. Especially in these coastal regions of high-relief topography, the horizontal resolution of the models, probably together with its physics, play an important role (Mottram et al., 2021). There,
the differences between altimetry and firn models may be influenced by an incorrect or inaccurate spatial distribution of the modelled firn thickness variations (Fig. 7). The modelled SMB components and their uncertainties have a direct impact on the modelled firn thickness. By assessing the spread of an ensemble of modelled firn thickness changes, Verjans et al. (2021) identified the RCMs as the largest contributor to the ensemble uncertainty. A precise parameterisation of firn compaction and surface snow density gains in importance in regions with high snowfall and large spatial variability of climatic conditions, such
as Dronning Maud Land and Enderby Land (Verjans et al., 2021). However, the firn compaction rate in the IMAU and GSFC firn model is determined by constant mean annual accumulation and not by instantaneous overburden pressure. This lessens the actual firn compaction variability potentially across all the areas of large accumulation variability (Kuipers Munneke et al., 2015).

In a relative sense, the adjustments over the firn models (that is, any version of $fv^{\mathrm{A}} - fv^{\mathrm{M}}$, e.g. Fig. 8d) generally increase
from the coast to the EAIS interior as the magnitude of the signal, the firn thickness variation, is very small in the interior due to the cold and dry climate. In these areas of low snowfall, the relative uncertainties in the firn models are virtually unaffected by the formulation of firn densification and surface snow density, but the input of RCM components is essential (Verjans





et al., 2021). Scambos et al. (2012) argue that RCMs might overestimate SMB in wind-glazed areas. These areas feature wind-polished glazed surfaces at the top of a coarsely recrystallised firn layer and are formed by constant katabatic winds. They have

near-zero SMB and occur on leeward faces of ice-sheet undulations and megadunes (Scambos et al., 2012). Large wind glazed areas are located across basin 4 and 8, where all four versions of adjustments reveal highest relative values (Fig. S18e–h).

In basin 4, towards the border to basins 1 and 3, the large relative adjustments (Fig. S18e–h) indicate disagreement between the models and altimetry. Neither the uncertainties due to different altimetry analysis nor the influence of the different firn model setups have a strong impact on $fv^A$ in this region (Fig. S17i–l). Furthermore, within $fv^M$ there are no large differences in

this region of basin 4 (Fig. S19d). The two models agree and the four versions of altimetry agree, but the models and altimetry do not agree. The reasons why discrepancies occur particularly in this region are not yet clear. Basin 8 is characterised by large megadune fields (Fahnestock et al., 2000; Dadic et al., 2013). Megadune fields cover more than $500{,}000\,\mathrm{km}^2$ of the East AIS plateau. The megadunes typically have an amplitude of 2 to $4\,\mathrm{m}$ and wavelengths of 2 to $5\,\mathrm{km}$ and are formed by a complex interaction of surface topography, snow accumulation and redistribution due to highly persistent katabatic winds.

While leeward slopes are wind glazed, windward slopes accumulate and are characterised by streamlined bumps or grooves (sastrugi) up to $1.5\,\mathrm{m}$ in height (Fahnestock et al., 2000; Frezzotti et al., 2002). The discrepancy between altimetry and the firn models across basin 8 can partly be explained by the lacking modelling of the formation of the complex spatial pattern of megadunes and their migration over time in the firn models. For basin 8, not only do the models and altimetry not match, but the relative differences between $fv^{Ma}$ and $fv^{Mb}$ (Fig. S19d) and between the different versions of $fv^A$ (discussed in Section 5.4)

are also high.

Discrepancies within the four versions of adjustments can further indicate which firn model (or which dominant patterns of one firn model) fits the altimetry better. Overall, the adjustments are smaller when involving Ma, the IMAU firn model (Fig. 9, Table 2). Amongst the different basins, this applies in particular for basins 4–6 (Fig. S30d-f and 31Sd-f). Across basin 1 the adjustments tend to be slightly smaller when involving Mb, the GSFC firn model (Fig. S30a and S31a).

Altimetric residuals, $r^A$, still include a non-negligible part ($53\,\%$ for A1a) of the variance of altimetric variations (Fig. 10c, Table 3). It is likely that $r^A$ still include real firn signals not captured by the dominant temporal patterns of the firn models. The psd of the underlying time series of $r^{A1a}$ yield a spectral index of $-1.7$ (Section 4.3, Fig. 11b). The remaining autocorrelation in the residuals suggests that temporally correlated signals such as real firn signals are still present. Also, the spatial patterns of the most dominant modes of $r^A$ reveal topography-dependent magnitudes and patterns, as one would expect from SMB and

its variations (Section 4.4, Fig. 12d–f). Besides other firn signals, the altimetric residuals additionally include altimetry errors (discussed in Section 5.4) and probably also further signals related to variations in ice flow dynamics or subglacial hydrology (not further discussed).

## 5.4 Altimetry errors

The differences between any version of $fv^A$ and $fv^M$ ('the adjustments', e.g. Fig. 8c) may include effects of altimetry errors,

in addition to firn model errors. Measurement noise in altimetry might explain another part of the fact that firn models show a smoother spatial pattern of variations than altimetry. Noise in altimetry can be a problem, especially in the interior of the





EAIS where the signal-to-noise ratio is low (Section 5.5). Over megadune areas (widely located in the interior across basin 8), conventional radar altimetry with pulse-limited footprints of $1.5$ to $2.5\,\mathrm{km}$ in diameter may not be capable of adequately observing the time-varying spatial patterns of megadunes.

A further limitation in radar altimetry is that measurements refer to the local topographic maxima within their footprints. Especially at the margins over complex topography, this can lead to sampling issues, as the elevation changes acquired there cannot capture the larger changes often found in the valleys. Laser altimeters are not affected by this since their footprints are much smaller (in the range of decimetres). However, ICESat had to be operated in campaign mode (Abshire et al., 2005). Thus, the sampling in areas with steep slopes can vary strongly during the period 2003–2009 as some of the months rely only on radar

altimetry measurements while other months include measurements from radar and laser altimetry. Moreover, radar altimetry results are affected by the time-varying radar waveform shape due to time-varying signal penetration (Davis and Ferguson, 2004; Rémy et al., 2012). Even though errors related to these effects are accounted for in the altimetry processing, they are not fully eliminated and may have an impact on the adjustments. In addition, these time-variable errors are also likely to be included in the altimetric residuals, $r^{\mathrm{A}}$, because $r^{\mathrm{A}}$ are temporally correlated just as the errors (Section 4.3, Fig. 11b).

Discrepancies within the adjustments (any version of $fv^{\mathrm{A}} - fv^{\mathrm{M}}$) can indicate which altimetry solution is closer to the firn models. Overall, the adjustments are smaller when involving A1, TUD altimetry (Fig. 9, Table 2). Amongst the different basins this applies in particular for basins 1, 5 and 6 (Fig. S30b, e, f and S31b, e, f). Across basin 8 the adjustments tend to be smaller when involving A2, JPL altimetry (Fig. S30h and S31h).

The differences in $fv^{\mathrm{A}}$ and the altimetric residuals, $r^{\mathrm{A}}$, between solutions based on the same firn model (A1a−A2a or

A1b−A2b) are displayed in Fig. 15b and d, respectively. They mirror the altimetry uncertainties due to a different analysis of the altimetry measurements. The median values (over all grid cells) of the absolute and relative residual differences $r^{\mathrm{A1}} - r^{\mathrm{A2}}$ are $\sim 4.7\,\mathrm{cm}$ and $\sim 96\,\%$, respectively. The residual differences are evaluated for every grid cell and only the time period after 2003 is considered. If the entire period was considered, the median values would increase considerably ($\sim 7.2\,\mathrm{cm}$ and $\sim 162\,\%$). For both periods, the residual differences are greater than the differences $fv^{\mathrm{A1}} - fv^{\mathrm{A2}}$ (Table 2, Fig. 15b) and also greater than

the uncertainty estimate of $fv^{\mathrm{A}}$ (Section 5.2, Fig. 15c). Thus, the altimetry uncertainties in the residuals are greater than the combined uncertainties of firn modelling and altimetry affecting the adjusted firn thickness variations.

The differences between $fv^{\mathrm{A1}}$ and $fv^{\mathrm{A2}}$ as well as between $r^{\mathrm{A1}}$ and $r^{\mathrm{A2}}$ mostly result from the combined effect of the various differences between the altimetry analysis of TUD and JPL (Section 2.1). The rms of $fv^{\mathrm{A1a}} - fv^{\mathrm{A2a}}$ is shown in Fig. 15b in a relative sense. The largest relative differences occur in regions of complex topography, such as in Victoria Land (at the margin

of basin 7) and next to the Amery Ice Shelf (at the margin of basin 4) and over almost the entire basin 8, for which we already discussed the influence of megadunes. In addition, stripes related to the satellite ground tracks are visible in the region of basins 1 to 2 (Fig. 15b). They seem to appear predominantly in $fv^{\mathrm{A2}}$ (Fig. S17b and d).

The following features may likely be quite clearly attributed to a difference in intermission/intermode calibration between TUD and JPL altimetry. The mode change of CryoSat-2 (LRM/SARIn mode; see e.g. Fig. 5 in Slater et al. (2018) for the

mode boundaries) is reflected in the difference of the residuals (Fig. 15d). Here, the main influence seems to come from JPL altimetry, as the areas at the mode boundary in basins 5–7 and 9–10, characterised by a higher rms value, are mainly



visible in $r^{A2}$ (Fig. S20f and h). In addition, the mode transition also appears to be reflected in $fv^{A2}$ particularly at basins 5 and 6 (Fig. S17b and d). The PCA carried out on $r^{A1a} - r^{A2a}$ and $r^{A1b} - r^{A2b}$ reveal a prominent drop between July 2010 and January 2011 together with overall linear trends before and after this drop in the first PC (Fig. 13g). The corresponding

spatial pattern (Fig. 13a) is most pronounced and coherent over the EAIS. The pattern of the first mode is an indicator for uncertainties/differences in deriving intermission offsets, as CryoSat-2 measurements begin in July 2010. The errors in the altimetry are not only seen in the first modes of the PCA of the residual differences. It is likely that the first modes of the PCA of the residuals themselves are also affected by altimetry errors. A comparison of the dominant modes of the residuals (Fig. 12) with those of the residual differences (Fig. 13) indicates partly similar features, which suggests similar causes. For example,

there are also remarkably large fluctuations in the first temporal patterns of the residuals between July 2009 and January 2011 (Fig. 12g).

### 5.5 Limitations of the approach

In regions of low signal-to-noise ratio the regression approach has a limited capability to distinguish between signal and error. This applies in particular for the interior of the EAIS (basin 8 and parts of basin 1 and 4). In these areas, the regression of

the altimetry data to $PC^M$ (the dominant temporal patterns in modelled firn thickness variations) may be dominated by noise in the altimetry data. In this study, we work with a constant spatial grid resolution of $10\,\mathrm{km}$ x $10\,\mathrm{km}$ regardless of the signal magnitude in each grid cell. To improve the signal-to-noise ratio, further work may geographically vary and adjust the spatial resolution to the spatial variability of the glaciological processes, that is in general a higher resolution on the coast and a coarser resolution in the interior.

We included altimetry measurements only over the period May 1992 to December 2017 as this represents the common period of TUD and JPL altimetry (Section 2). JPL altimetry data, however, are available until December 2020. Further investigations may include the JPL data after December 2017. These may incorporate accurate laser measurements from ICESat-2 characterised by low noise level and near-zero signal penetration (Nilsson et al., 2022; Otosaka et al., 2023a).

The stochastic model in the regression approach does not include co-variances in altimetry (Section 3.1), although errors in

the altimetry time series exhibit temporal correlations, as shown by Ferguson et al. (2004) and also in this study (Section 4.3). The consideration of temporal correlations is essential for a proper, realistic uncertainty estimation of long-term trends in particular (Williams et al., 2014). Thus, for inferring potentially statistically significant long-term signals in satellite altimetry future work may extend the stochastic model. This requires a comprehensive error characterisation for altimetry products, which is not given up to now. Nevertheless, different noise models (e.g., power-law, Generalized Gauss Markov, auto-regressive) could be

considered to empirically identify and apply the best fitting noise model to the regression approach (Bos et al., 2012; King and Watson, 2020). Another possibility for characterising errors could be the consideration of an ensemble of altimetry solutions and their spread as demonstrated by Willen et al. (2022).

Our study does not include independent observations to validate the benefits of $fv^A$. Most of the ground-based SMB observations are single point measurements and have a very sparse spatial and temporal coverage (Eisen et al., 2008). Thus, a

validation of $fv^A$ could only be performed for selected, distinctly local regions and/or certain time intervals. A conceivable



comparison could make use of stakes observations, as in the studies of Mottram et al. (2021) across Antarctica and Richter et al. (2021) in the Lake Vostok region.

## 5.6 Outlook

To improve firn model outputs, we underline the importance of refining the horizontal spatial resolution of RCMs to simulate
surface processes at a higher spatial distribution (Lenaerts et al., 2019). For Greenland, Noël et al. (2016) statistically down-scaled outputs from RACMO2.3 at 5.5 and 11 km to a high-resolution product of 1 km, leading e.g. to increased melt over certain areas. Similar work is in progress for Antarctica, downscaling RACMO2.3p2 at 27 km to 2 km (Noël et al., 2023). Furthermore, a more detailed physical parameterisation of the processes already considered and the inclusion of processes not yet simulated can improve the models (Agosta et al., 2019; Gutiérrez et al., 2021). An update of RACMO2.3p2 to RACMO2.4 with
enhanced physics may soon be available. This includes several new and updated parameterisations, such as a cloud, aerosol and radiation scheme or a new spectral albedo and radiative transfer scheme in snow scheme (van Dalum and van de Berg, 2023).

To improve altimetry products, measurement noise and correlated altimetry errors related in particular to time-variable signal penetration and scattering effects could be reduced by improving the methods of analysis. Helm et al. (2023) developed
a new retracker based on a deep convolutional neural network architecture, resulting in strongly reduced time-variable signal penetration. The new retracker could significantly improve the accuracy of elevation change products from the entire sequence of radar altimetry missions. Furthermore, improving the methods for intermission calibration would reduce uncertainties in altimetry estimates at various time scales. The patterns of estimated intermission offsets are spatially variant and related to the waveform parameters (topography and surface properties play a role here). However, this relation is not fully understood, so
that no functional relationship has yet been found and intermission offsets are determined empirically (Zwally et al., 2005; Khvorostovsky, 2012; Schröder et al., 2019a; Nilsson et al., 2022). Therefore, intermission calibration still remains one of the most challenging processing steps for inferring a long-term, multi-mission satellite altimetry estimate.

Future developments in firn modelling, satellite altimetry analysis and altimetry mission sensors will allow interannual firn signals to be identified and quantified with higher accuracy. This will further impact long-term estimates and reduce their
uncertainties. The regression approach presented in this study may set the stage for isolating long-term signals in satellite altimetry from the large interannual variations. For this reason, future studies should extend the approach with an appropriate stochastic model that accounts for covariances in altimetry to derive statistically significant long-term trends over 25 to 30 years. Longer (altimetry) time series will then further reduce trend uncertainties (Wouters et al., 2013). In this way, large uncertainties in inferring mass balance estimates of the EAIS (Otosaka et al., 2023b) may be reduced and the question whether
the EAIS is currently thickening or thinning (Nilsson et al., 2021) may be answered in the future.





## 6 Conclusions

We deliberately targeted spatially resolved variations in Antarctic firn thickness. For this purpose, we developed and presented a new approach for combining satellite altimetry and firn modelling estimates at a high temporal (monthly) and spatial (grid scale of $10\,\mathrm{km}$) resolution. On the one hand, our approach incorporates the strengths of the firn model, above all the capability to

capture the timing of firn thickness variations. On the other hand, our approach compensates for shortcomings of the firn model, foremost the accurate simulation of the location-dependent amplitudes of the variations. To do so, we fitted dominant temporal patterns of interannual to decadal variations in Antarctic firn thickness inferred from the firn models IMAU (Veldhuijsen et al., 2023) and GSFC (Medley et al., 2022a) to satellite altimetry observations from TUD (Schröder et al., 2019a) and JPL (Nilsson et al., 2022). In this way, we generated a new, combined data set, which we named the adjusted firn thickness variations, $fv^{\mathrm{A}}$.

Our guiding question was: How well can satellite altimetry and firn models resolve Antarctic firn thickness variations? This study shows that firn models and altimetry products provide complementary information on firn thickness variations. The combined data set, $fv^{\mathrm{A}}$, characterises spatially resolved variations better than either (1) firn models alone or (2) altimetry alone. (1) The $fv^{\mathrm{A}}$ outperform the modelled firn thickness variations, $fv^{\mathrm{M}}$, because $fv^{\mathrm{A}}$ improves the amplitudes of the variations compared with $fv^{\mathrm{M}}$. The amplitudes represent an improvement because they are observed by the altimeter satellites and their

patterns actually indicate more spatial and thereby meaningful information. However, one caveat should be noted. The improved observed amplitudes may also include effects of altimetry errors due to firn penetration. This is because the temporal variations of these errors correlate with the temporal variations of the signal, as both the time-variable signal and the errors are influenced by the SMB and firn processes. (2) The $fv^{\mathrm{A}}$ outperform the altimetric variations, $hv^{\mathrm{A}}$, because $fv^{\mathrm{A}}$ eliminates a large part of the altimetry errors. If one were to take $hv^{\mathrm{A}}$ alone, this would also incorporate all the errors of $hv^{\mathrm{A}}$. Over Antarc-

tica, or rather the entire area studied, this would introduce median absolute and relative uncertainties of $\sim\!7.2\,\mathrm{cm}$ and $\sim\!162\,\%$, respectively (evaluated on grid cell level). However, one caveat should be noted. By choosing $fv^{\mathrm{A}}$ instead of $hv^{\mathrm{A}}$, part of the observed firn signal is ignored.

How well the $fv^{\mathrm{A}}$ resolve real Antarctic firn thickness variations depends strongly on the region under investigation. Over Antarctica, median absolute and relative uncertainties of $fv^{\mathrm{A}}$ are $\sim\!4.3\,\mathrm{cm}$ and $\sim\!82\,\%$, respectively (evaluated on grid cell

level). Over the basin areas, the median relative uncertainties range from $59\,\%$ (basin 5) to $189\,\%$ (basin 8). Across basin 8, we also spatially resolved disagreements between $fv^{\mathrm{A}}$ and $fv^{\mathrm{M}}$. The large uncertainty and the disagreement are due to the presence of megadune fields. Overall, the differences between $fv^{\mathrm{A}}$ and $fv^{\mathrm{M}}$ are smallest when using the TUD altimetry and the IMAU firn model. Amongst the different basins, this is especially true for basins 5 and 6. From the spectral analysis of the altimetry residuals, $r^{\mathrm{A}}$, we find still autocorrelated signals that we could not attribute to firn thickness variations using the firn models.

We attribute this to a combination of altimetry errors (time-variable signal penetration, errors in intermission offsets) and firn model errors (incorrectly simulated/missing processes in the firn models).





## Appendix A: Impact of methodological changes

### A1 Methods

To investigate the impact of methodological changes on determining adjusted firn thickness variations, $fv^{\mathrm{A}}$, three modifications
of the original regression approach are tested.

In the first experiment E1, we simply subtract the altimetric variations, $hv^{\mathrm{E1}}$, from the modelled firn thickness variations,
$fv^{\mathrm{M}}$, according to

$$r^{\mathrm{E1}}(t) = hv^{\mathrm{E1}}(t) - fv^{\mathrm{M}}(t). \tag{A1}$$

$fv^{\mathrm{M}}$ is derived by least squares fit according to Eq. 4. $hv^{\mathrm{E1}}$ is derived by least squares fit according to

$$
\begin{aligned}
h(t)^{\mathrm{A}} = {} & a + b\,t + c\,(0.5\,t^2) \\
& + \mathrm{H}_1(t)\left[d_1\cos(\omega t) + d_2\sin(\omega t) + d_3\cos(2\omega t) + d_4\sin(2\omega t)\right] \\
& + \mathrm{H}_2(t)\left[d_5\cos(\omega t) + d_6\sin(\omega t) + d_7\cos(2\omega t) + d_8\sin(2\omega t)\right] \\
& + hv^{\mathrm{E1}}(t),
\end{aligned}
\tag{A2}
$$

with the parameters $a, b, c, d_{1,\dots,8}$ and the masks $\mathrm{H}_1, \mathrm{H}_2$ as in Eq. 1.

In the second experiment E2, $fv^{\mathrm{M}}$ at any grid cell is simply scaled to fit the altimetric variations. The regression reads

$$
\begin{aligned}
h(t)^{\mathrm{A}} = {} & a + b\,t + c\,(0.5\,t^2) \\
& + \mathrm{H}_1(t)\left[d_1\cos(\omega t) + d_2\sin(\omega t) + d_3\cos(2\omega t) + d_4\sin(2\omega t)\right] \\
& + \mathrm{H}_2(t)\left[d_5\cos(\omega t) + d_6\sin(\omega t) + d_7\cos(2\omega t) + d_8\sin(2\omega t)\right] \\
& + e\,fv^{\mathrm{M}}(t) \\
& + r^{\mathrm{E2}}(t),
\end{aligned}
\tag{A3}
$$

where $e$ is the scaling factor. We refer to $e\,fv^{\mathrm{M}} = fv^{\mathrm{E2}}$ as scaled firn thickness variations.

In the third experiment E3, we do not change the principle of the deterministic model Eq. 1 but we modify the dominant
temporal patterns $PC^{\mathrm{M}}$. Originally, $PC^{\mathrm{M}}$ are derived from standardised $fv^{\mathrm{M}}$ by PCA. In E3, $fv^{\mathrm{M}}$ are not standardised prior to
the PCA. The resulting modified adjusted firn thickness variations are referred to by $fv^{\mathrm{E3}}$. See also Table B1 for an overview
of the defined symbols and their terminology.





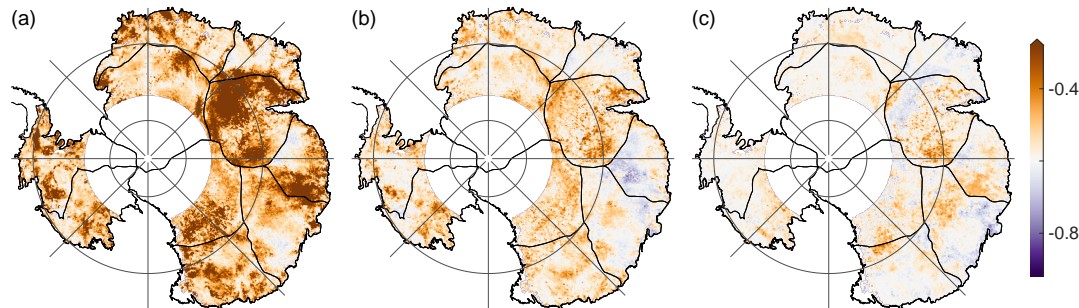

**Figure A1.** Differences between the coefficients of determination from A1a and the experiments E1, E2 and E3. (a) A1a−E1, (b) A1a−E2 and (c) A1a−E3.

We consider that regression method as best whose coefficient of determination, $R^2$, is maximum, i.e. which is able to describe most of the data variance. For the three experiments, the general from of Eq. 5 specifies to

$$R^2_{\mathrm{E1}} = 1 - \frac{SS(r^{\mathrm{E1}})}{SS(fv^{\mathrm{Ma}} + r^{\mathrm{E1}})} = 1 - \frac{SS(r^{\mathrm{E1}})}{SS(hv^{\mathrm{E1}})} \approx 1 - \frac{SS(r^{\mathrm{E1}})}{SS(hv^{\mathrm{A}})}, \tag{A4a}$$

$$R^2_{\mathrm{E2}} = 1 - \frac{SS(r^{\mathrm{E2}})}{SS(efv^{\mathrm{Ma}} + r^{\mathrm{E2}})} = 1 - \frac{SS(r^{\mathrm{E2}})}{SS(fv^{\mathrm{E2}} + r^{\mathrm{E2}})} \approx 1 - \frac{SS(r^{\mathrm{E2}})}{SS(hv^{\mathrm{E1}})} \approx 1 - \frac{SS(r^{\mathrm{E2}})}{SS(hv^{\mathrm{A}})}, \tag{A4b}$$

$$R^2_{\mathrm{E3}} = 1 - \frac{SS(r^{\mathrm{E3}})}{SS(fv^{\mathrm{E3}} + r^{\mathrm{E3}})} \approx 1 - \frac{SS(r^{\mathrm{E3}})}{SS(hv^{\mathrm{E1}})} \approx 1 - \frac{SS(r^{\mathrm{E3}})}{SS(hv^{\mathrm{A}})}. \tag{A4c}$$

Assuming that the changes in the adjusted parameters $b, c, d_{1,\ldots,8}$ due to different versions of regression are negligible, the following approximations are reasonable: $(efv^{\mathrm{Ma}} + r^{\mathrm{E2}}) \approx (fv^{\mathrm{E3}} + r^{\mathrm{E3}}) \approx hv^{\mathrm{E1}} \approx hv^{\mathrm{A}}$.

## A2 Results

The impact of methodological choices on the goodness of fit is tested based on the three modifications/experiments E1–E3 (Section A1). The results are given for using the IMAU firn model and TUD altimetry and should, therefore, be compared to the results from the regression approach A1a.

For every grid cell, Fig. A1 compares the coefficients of determination from the regression approach A1a, $R^2_{\mathrm{A1a}}$, to the coefficients of determination $R^2_{\mathrm{E1}}$, $R^2_{\mathrm{E2}}$ and $R^2_{\mathrm{E3}}$. $R^2_{\mathrm{A1a}}$ is larger than $R^2_{\mathrm{E1}}$, $R^2_{\mathrm{E2}}$ and $R^2_{\mathrm{E3}}$ over 96, 81 and 69 % of the total area, respectively. After calculating $R^2_{\mathrm{E1}}$, $R^2_{\mathrm{E2}}$ and $R^2_{\mathrm{E3}}$ for each grid cell, (basin) mean values are derived and listed by Table A1, columns 2–4. Averaged over the entire area, E1, E2 and E3 have mean $R^2$ values of 0.09, 0.35 and 0.43. For all three modifications, $R^2$ is smaller than $R^2_{\mathrm{A1a}}$ (Table 3, column A1a) and thus, their regression approaches describe less of the data variance than the original regression approach A1a. E2 and E3 describe slightly more of the data variance than A1a for one out of 10 basins (E1, basin 5: 44 versus 43 %; E2, basin 3: 49 versus 47 %). Moreover, Table A1 (columns 6–7) lists values of $R^2$ derived from basin averages time series ($\overline{\mathrm{E1}}$, $\overline{\mathrm{E2}}$ and $\overline{\mathrm{E3}}$). Values derived from basin averages time series are larger than values based on the calculations per grid cell, similar to the regression approach A1a (Table 3, column $\overline{\mathrm{A1a}}$ versus A1a).

The simple scaling factor $e$ adjusted during the regression approach after experiment E2 is displayed in Fig. S32.





**Table A1.** Explained variance or coefficients of determination, $R^2$, for each basin and each experiment E1, E2, E3 of methodological changes to the regression approach over the period after 2003. $R^2$ is first calculated for each grid cell according to Eq. A4a–A4c and after averaged over each basin. Values of $\overline{E1}$, $\overline{E2}$ and $\overline{E3}$ are calculated by first averaging the results from the experiments over each basin and then applying Eq. A4a–A4c.

| Basin | E1 | E2 | E3 | $\overline{E1}$ | $\overline{E2}$ | $\overline{E3}$ |
|---|---|---|---|---|---|---|
| 01 | 0.20 | 0.35 | 0.43 | 0.71 | 0.75 | 0.79 |
| 02 | 0.21 | 0.41 | 0.48 | 0.76 | 0.92 | 0.91 |
| 03 | 0.21 | 0.44 | 0.50 | 0.88 | 0.94 | 0.95 |
| 04 | -0.29 | 0.15 | 0.27 | -5.63 | -0.08 | 0.10 |
| 05 | 0.02 | 0.42 | 0.34 | -0.50 | 0.66 | 0.47 |
| 06 | 0.20 | 0.38 | 0.40 | 0.70 | 0.86 | 0.90 |
| 07 | 0.22 | 0.44 | 0.54 | 0.68 | 0.91 | 0.94 |
| 08 | -0.08 | 0.13 | 0.23 | 0.19 | 0.54 | 0.56 |
| 09 | 0.32 | 0.42 | 0.50 | 0.94 | 0.94 | 0.97 |
| 10 | 0.27 | 0.49 | 0.59 | 0.92 | 0.98 | 0.97 |
| 01–10* | 0.11 | 0.35 | 0.42 | 0.63 | 0.73 | 0.82 |

* refers to the entire area (considered as a single basin)



## Appendix B: List of symbols

**Table B1.** List of symbols and their terminology (columns 1–2). Sections and equations where the symbols are explained and defined (column 3). Different versions (see Table 1) of the respective symbols (column 4).

| Symbol | Terminology | References | Versions |
|---|---|---|---|
| $h^{\mathrm{A}}$ | Altimetric elevation changes* | Section 3.1, Eq. 1 | A1, A2 |
| $hv^{\mathrm{A}}$ | Altimetric variations | Section 3.1.2, Eq. 2 | A1a, A2a, A1b, A2b |
| $f^{\mathrm{M}}$ | Modelled firn thickness changes* | Section 3.1.1, Eq. 4 | Ma, Mb |
| $fv^{\mathrm{M}}$ | Modelled firn thickness variations | Section 3.1.1, Eq. 4 | Ma, Mb |
| $PC^{\mathrm{M}}_{1\ldots N}$ | $N$ dominant temporal patterns in modelled firn thickness variations | Section 3.1.1, Eq. 1, 3 | Ma, Mb |
| $e^{\mathrm{A}}_{1\ldots N}$ | $N$ observed scaling factors | Section 3.1, Eq. 1, 3 | A1a, A2a, A1b, A2b |
| $e^{\mathrm{M}}_{1\ldots N}$ | $N$ modelled scaling factors | Section 3.2 | Ma, Mb |
| $fv^{\mathrm{M}}_{90}$ | Truncated modelled firn thickness variations | Section 3.2 | Ma, Mb |
| $fv^{\mathrm{A}}$ | Adjusted firn thickness variations | Section 3.1.1, Eq. 3 | A1a, A2a, A1b, A2b |
| $r^{\mathrm{A}}$ | Altimetric residuals | Section 3.1, Eq. 1 | A1a, A2a, A1b, A2b |
| $fv^{\mathrm{E2}}$ | Scaled firn thickness variations | Appendix A1 | |
| $fv^{\mathrm{E3}}$ | Modified adjusted firn thickness variations | Appendix A1 | |

* or rather anomalies (Section 2)

*Data availability.* The TUD and JPL altimetry products are available for download at https://doi.pangaea.de/10.1594/PANGAEA.897390 (Schröder et al., 2019b) and https://doi.org/10.5067/L3LSVDZS15ZV (Nilsson et al., 2021), respectively. The GSFC firn model is available
for download at https://doi.org/10.5281/zenodo.7054574 (Medley et al., 2022b). Results and data of this study can be obtained from the authors without conditions.

*Author contributions.* CRediT Taxonomy: MK: conceptualization, data curation, formal analysis, investigation, methodology, software, visualization, writing - original draft. MH: conceptualization, funding acquisition, methodology, supervision, writing - review and editing. EB: formal analysis, writing - review and editing. MW: data curation, writing - review and editing. LS: funding acquisition, writing - review and
editing. SV, PKM, MvdB: resources (provision of the IMAU firn model), writing - review and editing.

*Competing interests.* MvdB is a member of the editorial board of the journal. All other authors declare that they have no conflict of interest.



*Acknowledgements.* MK and EB were funded by the Deutsche Forschungsgemeinschaft (DFG) as part of the Special Priority Program (SPP) 1158 'Antarctic Research with Comparative Investigations in Arctic Ice Areas' through grant HO 4232/10-1 (project number 442929109) and grant SCHE 1426/26-1 and 2 (project number 404719077), respectively. MW was funded through grant HO 4232/4-2 (project number 800  313917204) from the DFG as part of the SPP 1889 'Regional Sea Level Change and Society (SeaLevel)'. SV acknowledges funding by the Netherlands Organisation for Scientific Research (NWO) grant OCENW.GROOT.2019.091.



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
