# Peer review of "How well can satellite altimetry and firn models resolve Antarctic firn thickness variations?"

_The Cryosphere, 2023_

## Referee Comment (RC1)

**Review on:  How well can satellite altimetry and firn models resolve Antarctic firn thickness variations?**

by Kappelsberger et al 2023

This study combines firn model data and altimetry data to study the spatial and temporal variations in firn thickness over Antarctica. To make this new dataset they use two firn models, and altimetry data from multiple missions, both radar and laser. They do a thorough statistical analysis to validate the new data set. This is a really interesting study and research question. However, I do think major changes are needed before this paper can be published, there are too many grammar errors and overall it is very long and difficult to read and keep track of all the introduced symbols. Below are some more specific comments.

**Major comments:**

Overall this paper is very dense/wordy, it reads more like a report than a scientific paper. Some sentences and paragraphs seem to just be added as filling without a clear purpose. I suggest heavily cutting the text, especially in the introduction and data section. See some specific suggestions further down.

There are a lot of sentences where you use a full stop rather than a comma. Furthermore, there are a lot of grammar errors, like Line 94/95 "*Nevertheless, discrepancies still remain. (See Section 2.3 for further details on comparisons between altimetry and firn models.)*" It is not grammatically correct to start a new sentence with a paraphrase. All these things are disrupting the reading flow. This paper could really benefit from a proofread!
See some specific suggestions further down.

There are a lot of inconsistencies in the figures, especially with the colour bars (spatial plots). It seems random if the colour bars have a min or max arrow extension, or both arrow extensions. These colour bar inconsistencies are seen throughout the paper. Also, in Figure S13 red is negative and blue is positive, just below in Figure S14 purple/blue is negative and brown/red is positive.  Please go through them all and double-check if all the colour bars are correct.

Snow and firn are mentioned many times but the terms are not well defined here. The manuscript could benefit from a clear distinction between snow and firn. Further, sometimes it says snow/firn layers in plural and sometimes the layer is singular, and firn pack is also used. It seems like the terms are used interchangeably, please make sure to be consistent, as they have different meanings.

**Minor comments:**

In section 2.4 it seems like you start to present and discuss results (Fig. 1 and 2), this feels out of place in the method sections.

In Section 2.1 you refer to the two altimetry datasets as "TUD altimetry" and "JPL altimetry" in lines 124-126. But in the section, you also call them "TUD" and "JPL", or "TUD product" and "JPL product", or "Schröder et al. (2019a)" and "Nilsson et al. (2022)". When you make a definition in the beginning you should consistently use that.

Likewise in section 2.2, you refer to the two firn models as IMAU firn model and GSFC firn model, within the same section you call them "IMAU", "IMAU model", "GSFC", and "GSFC model". When you make a definition in the beginning you should consistently use that.

In the caption of Figure 2, you write "Color" which is American English other places you write "metre" which is British English. Please choose one way and be consistent.

**Specific comments:**

L21: The references " (Horwath et al., 2022; IPCC, 2021)" are here in alphabetical order, the rest of the references are in timely order oldest first.

L33: "in the snow and firn layer" This reads like there is no difference between snow and firn

L44: "statistically significant trends in the Antarctic (surface) mass balance" this statement needs a reference.

L52-53: "Earth system models have recently caught up in this regard (Lenaerts et al., 2019)". Is this sentences relevant? ESMs are not mentioned anymore.

L53: Suggest change "They are forced" to "They can be forced", because not all RCMs use reanalysis data.

L54: "data from 1979 onwards" I guess you refer to ERA-interim put we have ERA5 now starting in 1950 (Hersbach, et al 2020)

L58: "However, spatial variations in SMB show a poorer agreement. On a basin scale, the largest…" remove the full stop after "agreement" use a comma instead as the following sentence is a continuation.

L57-63: This could be written much shorter, it is not a review in Mottram et al 2021.

L64-70: Again this could be written much shorter, it is not a review in Verjans et al 2021. I do not think all the numbers are necessary

L71-79: In my opinion, this paragraph could be removed, as gravimetric mass balance is not the topic of this paper.

L80: Remove "By contrast"

L81: "They" who?

L83: Change "utilise(d)" to utilise

L84: Change "(ICESat-2)" to (ICESat), as your time period ends in 2017 and ICESat-2 was first launched in 2018.

L86: "laser signals are reflected at or near the ice sheet surface". I think you mean "laser signals are reflected at or near the surface".

L87: Here you have snow/firn layers in plural in L33 the layer is singular.

L87: "If elevation changes due to changing ice flow can either be neglected or subtracted…" Please argue where or when they can be neglected.

L94: "Nevertheless, discrepancies still remain. (See Section 2.3 for further details on comparisons between altimetry and firn models.)" Incorrect grammar, suggest rewriting to: Nevertheless, discrepancies between altimetry and firn models still remain and are discussed.

L98-99: "The reason likely lies in errors in the involved altimetry and modelling results" Is that not always the case? either the errors are from the altimetry or the model.

L99: "Therefore" what?

L99: What is meant by a "Steady-state" density model?

L102: I suggest merging sections 1.2 and 1.3 and removing the text about gravimetry in Sec. 1.3

L104: Remove "(Section 1.2)"

L111: I suggest also merging sections 1.2, and 1,4 and then writing the "purpose part" in the last paragraph in Sec. 1.2.

L113: Remove "(Section 1.1)"

L118-119: "For the first time, the entire spatial information present in both the altimetry products and modelling outputs, together with the high (monthly) temporal resolution of gridded altimetry products, is jointly exploited." This is a convoluted sentence, e.g "altimetry products" is mentioned twice, please rewrite.

L131-132: Suggest changing the full stop to a comma between "differs. Thus,"

L140: Here you write firn pack and firn layer as earlier.

L162-163: Why mention the Shepherd et al 2019 data when you are not using it? I suggest removing these lines.

L172-173: you write "In accordance with the altimetry data, we involve firn thickness changes from the grounded AIS excluding the Antarctic Peninsula and the period May 1992 to December 2017" This sentence says that you exclude the AP and the period 1992-2017, it should likely say that you use the period 1992-2017

L177-178: In SMB components you are missing refreeze/runoff

L179: "dynamically downscaled with RACMO2.3p2" does that mean that you have run the RACMO model?

L182: "firn layer" singular

L192: " Medley et al. (2022a) built a new model" should it not say parameterisation instead of model?

L198: Suggest to delete this line, it sounds like a rapport.

L199: Again "TUD, JPL"

L201: Missing reference to ERA-Interim.

L204: Define the sign of rates, is a positive rate making the surface go up?

L206: Remove "section 2.2"

L207:"more positive" does that mean thicker firn or a faster thickness rate?

L208: "less negative" slower thickness rate, or slower thinning rate?

L210: Do you mean the average ice sheet-wide seasonal amplitude in firn thickness? Or are you talking about ice sheet seasonal amplitude?

L211-213: It is unclear what these numbers are, Are they seasonal differences in firn thickness? Are they integrated over the entire ground AIS?

L215: Remove " In this section" it sounds like a rapport.

L215: Which basin definition?

L219: ". (For the JPL altimetry, h A2, and the GSFC firn model, f Mb, similar time series are shown in Fig. S1.)" using the parentheses is incorrect grammar.

L220: "Agreement between hA1 and fMa is generally good on interannual scales. Differences appear in the long-term trends." these are two very short sentences, I suggest removing the full stop.

L225: Here you say the entire period is 1993-2017, does it not start in 1992?

L230: put s on "model" since you are referring to both models.

L230-231: Using the parentheses is incorrect grammar.

L247: ". It is explained in Section 3.1.1" maybe change to ", this is explained in Section 3.1.1"

L262: Replace full stop after regression with comma

L284-286: "Comprehensive and general references…." this feels like it is a bit misplaced perhaps to put in the references after "respectively"

L305: Suggest to merge section 3.1.1 and 3.1.2 as sec. 3.1.2 is very short.

L306: Suggest remove " ('R squared')"

L311: Label eq 5a er 6 instead. It is a bit strange to have Eq 5 followed by eq 5a

L328: Shouldn't you also refer to fvE1?

L330: "These alternative variations are called scaled firn thickness variations. We refer to them by fvE2", replace full stop with a comma.

L332: "These alternative variations are called modified adjusted firn thickness variations. We refer to them by fvE3", replace full stop with a comma.

L345-355: Please add references for the Kolmogorov-Smirnov test

L360-361: ". (For example, power-law with κ = −1 and κ = −2 represents flicker and random walk noise, respectively.)" using the parentheses is incorrect grammar

L365-366: ". (Note that the residuals may additionally contain signals related to variations in ice flow dynamics or subglacial hydrology.)" using the parentheses is incorrect grammar

L398: "standard deviation" => std, the abbreviation is already introduced.

L402:"(In the case of  data gaps in the altimetry time series, this equality holds approximately.)."

L409: Remove "(Section 3.1)"

L431: "(The rms of all versions of fvA and fvM is illustrated in Fig. S17a–d and Fig. S18a, b, respectively.)"

L441:"(Corresponding rms maps of differences are displayed in Fig. S17–S19)."

L448: Section 4.2.4, you already have a section with the same title, this is confusing.

L456-457: "(Fig. S20 and Fig. S21 further shows maps of the residuals rms and of R2 for different versions of regression and both time periods. Table S1 lists basin averages of R2 for the period before 2003.)

L472: "(Basin-mean time series of all regression results and versions are presented in Fig. S22–S24.)"

L478-479: "(For the larger subset of selected grid points, Fig. S25 and S26 display the psd of the regression results from A1a and A2a, respectively.)"

L485-486: "(HECTOR only yields numerical stable results for $-3$.)"

L495: "(For all versions and both PCA, the original and rescaled patterns are illustrated in Fig. S27–S29)."

L507: "power spectral density (psd)" => psd, the abbreviation is already introduced.

L510-511: "(Unlike our analysis, they did not co-estimate a quadratic or seasonal term.)"

L548-549:"(The histograms and cumulative histograms for all basins are shown in Fig. S30 and S31, respectively.)

L556-557: "(See also Table 2 for an overview of the various differences in firn thickness variations fv and their description.)"

L595: "also high" please put a number on that.

L610: What is meant by " measurement noise"?

L731: Changes the full stop between noted and By to a comma, as it is the same sentence

**Figures:**
Fig1: I suggest to refer to Fig3 in the caption for basin locations.

Fig3: I suggest that you put all the areas/regions which you mention in the text

Fig3 caption: you write ". Drainage basins of the EAIS and WAIS used in this study (thick black lines) following Rignot et al. (2011a, b)." but then in L298-299 you talk about regions and multiple Rignot basins forming one basin. Please clarify when you talk about Rignot basins, "your" basins or other regions

Fig9 caption: Replace full stop with and "Histograms. Vertical"

Fig10: The left colour bar has only max extend while the right has both min and max extend. Also are there even Coefficients of determination for the regression that are negative, like the min extend suggests?

Fig11: Shouldn't there be a min extension on the colour bar?

Fig A1: Shouldn't there be a min extension on the colour bar?

**References:**

Hersbach, H, Bell, B, Berrisford, P, et al. The ERA5 global reanalysis. *Q J R Meteorol Soc*. 2020; 146: 1999–2049. https://doi.org/10.1002/qj.3803

---

## Author Comment (AC1)

**Reply on Review 1 (Kappelsberger et al., 2023)**

Thank you very much for the very detailed and helpful review of our manuscript. The comments are very helpful in improving and shortening the manuscript. Below we respond to each comment and describe how we plan to revise the manuscript to make it more concise and thus easier to read. Reviewer comments are marked in italics.

**1 Major comments**

**1.1** *Overall this paper is very dense/wordy, it reads more like a report than a scientific paper. Some sentences and paragraphs seem to just be added as filling without a clear purpose. I suggest heavily cutting the text, especially in the introduction and data section. See some specific suggestions further down.*

- We fully agree that the paper will benefit greatly from being shortened. The following parts of the manuscript will be significantly shortened or even deleted:
    - L55-63 (introduction)
    - L66-70 (introduction)
    - L72-79 (introduction)
    - L91-101 (introduction)
    - L195-231, Fig. 2 (data)
    - L320-326 (methods)
    - L414-419, Fig. 6 (results)
- In addition, we plan to change our methods slightly for ease of presentation and explanation.
    1. Data section: Here, we will describe that, we remove the offset, linear, quadratic and seasonal signals from the altimetric elevation changes, $h^{\mathrm{A}}$, and from the modelled firn thickness changes, $f^{\mathrm{M}}$, to obtain the altimetric variations, $hv^{\mathrm{A}}$, and the modelled firn thickness variations, $fv^{\mathrm{M}}$. Thus, $hv^{\mathrm{A}}$ and $fv^{\mathrm{M}}$ will represent our "input data sets" for our methods below.
    2. Methods section: Based on this, we will use $hv^{\mathrm{A}}$ as the input to our regression approach. This will simplify Eq. (1) to

$$hv^{\mathrm{A}}(t) = \sum_{n=1}^{N} e_n^{\mathrm{A}} PC_n^{\mathrm{M}}(t) + r^{\mathrm{A}}(t).$$

Thus, Eq. 2 and its accompanying text will no longer be needed at this stage as we will remove the offset, linear, quadratic and seasonal signals before. Likewise, Eq. 4 and its text are also no longer needed at this stage. We might change the order of the subchapters 3.1 and 3.2 due to the planned changes.

We note that by splitting the regression approach (Eq. 1) into two separate steps of parameter estimation (one for the offset, linear, quadratic and seasonal part, and a second one for the interannual patterns), we will imply a slightly different treatment of correlations between those parameters than in our original manuscript. However, the effect on the results will be small. Our conclusions will not be affected. With these changes the methods section will be shorter and we believe, much easier to read and understand.

**1.2** *There are a lot of sentences where you use a full stop rather than a comma.*

➢ Thank you for your detailed and specific suggestions below to improve the readability of our manuscript. We will take all these suggestions into account and rephrase or combine the relevant sentences in accordance with punctuation guidelines (Purdue OWL, 2024).

Originally, we have frequently chosen to write two shorter sentences rather than one longer sentence because less convoluted sentences are easier to write and read, especially for non-native English speakers. Nevertheless, we admit and agree that some of our short sentences may read a bit odd and we will improve those based on your suggestions.

**1.3** *Furthermore, there are a lot of grammar errors, like Line 94/95 "Nevertheless, discrepancies still remain. (See Section 2.3 for further details on comparisons between altimetry and firn models.)" It is not grammatically correct to start a new sentence with a paraphrase. All these things are disrupting the reading flow. This paper could really benefit from a proofread! See some specific suggestions further down.*

➢ We will go through all these sentences, remove the parentheses and rephrase the sentences if necessary.

**1.4** *There are a lot of inconsistencies in the figures, especially with the colour bars (spatial plots). It seems random if the colour bars have a min or max arrow extension, or both arrow extensions. These colour bar inconsistencies are seen throughout the paper. Also, in Figure S13 red is negative and blue is positive, just below in Figure S14 purple/blue is negative and brown/red is positive. Please go through them all and double-check if all the colour bars are correct.*

➢ The min and max arrow extensions of the colour bars are not random. We only use the arrows when there are values beyond the colour scale range. To avoid confusion and to clarify that there are no inconsistencies, we will add the following sentence to each figure that has a mixture of no arrow extensions and arrow extensions: "Colour bar arrows indicate that the value range exceeds the limits of the colour scale."

➢ The colour scale in Fig. S13 is indeed inconsistent to the other similar figures. We will change the Fig. S13 colour bar accordingly.

**1.5** *Snow and firn are mentioned many times but the terms are not well defined here. The manuscript could benefit from a clear distinction between snow and firn. Further, sometimes it says snow/firn layers in plural and sometimes the layer is singular, and firn pack is also used. It seems like the terms are used interchangeably, please make sure to be consistent, as they have different meanings.*

➢ These terms should indeed be clarified and consistently used, thank you. For the most part, we have used the term 'firn' to refer to both snow and firn. For example, we use 'firn thickness changes' rather than 'snow and firn thickness changes'. Also, the top layer of the firn model, which could be considered a snow layer, is already considered part of the firn layer.

The first time the terms 'snow' and 'firn' are used in the Introduction, we will define/clarify: "Snow is typically used for the seasonal snow cover, i.e. less than a year old. Firn refers to multiyear snow and is defined as the transition from freshly fallen snow to glacier ice (van den Broeke, 2008). However, we do not consider a separate snow layer, it is also referred to by the term firn layer."

**2 Minor comments**

**2.1** *In section 2.4 it seems like you start to present and discuss results (Fig. 1 and 2), this feels out of place in the method sections.*

> ➢ Section 2.4 presents the original input data sets, without applying any of the methods described afterwards in Section 3. This is done to ensure that the reader is familiar with the original data sets used. In Fig. 1 (dash-dotted lines) and Fig. 2, we show the basin-mean time series and rms maps because this kind of presentation is commonly used when comparing results from altimetry and firn modelling. However, we acknowledge that this section feels out of place as we are already discussing the data. We will remove Section 2.4 as the content mentioned is either provided elsewhere in the manuscript or, as the data sets used have been published and discussed, can be assumed to be known. We will also remove Fig. 2 and we will modify Fig. 1.

**2.2** *In Section 2.1 you refer to the two altimetry datasets as "TUD altimetry" and "JPL altimetry" in lines 124-126. But in the section, you also call them "TUD" and "JPL", or "TUD product" and "JPL product", or "Schröder et al. (2019a)" and "Nilsson et al. (2022)". When you make a definition in the beginning you should consistently use that. Likewise in section 2.2, you refer to the two firn models as IMAU firn model and GSFC firn model, within the same section you call them "IMAU", "IMAU model", "GSFC", and "GSFC model". When you make a definition in the beginning you should consistently use that.*

> ➢ In fact, we use many abbreviations. This may be overwhelming rather than helpful for clarity. We therefore suggest that we no longer use the abbreviations TUD, TUD altimetry, JPL, JPL altimetry, IMAU, IMAU model, GSFC and GSFC model. Instead, we will use the references of the data sets.

**2.3** *In the caption of Figure 2, you write "Color" which is American English other places you write "metre" which is British English. Please choose one way and be consistent.*

> ➢ We will change it to "colour" as we are using British English.

**3 Specific comments**

*L21: The references " (Horwath et al., 2022; IPCC, 2021)" are here in alphabetical order, the rest of the references are in timely order oldest first.*

> ➢ Absolutely, that is not consistent. To be precise, the first reference refers to the first part of the sentence and the second reference refers to the second part. We will change this accordingly: "Ice-mass loss from Antarctica contributed ~6 % to this rise (Horwath et al., 2022), and is likely to continue (IPCC, 2021)."

*L33: "in the snow and firn layer" This reads like there is no difference between snow and firn*

> ➢ We will change this to "in the firn layer" according to our definition in 1.5.

*L44: "statistically significant trends in the Antarctic (surface) mass balance" this statement needs a reference.*

> ➢ We will add the reference King and Watson (2020).

*L52-53: "Earth system models have recently caught up in this regard (Lenaerts et al., 2019)". Is this sentences relevant? ESMs are not mentioned anymore.*

> ➢ We will delete this sentence.

*L53: Suggest change "They are forced" to "They can be forced", because not all RCMs use reanalysis data.*

➢ We will combine this sentence with the previous sentence as follows: "When the main goal of RCMs is to realistically simulate the ice sheet weather, as is the case here, they are forced by atmospheric reanalysis products (van Wessem et al., 2018; Agosta et al., 2019)."

*L54: "data from 1979 onwards" I guess you refer to ERA-interim put we have ERA5 now starting in 1950*

➢ Thanks for this note. In order to keep the manuscript shorter, we will delete this part of the sentence.

*L58: "However, spatial variations in SMB show a poorer agreement. On a basin scale, the largest…" remove the full stop after "agreement" use a comma instead as the following sentence is a continuation.*

➢ This part will be shortened and reworded.

*L81: "They" who?*

➢ The observations from satellite altimetry. We will replace "They" with "These measurements".

*L87: Here you have snow/firn layers in plural in L33 the layer is singular.*

➢ According to our definition above (see 1.5), we will change this to: „(…) while radar signals penetrate into the upper firn layer." Here, and everywhere else we will use the singular.

*L87: "If elevation changes due to changing ice flow can either be neglected or subtracted…" Please argue where or when they can be neglected.*

➢ We will rephrase: "Altimetric elevation changes can be compared to modelled firn thickness changes in regions where changes in ice flow are either insignificant or are known and can be subtracted".

*L94: "Nevertheless, discrepancies still remain. (See Section 2.3 for further details on comparisons between altimetry and firn models.)" Incorrect grammar, suggest rewriting to: Nevertheless, discrepancies between altimetry and firn models still remain and are discussed.*

➢ This sentence is not part of the revised manuscript anymore.

*L98-99: "The reason likely lies in errors in the involved altimetry and modelling results" Is that not always the case? either the errors are from the altimetry or the model.*

➢ In the revised manuscript, we plan to delete this sentence and much of the paragraph to which it belongs.

*L99: "Therefore" what?*

➢ Because of the large errors, the models cannot be used in a rigorous, deterministic manner to derive altimetric ice mass changes. Therefore, a simplified approach is commonly applied. However, in the revised manuscript that sentence will be deleted.

*L99: What is meant by a "Steady-state" density model?*

➢ A steady-state density model refers to a spatially but not temporally resolved density "map". Thus, only a single density "map" is used to translate all time slices of surface elevation change "maps/grids" into mass change "maps/grids". This implies the assumption that the density to which the elevation changes refer is steady/constant over time. However, in the revised manuscript this sentence will be deleted.

*L118-119: "For the first time, the entire spatial information present in both the altimetry products and modelling outputs, together with the high (monthly) temporal resolution of gridded altimetry products, is jointly exploited." This is a convoluted sentence, e.g "altimetry products" is mentioned twice, please rewrite.*

➢ We will rewrite: "For the first time, the entire spatial and temporal information present in the altimetry products is exploited together with the modelling results."

*L140: Here you write firn pack and firn layer as earlier.*

➢ We will change this to: „(…) reduce the sensitivity to variations in the properties of the upper firn layer."

*L177-178: In SMB components you are missing refreeze/runoff*

➢ The IMAU firn model is not forced with all SMB components, but with the fluxes and boundary conditions at the top of the firn layer. Refreezing is a process happening in the firn itself, so this is calculated within the IMAU-FDM model, same as runoff.

*L179: "dynamically downscaled with RACMO2.3p2" does that mean that you have run the RACMO model?*

➢ No, this is just a description of the data we used. The reference for the RACMO model is van Wessem et al. (2018). We will clarify this in the revised manuscript.

*L182: "firn layer" singular*

➢ We will keep it in the singular, the "firn layer".

The following comments (L201-230) refer to Sections 2.3 and 2.4, which will be deleted. For this reason, we provide only some explanation, but no proposed changes:

*L201: Missing reference to ERA-Interim.*

➢ The reference of ERA-Interim, Dee et al. (2011), was already mentioned in L55.

*L204: Define the sign of rates, is a positive rate making the surface go up?*

➢ Positive: +; negative: -. Yes, if the ice sheet surface is going up, the rate is positive.

*L207:"more positive" does that mean thicker firn or a faster thickness rate?*

➢ "More positive rates" means larger/faster thickening rates (not faster densification rates);
e.g.: modelled rate v1.1: +2 cm/yr; modelled rate v1.2A: +5 cm/yr

*L208: "less negative" slower thickness rate, or slower thinning rate?*

➢ "Less negative rates" means lower/slower thinning rates;
e.g.: modelled rate v1.1: -5 cm/yr; modelled rate v1.2A: -2 cm/yr

*L210: Do you mean the average ice sheet-wide seasonal amplitude in firn thickness? Or are you talking about ice sheet seasonal amplitude?*

➢ The estimates of the average ice-sheet-wide seasonal amplitude give discrepant results.

*L211-213: It is unclear what these numbers are, Are they seasonal differences in firn thickness? Are they integrated over the entire ground AIS?*

➢ These numbers are the seasonal amplitudes in firn thickness averaged over the ice sheet and based on the various altimetry products and firn models.

*L215: Which basin definition?*

➢ The basin definition used in this study.

*L220: "Agreement between hA1 and fMa is generally good on interannual scales. Differences appear in the long-term trends." these are two very short sentences, I suggest removing the full stop.*

➢ Deleted.

*L225: Here you say the entire period is 1993-2017, does it not start in 1992?*

➢ Indeed, it starts in May 1992.

*L230: put s on "model" since you are referring to both models.*

➢ Deleted.

*L262: Replace full stop after regression with comma*

➢ We will rephrase these sentences on the basis of your comments and those of the second reviewer: "The stochastic model of our regression is represented by a different weighting of observations from two time periods. As results from the older altimetry missions generally have a higher noise level (Schröder et al., 2019a; Nilsson et al., 2022), $hv^A$ after 2003 are weighted by 1, while $hv^A$ before 2003 are given a different (usually lower) weight."

*L306: Suggest remove " ('R squared')"*

➢ Instead of removing "R squared", we plan to remove "the coefficient of determination" and refer to $R^2$ as R squared throughout.

*L328: Shouldn't you also refer to fvE1?*

➢ $fv^{E1}$ would be equal to $fv^M$ since we do not adjust/scale anything.

*L330: "These alternative variations are called scaled firn thickness variations. We refer to them by fvE2", replace full stop with a comma.*

➢ We will replace the full stop with ", and".

*L332: "These alternative variations are called modified adjusted firn thickness variations. We refer to them by fvE3", replace full stop with a comma.*

➢ We will replace the full stop with ", and".

*L345-355: Please add references for the Kolmogorov-Smirnov test*

➢ We will add Massey (1951), Miller (1956) and Marsaglia et al. (2003).

*L610: What is meant by " measurement noise"?*

➢ This refers to random errors in the altimetry results caused by the measurement (method) itself. We will briefly introduce that different effects can be responsible for errors in altimetry.

*L731: Changes the full stop between noted and By to a comma, as it is the same sentence*

➢ We will combine these sentence and the sentences in L726: "However, it is important to note that (…)".

**4 Figures**

*Fig3 caption: you write ". Drainage basins of the EAIS and WAIS used in this study (thick black lines) following Rignot et al. (2011a, b)." but then in L298-299 you talk about regions and multiple Rignot basins forming one basin. Please clarify when you talk about Rignot basins, "your" basins or other regions*

➢ In Fig. 3 we will clarify: "Drainage basins of the EAIS and WAIS used in this study (thick black lines), slightly modified from the definition of Rignot et al. (2011a, b)."

➢ In L298-299 we will change "regions" to "basins": "We individually apply the PCA to $fv^M$ for 10 selected basins (Fig. 3). To define the basins, we make use (…)"

*Fig3: I suggest that you put all the areas/regions which you mention in the text*

➢ Good idea, thanks. We will incorporate the following modified figure:

[Figure]

*Fig10: The left colour bar has only max extend while the right has both min and max extend. Also are there even Coefficients of determination for the regression that are negative, like the min extend suggests?*

➢ The arrow extensions are correct: (a, b) max extension only, (c) min extension only. Yes, there are very few grid cells where the coefficients of determination for the period after 2003 are negative. This is due to the different weighting of altimetry observations before and after 2003. Without weighting, the coefficients of determination would all be positive. However, due to the different weighting, it is possible that the adjusted firn thickness variations describe less of the variance of

the altimetric variations for one of the two periods. This is more likely for the period before 2003, as altimetric observations before 2003 are usually weighted lower (see Fig. S21). We have tried to explain this in L406-409. In the revised manuscript we will explain this more clearly.

*Fig11: Shouldn't there be a min extension on the colour bar?*

➢ No, there are no values below −3 because kappa values by HECTOR are always greater than or equal to −3. This is explained in L486. We will rephrase.

*Fig A1: Shouldn't there be a min extension on the colour bar?*

➢ There shouldn't be a min extension. However, we actually made a mistake when labelling the colour scale. The labels were in the wrong place. The correct labelling is shown below.

[Figure]

**All the following comments will be revised as suggested. Thank you for your scrutiny which will help us to improve many details of the manuscript.**

- *L57-63: This could be written much shorter, it is not a review in Mottram et al 2021.*
- *L64-70: Again this could be written much shorter, it is not a review in Verjans et al 2021. I do not think all the numbers are necessary*
- *L71-79: In my opinion, this paragraph could be removed, as gravimetric mass balance is not the topic of this paper.*
- *L80: Remove "By contrast"*
- *L83: Change "utilise(d)" to utilise*
- *L84: Change "(ICESat-2)" to (ICESat), as your time period ends in 2017 and ICESat-2 was first launched in 2018.*
- *L86: "laser signals are reflected at or near the ice sheet surface". I think you mean "laser signals are reflected at or near the surface".*
- *L102: I suggest merging sections 1.2 and 1.3 and removing the text about gravimetry in Sec. 1.3*
- *L104: Remove "(Section 1.2)"*
- *L111: I suggest also merging sections 1.2, and 1,4 and then writing the "purpose part" in the last paragraph in Sec. 1.2.*
- *L113: Remove "(Section 1.1)"*
- *L131-132: Suggest changing the full stop to a comma between "differs. Thus,"*
- *L162-163: Why mention the Shepherd et al 2019 data when you are not using it? I suggest removing these lines.*

- *L172-173: you write "In accordance with the altimetry data, we involve firn thickness changes from the grounded AIS excluding the Antarctic Peninsula and the period May 1992 to December 2017" This sentence says that you exclude the AP and the period 1992-2017, it should likely say that you use the period 1992-2017*
- *L192: " Medley et al. (2022a) built a new model" should it not say parameterisation instead of model?*
- *L198: Suggest to delete this line, it sounds like a rapport.*
- *L199: Again "TUD, JPL"*
- *L206: Remove "section 2.2"*
- *L215: Remove " In this section" it sounds like a rapport.*
- *L219: ". (For the JPL altimetry, h A2, and the GSFC firn model, f Mb, similar time series are shown in Fig. S1.)" using the parentheses is incorrect grammar.*
- *L230-231: Using the parentheses is incorrect grammar.*
- *L247: ". It is explained in Section 3.1.1" maybe change to ", this is explained in Section 3.1.1"*
- *L284-286: "Comprehensive and general references…." this feels like it is a bit misplaced perhaps to put in the references after "respectively"*
- *L305: Suggest to merge section 3.1.1 and 3.1.2 as sec. 3.1.2 is very short.*
- *L311: Label eq 5a er 6 instead. It is a bit strange to have Eq 5 followed by eq 5a*
- *L360-361: ". (For example, power-law with $\kappa = -1$ and $\kappa = -2$ represents flicker and random walk noise, respectively.)" using the parentheses is incorrect grammar*
- *L365-366: ". (Note that the residuals may additionally contain signals related to variations in ice flow dynamics or subglacial hydrology.)" using the parentheses is incorrect grammar*
- *L398: "standard deviation" => std, the abbreviation is already introduced.*
- *L402:"(In the case of data gaps in the altimetry time series, this equality holds approximately.)."*
- *L409: Remove "(Section 3.1)"*
- *L431: "(The rms of all versions of fvA and fvM is illustrated in Fig. S17–d and Fig. S18a, b, respectively.)"*
- *L441:"(Corresponding rms maps of differences are displayed in Fig. S17–S19)."*
- *L448: Section 4.2.4, you already have a section with the same title, this is confusing.*
- *L456-457: "(Fig S20 and Fig. S21 further shows maps of the residuals rms and of R2 for different versions of regression and both time periods. Table S1 lists basin averages of R2 for the period before 2003.)*
- *L472: "(Basin-mean time series of all regression results and versions are presented in Fig. S22–S24.)"*
- *L478-479: "(For the larger subset of selected grid points, Fig. S25 and S26 display the psd of the regression results from A1a and A2a, respectively.)"*
- *L485-486: "(HECTOR only yields numerical stable results for –3.)"*
- *L495: "(For all versions and both PCA, the original and rescaled patterns are illustrated in Fig. S27–S29)."*
- *L507: "power spectral density (psd)" => psd, the abbreviation is already introduced.*
- *L510-511: "(Unlike our analysis, they did not co-estimate a quadratic or seasonal term.)"*
- *L548-549:"(The histograms and cumulative histograms for all basins are shown in Fig. S30 and S31, respectively.)*

- *L556-557: "(See also Table 2 for an overview of the various differences in firn thickness variations fv and their description.)"*
- *L595: "also high" please put a number on that.*
- *Fig1: I suggest to refer to Fig3 in the caption for basin locations.*
- *Fig9 caption: Replace full stop with and "Histograms. Vertical"*

Additional references:

- Marsaglia, G., Tsang, W., and Wang, J.: Evaluating Kolmogorov's Distribution, Journal of Statistical Software, 8, 1–4, https://doi.org/10.18637/jss.v008.i18, 2003.
- Massey, F.: The Kolmogorov-Smirnov Test for Goodness of Fit, Journal of the American Statistical Association, 46, 68–78, 1951.
- Miller, L.: Table of Percentage Points of Kolmogorov Statistics, Journal of the American Statistical Association, 51, 111–121, 1956.
- Purdue OWL: Identifying Independent and Dependent Clauses. https://owl.purdue.edu/owl/general_writing/punctuation/independent_and_dependent_clauses/, last access: 29 January 2024.

---

## Author Comment (AC2)

**Reply on Review 2 (Kappelsberger et al., 2023)**

Thank you very much for this supportive feedback on our manuscript. Your suggestions, such as including a flowchart visualising the entire workflow, are very helpful in improving the presentation of our manuscript. Below we respond to each comment and describe how we plan to revise the manuscript to make it shorter and easier to follow and read. Reviewer comments are marked in italics.

**1 Major concerns**

*The manuscript is very long and for a long time, I was unable to see where we were going and how all the notation, techniques, differences etc. were to be used. I realize that the authors want to be systematic by introducing all the methodology in Section 3 before using it in Section 4 and discussing it in Section 5. But this means that, until somewhere in Sections 4 and 5, I had been loaded with a large amount of notation, techniques etc. without really knowing why I needed to know this.*

**1.1** *Try to shorten the manuscript. I know you want to be thorough and systematic, but is there really no way of making it shorter? It is a very long read.*

> We fully agree that the paper will benefit greatly from being shortened. The following parts of the manuscript will be significantly shortened or even deleted:
>   - L55-63 (introduction)
>   - L66-70 (introduction)
>   - L72-79 (introduction)
>   - L91-101 (introduction)
>   - L195-231, Fig. 2 (data)
>   - L320-326 (methods)
>   - L414-419, Fig. 6 (results)

> In addition, we plan to change our methods slightly for ease of presentation and explanation.
>   1. Data section: Here, we will describe that we remove the offset, linear, quadratic and seasonal signals from the altimetric elevation changes, $h^A$, and from the modelled firn thickness changes, $f^M$, to obtain the altimetric variations, $hv^A$, and the modelled firn thickness variations, $fv^M$. Thus, $hv^A$ and $fv^M$ will represent our "input data sets" for our methods below.
>   2. Methods section: Based on this, we will use $hv^A$ as the input to our regression approach. This will simplify Eq. (1) to

$$hv^A(t) = \sum_{n=1}^{N} e_n^A \, PC_n^M(t) + r^A(t).$$

> Thus, Eq. 2 and its accompanying text will no longer be needed at this stage as we will remove the offset, linear, quadratic and seasonal signals before. Likewise, Eq. 4 and its text are also no longer needed at this stage. We might change the order of the subsections 3.1 and 3.2 due to the planned changes.

> We note that by splitting the regression approach (Eq. 1) into two separate steps of parameter estimation (one for the offset, linear, quadratic and seasonal part, and a second one for the interannual patterns), we will imply a slightly different treatment of correlations between those parameters than in our original manuscript. However, the effect on the results will be small. Our conclusions will not be affected. With these changes the methods section will be shorter and we believe, much easier to read and understand.

**1.2** *Somewhere in the beginning make an overview of what the problem is, and the pathway to solve it. Try to include a cartoon or flowchart showing the data coming in, all the intermediate products, the residuals, the analyses done on the residuals, and the use you make of these things. Include also the notation in each box of the chart and the relevant section numbers - not just where they are derived in Section 3 (as in Table B1) but also where they are calculated in Section 4 and used in Section 5. That would provide a road map for the reader making the journey through it all easier to navigate in.*

➢ This is a very good idea, thank you. At the beginning of the methods section we will include a flowchart showing the whole workflow. The flow chart might have the following structure:

[Figure]

Here, the (intermediate) results, their notation and where they are presented for the first time (Section 4/5) are shown in grey boxes. The main methodological steps (Section 3) to derive these (intermediate) results are shown in white boxes. Please note that the (sub)section numbers in the flowchart do not correspond to those of the submitted manuscript, as we plan to slightly change, swap or combine some (sub)sections.

**2 Minor issues and typos**

*L53: SOME RCMs specialize…*

➤ We will rephrase: "When the main goal of RCMs is to realistically simulate the ice sheet weather, as is the case here, they are forced by atmospheric reanalysis products (van Wessem et al., 2018; Agosta et al., 2019)."

*L178: ERA5: What is the resolution of ERA5? 31 km, right? And this is "downscaled" to 27 km, right? Please argue why? Do you have reason to believe the 27 km RACMO data is better than the 31 km ERA5 data as input to the FDM?*

➤ Yes, the resolution of ERA5 is 31 km. ERA5 is "dynamically downscaled" to 27 km by RACMO2.3p2 (van Wessem et al., 2018). Lenaerts et al. (2019) describe this method as follow: "(…) dynamical downscaling, that is, to use a high-resolution regional (atmospheric) climate model (...) forced at its boundaries with (...) reanalysis data. This ensures a sufficiently high resolution over the ice sheet as well as an explicit, physically realistic calculation of all SMB components."

Thus, RACMO2.3p3 has two advantages over ERA5. First, the spatial resolution is improved. An improvement from 31 km to 27 km does not sound like much, but for each grid cell it already means a 25 % reduction in area. By improving the spatial resolution, processes such as katabatic winds or orographic precipitation can be better simulated in complex, mountainous regions (Hansen et al., 2021). This leads directly to the second benefit: the improvement of simulating physical processes for the special case of polar regions. RACMO2.3p2 is coupled to a multilayer snow model (there is an interaction between the atmosphere and snow surface), snow albedo is considered, a drifting snow scheme is included (van Wessem et al., 2018).

*L181: MERRA2 is downscaled from what resolution to 12.5 km? And how?*

➤ MERRA-2 has a resolution of 0.625° longitude x 0.5° latitude, giving a resolution of 24 km x 56 km at a latitude of ±70° (Medley et al., 2022). Medley et al. (2022) used a downscaled version of MERRA-2, which they called "replay" MERRA-2 run. This replay run was produced within the NASA Downscaling Project (Tian et al., 2017).

In the revised manuscript, we will remove subsection 2.3, to which the following three comments refer:

- *L204: The use of parentheses and the two short sentences "Rates…" and "The three…" is quite clumsy here. Please rephrase.*
- *L206: "evaluated by comparing to"*
- *L214: Try to think of a better title for this sub-section*

*L219 (and many other places, e.g., 401): The use of parenthesis after a full stop is not the usual way of doing it. Either a parenthesis refers to and is part of the sentence that is full-stopped, or maybe it should not be a parenthesis at all.*

➤ We will go through all these sentences, remove the parentheses and rephrase the sentences if necessary.

*L262: The sentence starting with "Observations" is very difficult to read. Perhaps start with the fact that you find higher noise levels from the older sections and use this to motivate why you introduce a weighting. Also, say which variable (r^A?) represents this noise you talk of.*

*L265-269: This is very difficult to understand. Please see if you can rewrite it more clearly.*

➢ We will rephrase the whole paragraph and also consider the comment on L321. The following proposal for reformulation already refers to our proposal for the modified methodology, where $hv^A$, rather than $h^A$, applies as observations of the regression approach: "The stochastic model of our regression is represented by a different weighting of observations from two time periods. As results from the older altimetry missions generally have a higher noise level (Schröder et al., 2019a; Nilsson et al., 2022), $hv^A$ after 2003 are weighted by 1, while $hv^A$ before 2003 are given a different (usually lower) weight. The weight before 2003 is defined (individually for every grid point) by the ratio of the noise variance of $hv^A$ before and after 2003. We assess the noise by the high-pass filtered version of $hv^A$ separately for both periods (cf. Groh et al., 2019). The high-pass filtering consists of removing a low-pass filtered version of $hv^A$, where the low-pass filter is a Gaussian filter with a 6σ = 12 months filter width."

*Eqn 1, 2, and 4: These equations all include a, b, c, d1, …, but they are different (and subject to different regressions) in the three equations, right? Either change the notation or write this out very clearly.*

➢ Eq. 1 and 4 are two different regressions. Eq. 2 is not subjected to a different regression, only some of the signals estimated by Eq. 1 are removed. However, to simplify the methodological presentation, we plan to apply Eq. 2 as a separate regression before fitting the dominant temporal patterns in modelled firn thickness variations to altimetry (see the explanations above under 1.1).

*L321: "deterministic": What do you mean by deterministic? And later on (L669+673), you talk of "stochastic". Exactly what is stochastic? I cannot see any noise terms added anywhere in your methodology.*

➢ In general, the mathematical model of a least-squares adjustment consists of (1) the deterministic or functional model and (2) the stochastic model given by the variance-covariance matrix. Here, the deterministic model is given by Eq. 1. The stochastic model is not given by equations, but it is described in L262-268. The term "weighted regression" means that our stochastic model is only a diagonal matrix, so only variances and no covariances are considered. The term 'stochastic model' should be clearer with the reformulation of L262-268.

*L371-375: Was very difficult to read. I think I understood it when the results were shown later on, but when reading it here I did not get it.*

➢ To make that part easier to understand we plan to include concrete matrix sizes: "For each PCA, we set up one aggregated 'super data matrix' in which we arrange the time series of residuals/residual differences for grid cells and for the different versions into a single set of time series. Specifically, our data sets comprises m = 90638 points in space (entire area under investigation) and p = 108 points in time (2003–2017). Thus, for the first and second PCA, the super data matrix has the size of 4m x p and 2m x p, respectively. The PCA is conducted to identify the dominant temporal patterns (PCs), which are shared by all versions, together with their space-dependent and version-dependent amplitudes, i.e. their spatial patterns (EOFs). Each identified mode thus consists of one joint PC (1 x p) and four/two EOFs (4m x 1/2m x 1) in the case of the first/second PCA."

*L380: You say that the fv^Ma are standardized prior to PCA, but in Fig 4 you say that the EOFs have units of m. How can the EOFs have units if the input is standardized and thereby non-dimensional?*

➢ Thank you very much for pointing this out. Indeed, since $fv^M$ are standardized prior to PCA, the EOFs should not have a unit. We will remove the unit of meter in Fig. 4.

*Fig 7: Why are d-f not identical (or at least similar) to the EOFs in Fig 4? Are you not projecting the model signal on to the PCs that came out of a PCA on exactly that signal? Should that not be a way of recovering the EOFs, i.e., by projecting the signal onto the PCs? Or does it have to do with standardized vs non-standardized signals?*

➢ Exactly, it has to do with the standardisation. In the revised manuscript, we plan to explain it in a different way: "To regain interpretable magnitudes of the EOFs, the EOFs are multiplied by the corresponding std of the time series of $fv^M$ for each grid cell (which was previously used for standardisation). After this restoration of the signal amplitudes, we no longer speak of EOFs but of modelled scaling factors, $e^{M}$".

Also, we will note in Fig. 6: "(d-f) is the same as Fig. 4a-c but with restored signal amplitudes for each grid cell." In other words, Fig. 4a-c is the standardised version of Fig. 6d-f. The patterns of Fig. 4a-c and Fig. 6d-f are similar, but as the standardisation is done for each grid cell, this may not be obvious.

*L455: What is R_s? I cannot remember having this introduced before.*

➢ True. It should be $R^2$.

*L478: "underlying time series IS displayed"*

➢ Here, we wanted to refer to both time series, $fv^A$ and $r^A$. To clarify that, we will write: "The underlying time series of $fv^A$ and $r^A$ are (…)".

*L635: The sentence starting with "Thus, the …" is difficult to understand.*

➢ We will delete this sentence as it repeats the previous one.

*L712: The sentence starting with "We deliberately" does not read well. Particularly the word "deliberately" seems odd. Please try to rephrase the sentence.*

➢ We will rephrase the first and second sentence of the paragraph: "We developed a new approach that combines satellite altimetry and firn modelling results to resolve Antarctic firn thickness variations at a high temporal (monthly) and spatial (grid scale of 10km) resolution."

*L723: outperforms*

➢ We will rephrase: "The adjusted firn thickness variations, $fv^A$, outperform …"

*L735-736: The sentence starting with "Across basin 8" makes it sound as if you only did this spatial analysis over basin 8, but didn't you do it over all basins?*

➢ Yes, the analysis was done for every grid cell but particularly in drainage basin 8, we found differences in the spatial patterns between $fv^A$ and $fv^M$. We will delete this sentence.

*L736: Suggest to combine the sentence starting with "The large" with the one before.*

> ➢ As we delete the previous sentence, we will rephrase: "The large uncertainty in basin 8 is likely due to the presence of megadune fields".

*L746: Do you not rather subtract the modeled firn thickness variations from the altimetric variations? That is what eqn A1 says, but your text says the opposite.*

> ➢ Thank you, yes, it should be: "we simply subtract the modelled firn thickness variations from the altimetric variations".

*L791: Where can the IMAU-FDM data be found?*

> ➢ The IMAU-FDM is on GitHub and Zenodo: https://github.com/brils001/IMAU-FDM or https://zenodo.org/records/5172513. For Antarctica, the data can be also provided by the authors. We will include this information in the revised manuscript.

**All the following comments will be revised as suggested**:

- *L294: "scale it SUCH that"*
- *L481: "HAS stronger"*
- *L482: "IS closer to"*
- *L601: includes*
- *L691: "in THE snow"*
- *L725+731: The sentences "However, one caveat should be noted." are a bit odd and short. Suggest you combine them somehow with the sentences coming after.*
- *L733: resolveS*
- *L734: "evaluated AT grid cell level"*
- *L735: Perhaps underline that the basin 5 and 8 numbers are also calculated at grid cell level as in the previous sentence.*
- *L736: "are due" should perhaps be "are likely due" or some other modifier to weaken the claim.*
- *L745: "TO the original"*

Additional references:

- Hansen, N., Langen, P., Boberg, F., Forsberg, R., Simonsen, S., Thejll, P., Vandecrux, B., and Mottram, R.: Downscaled surface mass balance in Antarctica: impacts of subsurface processes and large-scale atmospheric circulation, The Cryosphere, 2021, 4315–4333, https://doi.org/10.5194/tc-15-4315-2021, 2021.
- Tian, B., Lee, H., Waliser, D. E., Ferraro, R., Kim, J., Case, J., Iguchi, T., Kemp, E., Wu, D., Putman, W., and Wang, W.: Development of a Model Performance Metric and Its Application to Assess Summer Precipitation over the U.S. Great Plains in Downscaled Climate Simulations, J. Hydrometeorol., 18, 2781–2799, https://doi.org/10.1175/JHM-D-17-0045.1, 2017.

---

## Author Response (AR1)

**Author's response**

We would like to thank both reviewers once again for their very detailed and supportive feedback on our manuscript. The comments and suggestions have greatly improved our manuscript and helped us a lot to shorten and clarify it. Below we provide a point-by-point response to each comment, sometimes referring to our responses to the review giving prior to the revision. Reviewers' comments are in italics.

**REVIEW 1**

**1 Major comments**

**1.1** *Overall this paper is very dense/wordy, it reads more like a report than a scientific paper. Some sentences and paragraphs seem to just be added as filling without a clear purpose. I suggest heavily cutting the text, especially in the introduction and data section. See some specific suggestions further down.*
- We have significantly shortened the manuscript, in particular the introduction and the data section. We have reduced the main text length from 741 lines to 581 lines.
- We have slightly modified our methods for ease of presentation and explanation, as previously described in the responses to the reviews. We now use the altimetric variations as input to our regression approach. We have therefore restructured the method sections as follows:
    - 3.1 Basic approach
    - 3.2 Principal component analysis of modelled firn thickness variation
    - 3.3 Regression approach
    - 3.4 Different versions of adjusted firn thickness variations
    - 3.5 Assessment methods
- Further, we have reduced the number of figures or subfigures. We have removed Fig. 2, 5a and 5f, 6, 9b, 10a, 11a-c, 12a-c and 16b.

**1.2** *There are a lot of sentences where you use a full stop rather than a comma.*
- We have rephrased or combined the relevant sentences.

**1.3** *Furthermore, there are a lot of grammar errors, like Line 94/95 "Nevertheless, discrepancies still remain. (See Section 2.3 for further details on comparisons between altimetry and firn models.)" It is not grammatically correct to start a new sentence with a paraphrase. All these things are disrupting the reading flow. This paper could really benefit from a proofread! See some specific suggestions further down.*
- We have removed the parentheses and rephrased the sentences if necessary.

**1.4** *There are a lot of inconsistencies in the figures, especially with the colour bars (spatialplots). It seems random if the colour bars have a min or max arrow extension, or both arrow extensions. These colour bar inconsistencies are seen throughout the paper. Also, in Figure S13 red is negative and blue is positive, just below in Figure S14 purple/blue is negative and brown/red is positive. Please go through them all and double-check if all the colour bars are correct.*
- As mentioned in the response to the review, we have added the following sentence to each figure that has a mixture of no arrow extensions and arrow extensions: "Colour bar arrows indicate that the value range exceeds the limits of the colour scale."
- We have changed the Fig. S13 colour bar accordingly.

**1.5** *Snow and firn are mentioned many times but the terms are not well defined here. The manuscript could benefit from a clear distinction between snow and firn. Further, sometimes it says snow/firn layers in plural*

*and sometimes the layer is singular, and firn pack is also used. It seems like the terms are used interchangeably, please make sure to be consistent, as they have different meanings.*

- We have clarified the terms in the introduction when explaining surface mass balance (SMB): "It refers to processes occurring on the surface of the ice sheet in the snow and firn layer. Snow refers to the seasonal snow cover, i.e. it is less than a year old. Firn refers to multiyear snow and is defined as the transition from snow to glacier ice (van den Broeke, 2008). In the following, we refer to both snow and firn by the term firn layer."

**2 Minor comments**

**2.1** *In section 2.4 it seems like you start to present and discuss results (Fig. 1 and 2), this feels out of place in the method sections.*

- We have removed Section 2.4.

**2.2** *In Section 2.1 you refer to the two altimetry datasets as "TUD altimetry" and "JPL altimetry" in lines 124-126. But in the section, you also call them "TUD" and "JPL", or "TUD product" and "JPL product", or "Schröder et al. (2019a)" and "Nilsson et al. (2022)". When you make a definition in the beginningyou should consistently use that. Likewise in section 2.2, you refer to the two firn models as IMAU firn model and GSFC firn model, within the same section you call them "IMAU", "IMAU model", "GSFC", and "GSFC model". When you make a definition in the beginning you should consistently use that.*

- We no longer use the abbreviations TUD, TUD altimetry, JPL, JPL altimetry, IMAU, IMAU model, GSFC and GSFC model. Instead, we now use only the references of the data sets.

**2.3** *In the caption of Figure 2, you write "Color" which is American English other places you write "metre" which is British English. Please choose one way and be consistent.*

- We have changed it to "colour" as we are using British English.

**3 Specific comments**

*L21: The references " (Horwath et al., 2022; IPCC, 2021)" are here in alphabetical order, the rest of the references are in timely order oldest first.*

- We have changed this to: "Ice-mass loss from Antarctica contributed ~6 % to this rise (Horwath et al., 2022), and is likely to continue (IPCC, 2021)."

*L33: "in the snow and firn layer" This reads like there is no difference between snow and firn*

- Please see our answer in 1.5.

*L53: Suggest change "They are forced" to "They can be forced", because not all RCMs use reanalysis data.*

- We have combined this sentence with the previous one as follows: "When the main goal of RCMs is to realistically simulate the ice sheet weather, as is the case here, they are forced by atmospheric reanalysis products and thoroughly evaluated against hundreds of in situ observations of SMB (van Wessem et al., 2018; Agosta et al., 2019)."

*L58: "However, spatial variations in SMB show a poorer agreement. On a basin scale, the largest…" remove the full stop after "agreement" use a comma instead as the following sentence is a continuation.*

- We have greatly shortened this and the following sentences: "However, the spatial patterns of the different SMB estimates differ substantially on a regional and local scale."

*L81: "They" who?*

- We have replaced "They" with "These measurements".

*L87: Here you have snow/firn layers in plural in L33 the layer is singular.*

- According to our definition above (see 1.5), we have changed this to: „(…) while radar signals penetrate into the upper firn layer." Here, and everywhere else we will use the singular."

*L118-119: "For the first time, the entire spatial information present in both the altimetry products and modelling outputs, together with the high (monthly) temporal resolution of gridded altimetry products, is jointly exploited." This is a convoluted sentence, e.g "altimetry products" is mentioned twice, please rewrite.*

- We have rephrased: "For the first time, the full spatial and temporal information present in the altimetry products is exploited together with the modelling results."

*L131-132: Suggest changing the full stop to a comma between "differs. Thus,"*

- We have rephrased and combined both sentences: "While the orbit configurations of the missions entail different limits of coverage close to the poles, all mentioned missions cover at least up to 81.5° S."

*L177-178: In SMB components you are missing refreeze/runoff*

- The IMAU firn model is not forced with all SMB components, but with the fluxes and boundary conditions at the top of the firn layer. Refreezing is a process happening in the firn itself, so this is calculated within the IMAU-FDM model, same as runoff.
- We have revised the description of the GSFC firn model und removed "runoff" as a forcing parameter (L181 of the submitted version). The MERRA-2 land ice runoff flux is provided at monthly resolution (Medley et al., 2022). Medley et al. (2022) use their own degree-day model to generate meltwater fluxes for the input to the firn model. Thus, runoff is also calculated in the model, provided as an output and compared with the MERRA-2 runoff.

*L179: "dynamically downscaled with RACMO2.3p2" does that mean that you have run the RACMO model?*

- As this is not the case, we have rephrased: "The firn model from Veldhuijsen et al. (2023) is forced with three-hourly fields of surface temperature, 10m wind speed and SMB components (snowfall, rainfall, sublimation, snowdrift erosion, snowmelt) from RACMO2.3p2 (van Wessem et al., 2018). RACMO2.3p2 uses a spatial resolution of 27km x 27km and is forced by the ERA5 atmospheric reanalysis data (Hersbach et al., 2020)."

*L262: Replace full stop after regression with comma*

- We have rephrased these sentences: "The stochastic model of our regression in Eq. 1 prescribes a different weighting of observations from two time periods. As results from the older altimetry missions generally have a higher noise level (Schröder et al., 2019a; Nilsson et al., 2022), $hv^A$ after 2003 are weighted by 1, while $hv^A$ before 2003 are given a different (usually lower) weight, which is defined, individually for every grid point, by the ratio of the noise variance of hv195 A before and after 2003."

*L306: Suggest remove " ('R squared')"*

- Instead of removing "R squared", we have removed "the coefficient of determination" and we now refer to $R^2$ as R squared throughout.

*L328: Shouldn't you also refer to fvE1?*

- We have added: "Note, that we do not introduce $fv^{E1}$ as this would correspond to $fv^M$."

*L330: "These alternative variations are called scaled firn thickness variations. We refer to them by fvE2", replace full stop with a comma.*

- We have rephrased: "We refer to the results as scaled firn thickness variations, $fv^{E2}$."

*L332: "These alternative variations are called modified adjusted firn thickness variations. We refer to them by fvE3", replace full stop with a comma.*
- We have rephrased: "We refer to the result as modified adjusted firn thickness variations, $fv^{E3}$."

*L345-355: Please add references for the Kolmogorov-Smirnov test*
- We have added Massey (1951), Miller (1956) and Marsaglia et al. (2003).

*L595: "also high" please put a number on that.*
- We have rephrased this sentence: "In case of basin 8, models and altimetry disagree (Fig. S17e–h), as well as the different versions of $fv^M$ (Fig. S16d) and the different versions of $fv^A$ (Fig. S15i–l). The latter is discussed in Sect. 5.4." We have chosen this qualitative way of describing the disagreement as the absolute numbers may not be essential. This style fits with the description of basin 4 at the beginning of the paragraph.

*L610: What is meant by " measurement noise"?*
- We have rephrased: "Noise in the altimetry measurements might explain (…)"

*L731: Changes the full stop between noted and By to a comma, as it is the same sentence*
- Here and in L726, we have deleted "one caveat should be noted" and we have combined "However" with the following sentence.

**4 Figures**
*Fig3 caption: you write ". Drainage basins of the EAIS and WAIS used in this study (thick black lines) following Rignot et al. (2011a, b)." but then in L298-299 you talk about regions and multiple Rignot basins forming one basin. Please clarify when you talk about Rignot basins, "your" basins or other regions*
- In Fig. 3, we have clarified: "Drainage basins of the EAIS and WAIS used in this study (thick black lines), slightly modified from the definition of Rignot et al. (2011a, b)."
- In the following, we have changed "regions" to "basins".

*Fig3: I suggest that you put all the areas/regions which you mention in the text*
- We have included the region names in Fig. 3.

*Fig10: The left colour bar has only max extend while the right has both min and max extend. Also are there even Coefficients of determination for the regression that are negative, like the min extend suggests?*
- Please see our previous response. Further, Sect. 4.2.1 illustrates the derivation of adjusted firn thickness variations for a selected grid point. There, we have rephrased L408-409: "Because of the different weighting of $hv^A$ before and after 2003 (Sect. 3.3), $R^2_A$ can indeed be negative and distinguishing the two periods is reasonable."

*Fig11: Shouldn't there be a min extension on the colourbar?*
- In the text (L486-487 of the submitted version) we have rephrased and explained: "The employed software to estimate κ (Bos et al., 2012) has -3 as its minimum output value."

*Fig A1: Shouldn't there be a min extension on the colour bar?*
- Please see also our previous response. As it was just the labelling in the wrong place, we have corrected that.

**The following comments refer to sentences or parts of the manuscript that have been deleted and are no longer included in the revised version. Therefore, these comments are no longer addressed point by point. Explanations and replies to specific comments and questions have been provided in the response to the review.**

- *L44: "statistically significant trends in the Antarctic (surface) mass balance" this statement needs a reference.*
- *L52-53: "Earth system models have recently caught up in this regard (Lenaerts et al., 2019)". Is this sentences relevant? ESMs are not mentioned anymore.*
- *L54: "data from 1979 onwards" I guess you refer to ERA-interim put we have ERA5 now starting in 1950*
- *L87: "If elevation changes due to changing ice flow can either be neglected or subtracted…" Please argue where or when they can be neglected.*
- *L94: "Nevertheless, discrepancies still remain. (See Section 2.3 for further details on comparisons between altimetry and firn models.)" Incorrect grammar, suggest rewriting to: Nevertheless, discrepancies between altimetry and firn models still remain and are discussed.*
- *L98-99: "The reason likely lies in errors in the involved altimetry and modelling results" Is that not always the case? either the errors are from the altimetry or the model.*
- *L99: "Therefore" what?*
- *L99: What is meant by a "Steady-state" density model?*
- *L140: Here you write firn pack and firn layer as earlier.*
- *L198: Suggest to delete this line, it sounds like a rapport.*
- *L199: Again "TUD, JPL"*
- *L201: Missing reference to ERA-Interim.*
- *L204: Define the sign of rates, is a positive rate making the surface go up?*
- *L206: Remove "section 2.2"*
- *L207:"more positive" does that mean thicker firn or a faster thickness rate?*
- *L208: "less negative" slower thickness rate, or slower thinning rate?*
- *L210: Do you mean the average ice sheet-wide seasonal amplitude in firn thickness? Or are you talking about ice sheet seasonal amplitude?*
- *L211-213: It is unclear what these numbers are, Are they seasonal differences in firn thickness? Are they integrated over the entire ground AIS?*
- *L215: Which basin definition?*
- *L215: Remove " In this section" it sounds like a rapport.*
- *L219: ". (For the JPL altimetry, h A2, and the GSFC firn model, f Mb, similar time series are shown in Fig. S1.)" using the parentheses is incorrect grammar.*
- *L220: "Agreement between hA1 and fMa is generally good on interannual scales. Differences appear in the long-term trends." these are two very short sentences, I suggest removing the full stop.*
- *L225: Here you say the entire period is 1993-2017, does it not start in 1992?*
- *L230: put s on "model" since you are referring to both models.*
- *L230-231: Using the parentheses is incorrect grammar.*
- *L247: ". It is explained in Section 3.1.1" maybe change to ", this is explained in Section 3.1.1"*
- *Fig9 caption: Replace full stop with and "Histograms. Vertical"*

**All the following comments will be revised as suggested.**

- *L57-63: This could be written much shorter, it is not a review in Mottram et al 2021.*
- *L64-70: Again this could be written much shorter, it is not a review in Verjans et al 2021. I do not think all the numbers are necessary*

- *L71-79: In my opinion, this paragraph could be removed, as gravimetric mass balance is not the topic of this paper.*
- *L80: Remove "By contrast"*
- *L83: Change "utilise(d)" to utilise*
- *L84: Change "(ICESat-2)" to (ICESat), as your time period ends in 2017 and ICESat-2 was first launched in 2018.*
- *L86: "laser signals are reflected at or near the ice sheet surface". I think you mean "laser signals are reflected at or near the surface".*
- *L102: I suggest merging sections 1.2 and 1.3 and removing the text about gravimetry in Sec. 1.3*
- *L104: Remove "(Section 1.2)"*
- *L111: I suggest also merging sections 1.2, and 1,4 and then writing the "purpose part" in the last paragraph in Sec. 1.2.*
- *L113: Remove "(Section 1.1)"*
- *L162-163: Why mention the Shepherd et al 2019 data when you are not using it? I suggest removing these lines.*
- *L172-173: you write "In accordance with the altimetry data, we involve firn thickness changes from the grounded AIS excluding the Antarctic Peninsula and the period May 1992 to December 2017" This sentence says that you exclude the AP and the period 1992-2017, it should likely say that you use the period 1992-2017*
- *L182: "firn layer" singular*
- *L192: " Medley et al. (2022a) built a new model" should it not say parameterisation instead of model?*
- *L284-286: "Comprehensive and general references…." this feels like it is a bit misplaced perhaps to put in the references after "respectively"*
- *L305: Suggest to merge section 3.1.1 and 3.1.2 as sec. 3.1.2 is very short.*
- *L311: Label eq 5a er 6 instead. It is a bit strange to have Eq 5 followed by eq 5a*
- *L360-361: ". (For example, power-law with κ = −1 and κ = −2 represents flicker and random walk noise, respectively.)" using the parentheses is incorrect grammar*
- *L365-366: ". (Note that the residuals may additionally contain signals related to variations in ice flow dynamics or subglacial hydrology.)" using the parentheses is incorrect grammar*
- *L398: "standard deviation" => std, the abbreviation is already introduced.*
- *L402:"(In the case of data gaps in the altimetry time series, this equality holds approximately.)."*
- *L409: Remove "(Section 3.1)"*
- *L431: "(The rms of all versions of fvA and fvM is illustrated in Fig. S17a–d and Fig. S18a, b, respectively.)"*
- *L441:"(Corresponding rms maps of differences are displayed in Fig. S17–S19)."*
- *L448: Section 4.2.4, you already have a section with the same title, this is confusing.*
- *L456-457: "(Fig S20 and Fig. S21 further shows maps of the residuals rms and of R2 for different versions of regression and both time periods. Table S1 lists basin averages of R2 for the period before 2003.)*
- *L472: "(Basin-mean time series of all regression results and versions are presented in Fig. S22–S24.)"*
- *L478-479: "(For the larger subset of selected grid points, Fig. S25 and S26 display the psd of the regression results from A1a and A2a, respectively.)"*
- *L485-486: "(HECTOR only yields numerical stable results for −3.)"*
- *L495: "(For all versions and both PCA, the original and rescaled patterns are illustrated in Fig. S27–S29)."*
- *L507: "power spectral density (psd)" => psd, the abbreviation is already introduced.*
- *L510-511: "(Unlike our analysis, they did not co-estimate a quadratic or seasonal term.)"*

- *L548-549:"(The histograms and cumulative histograms for all basins are shown in Fig. S30 and S31, respectively.)*
- *L556-557: "(See also Table 2 for an overview of the various differences in firn thickness variations fv and their description.)"*
- *Fig1: I suggest to refer to Fig3 in the caption for basin locations.*

Additional references:

Marsaglia, G., Tsang, W., and Wang, J.: Evaluating Kolmogorov's Distribution, Journal of Statistical Software, 8, 1–4, https://doi.org/10.18637/jss.v008.i18, 2003.

Massey, F.: The Kolmogorov-Smirnov Test for Goodness of Fit, Journal of the American Statistical Association, 46, 68–78, 1951.

Miller, L.: Table of Percentage Points of Kolmogorov Statistics, Journal of the American Statistical Association, 51, 111–121, 1956.

**REVIEW 2**

**1 Major concerns**

*The manuscript is very long and for a long time, I was unable to see where we were going and how all the notation, techniques, differences etc. were to be used. I realize that the authors want to be systematic by introducing all the methodology in Section 3 before using it in Section 4 and discussing it in Section 5. But this means that, until somewhere in Sections 4 and 5, I had been loaded with a large amount of notation, techniques etc. without really knowing why I needed to know this.*

**1.1** *Try to shorten the manuscript. I know you want to be thorough and systematic, but is there really no way of making it shorter? It is a very long read.*

- We have significantly shortened the manuscript, in particular the introduction and the data section. We have reduced the main text length from 741 lines to 581 lines.
- In addition, we have slightly modified our methods for ease of presentation and explanation, as previously described in the responses to the reviews. We now use the altimetric variations as input to our regression approach. We have therefore restructured the method sections as follows:
  - 3.1 Basic approach
  - 3.2 Principal component analysis of modelled firn thickness variation
  - 3.3 Regression approach
  - 3.4 Different versions of adjusted firn thickness variations
  - 3.5 Assessment methods
- Further, we have reduced the number of figures or subfigures. We have removed Fig. 2, 5a and 5f, 6, 9b, 10a, 11a-c, 12a-c and 16b.

**1.2** *Somewhere in the beginning make an overview of what the problem is, and the pathway to solve it. Try to include a cartoon or flowchart showing the data coming in, all the intermediate products, the residuals, the analyses done on the residuals, and the use you make of these things. Include also the notation in each box of the chart and the relevant section numbers - not just where they are derived in Section 3 (as in Table B1) but also where they are calculated in Section 4 and used in Section 5. That would provide a road map for the reader making the journey through it all easier to navigate in.*

- We have included a flowchart showing the entire workflow together with the results, their notation, the section where the results are first presented and the main methodological steps to derive these results and the sections where the methods are explained.

**2 Minor issues and typos**

*L53: SOME RCMs specialize…*

- We have rephrased: "When the main goal of RCMs is to realistically simulate the ice sheet weather, as is the case here, they are forced by atmospheric reanalysis products and thoroughly evaluated against hundreds of in situ observations of SMB (van Wessem et al., 2018; Agosta et al., 2019)."

*L178: ERA5: What is the resolution of ERA5? 31 km, right? And this is "downscaled" to 27 km, right? Please argue why? Do you have reason to believe the 27 km RACMO data is better than the 31 km ERA5 data as input to the FDM?*

- Please see our response to the review.

*L181: MERRA2 is downscaled from what resolution to 12.5 km? And how?*

- Please see our response to the review. As the downscaling was part of the NASA Downscaling Project (Tian et al., 2017), we have added this reference to the description of the firn model.

*L262: The sentence starting with "Observations" is very difficult to read. Perhaps start with the fact that you find higher noise levels from the older sections and use this to motivate why you introduce a weighting. Also, say which variable ($r^A$?) represents this noise you talk of.*
*L265-269: This is very difficult to understand. Please see if you can rewrite it more clearly.*

- We have rephrased the whole paragraph (L262-268) and have also considered the comment on L321: "The stochastic model of our regression in Eq. 1 prescribes a different weighting of observations from two time periods. As results from the older altimetry missions generally have a higher noise level (Schröder et al., 2019a; Nilsson et al., 2022), $hv^A$ after 2003 are weighted by 1, while $hv^A$ before 2003 are given a different (usually lower) weight, which is defined, individually for every grid point, by the ratio of the noise variance of $hv^A$ before and after 2003. We assess the noise by the high-pass filtered version of $hv^A$ separately for both periods (cf. Groh et al., 2019). The high-pass filtering consists of removing a low-pass filtered version of $hv^A$, where the low-pass filter is a Gaussian filter with a $6\sigma = 12$ months filter width."

*L321: "deterministic": What do you mean by deterministic? And later on (L669+673), you talk of "stochastic". Exactly what is stochastic? I cannot see any noise terms added anywhere in your methodology.*

- Please see our response to the review. The term stochastic model should now be clarified through the rewording of paragraph L262-268 (see above).

*L371-375: Was very difficult to read. I think I understood it when the results were shown later on, but when reading it here I did not get it.*

- We have included concrete matrix sizes for clarification: "The first PCA is applied to four versions of standardised residuals ($r^{A1a}$, $r^{A1b}$, $r^{A2a}$, $r^{A2b}$). The second PCA is applied to two versions of standardised residual differences ($r^{A1a}$–$r^{A2a}$ and $r^{A1b}$–$r^{A2b}$). For each PCA, we set up one aggregated 'super data matrix' in which we arrange the time series for each grid cell and each version into a single set of time series. Specifically, our data sets comprises m = 90638 points in space (entire area under investigation) and p = 108 points in time (2003–2017). Thus, for the first and second PCA, the super data matrix has the size of 4m x p and 2m x p, respectively. The PCA is conducted to identify the dominant temporal patterns (PCs), which are shared by all versions, together with their space-dependent and version-dependent spatial patterns (EOFs). Each identified mode thus consists of one joint PC (1 x p) and four, or two, EOFs (4m x 1 or 2m x 1) in the case of the first, or second, PCA, respectively."

*L380: You say that the fv^Ma are standardized prior to PCA, but in Fig 4 you say that the EOFs have units of m. How can the EOFs have units if the input is standardized and thereby non-dimensional?*

- We have removed the unit of meter in Fig. 4.

*Fig 7: Why are d-f not identical (or at least similar) to the EOFs in Fig 4? Are you not projecting the model signal on to the PCs that came out of a PCA on exactly that signal? Should that not be a way of recovering the EOFs, i.e., by projecting the signal onto the PCs? Or does it have to do with standardized vs non-standardized signals?*

- First, we now explain the relation between the EOFs and the scaling factors in a simpler way: "To regain interpretable magnitudes of the EOFs, the EOFs are multiplied by the std of the time series of $fv^M$ for each grid cell, which was previously used for standardisation. After this restoration of the signal amplitudes, we no longer speak of EOFs but of modelled scaling factors, $e^M$."
- Second, we now note in Fig. 7 (referring to the submitted manuscript): "(d–f) is the same as Fig. 4a–c but with restored signal amplitudes for each grid cell."

*L455: What is R_s? I cannot remember having this introduced before.*

- We have corrected this to $R^2$.

*L482: "IS closer to"*

- We have rephrased: "We find a stronger autocorrelation for the time series of $fv^{A1a}$ than for that of $r^{A1a}$, (…)"

*L712: The sentence starting with "We deliberately" does not read well. Particularly the word "deliberately" seems odd. Please try to rephrase the sentence.*

- We have rephrased: "We developed a new approach that combines satellite altimetry and firn modelling results to resolve Antarctic firn thickness variations at a high temporal and spatial resolution, namely by monthly 10 km grids."

*L723: outperforms*

- We have rephrased: "The adjusted firn thickness variations, $fv^A$, outperform …"

*L725+731: The sentences "However, one caveat should be noted." are a bit odd and short. Suggest you combine them somehow with the sentences coming after.*

- We have deleted "one caveat should be noted" and we have combined "However" with the following sentence.

*L734: "evaluated AT grid cell level"*

- We have rephrased it as: "Over all grid cells of Antarctica, median absolute and relative (…)"

*L735: Perhaps underline that the basin 5 and 8 numbers are also calculated at grid cell level as in the previous sentence.*

- We have rephrased it as: "Over all grid cells of individual basins, the median relative uncertainties (…)"

*L746: Do you not rather subtract the modeled firn thickness variations from the altimetric variations? That is what eqn A1 says, but your text says the opposite.*

- We have corrected that to: "(…), we simply subtract the modelled firn thickness variations, $fv^M$, from the altimetric variations, $hv^A$, (…)".

*L791: Where can the IMAU-FDM data be found?*

- We have added the following information to the data availability statement: "The code of the firn model from Veldhuijsen et al. (2023) is available at https://github.com/brils001/IMAU-FDM and https://zenodo.org/records/5172513 (Brils et al., 2021). The firn model data from Veldhuijsen et al. (2023) and the results of this study can be obtained from the authors without conditions."

**The following comments refer to sentences or parts of the manuscript that have been deleted and are no longer included in the revised version. Therefore, these comments are no longer addressed point by point. Explanations and replies to specific comments and questions have been provided in the response to the review.**

- *L204: The use of parentheses and the two short sentences "Rates…" and "The three…" is quite clumsy here. Please rephrase.*
- *L206: "evaluated by comparing to"*
- *L214: Try to think of a better title for this sub-section*
- *Eqn 1, 2, and 4: These equations all include a, b, c, d1, …, but they are different (and subject to different regressions) in the three equations, right? Either change the notation or write this out very clearly.*
- *L478: "underlying time series IS displayed"*
- *L635: The sentence starting with "Thus, the …" is difficult to understand.*
- *L735-736: The sentence starting with "Across basin 8" makes it sound as if you only did this spatial analysis over basin 8, but didn't you do it over all basins?*
- *L736: Suggest to combine the sentence starting with "The large" with the one before.*

**All the following comments will be revised as suggested.**

- *L219 (and many other places, e.g., 401): The use of parenthesis after a full stop is not the usual way of doing it. Either a parenthesis refers to and is part of the sentence that is full-stopped, or maybe it should not be a parenthesis at all.*
- *L294: "scale it SUCH that"*
- *L481: "HAS stronger"*
- *L601: includes*
- *L691: "in THE snow"*
- *L733: resolveS*
- *L736: "are due" should perhaps be "are likely due" or some other modifier to weaken the claim.*
- *L745: "TO the original"*

Additional references:

Tian, B., Lee, H., Waliser, D. E., Ferraro, R., Kim, J., Case, J., Iguchi, T., Kemp, E., Wu, D., Putman, W., and Wang, W.: Development of a Model Performance Metric and Its Application to Assess Summer Precipitation over the U.S. Great Plains in Downscaled Climate Simulations, J. Hydrometeorol., 18, 2781–2799, https://doi.org/10.1175/JHM-D-17-0045.1, 2017.

---

## Referee Report (RR1)

**Review on:  How well can satellite altimetry and firn models resolve Antarctic firn thickness variations?**

by Kappelsberger et al 2023

Thanks for implementing the comments, I think the manuscript reads much better now. I would like to see the following done/answered before publication.

**Minor comments:**

I believe that the abstract would be more robust if you could quantify some of your main findings.

The firn model from Veldhuijsen uses RACMO/ERA5 and the firn model from Medley uses MERRA-2, why different firn models and different reanalysis datasets? does that introduce extra uncertainty?

Consider merging sect. 3.5.3 with another section, as it is very short to stand alone

Consider merging sect. 4.4 with another section, as it is very short to stand alone

R-squard vs $R^2$ you use both in the manuscript like in sect 4.2.4 lines 308 and 319. Also you use R-squard in the caption of Fig 9. and  $R^2$ in the caption of Table 3. Choose one for consistency

You write rmsm std, and psd with small letters but RCM, EOF, PCA, and SMB in capital letters, is there any reason for the difference?

Please go through all the figures again and make sure that they have units, if they are unitless that should be indicated, I have given some examples below.

**Specific comments:**

L269. Replace Fig 5 with Figure 5, as it is the beginning of the sentence. As per the submission guidelines.

L 323. Same as above

L 395. Same as above

L527, Is RACMO2.4 not a polar version? referred to as RACMO2.4p1

L540-545. Might be useful with a reference here, maybe this one, https://www.nature.com/articles/s43017-023-00507-9

**Figures:**
Figure 4 has no units on the y-axis nor does the colorbar

Figure 7d. what is the unit, if unitless it should be stated or indicated with (-)

Figure 10b. what is the unit, if unitless it should be stated or indicated with (-)

Figure 11d. what is the unit, if unitless it should be stated or indicated with (-)

Figure 12d. what is the unit, if unitless it should be stated or indicated with (-)

Figure 14. what is the unit, if unitless it should be stated or indicated with (-)

Figures S6 to S9 have no units on the y-axis

Figure S24. what is the unit, if unitless it should be stated or indicated with (-)

Figure S26. what is the unit, if unitless it should be stated or indicated with (-)

Figure S30. what is the unit, if unitless it should be stated or indicated with (-), also can the scaling factor be less than zero, since the colourbar has a minimum extension?

**References:**

The Firn Symposium team. Firn on ice sheets. *Nat Rev Earth Environ* 5, 79–99 (2024). https://doi.org/10.1038/s43017-023-00507-9

---

## Author Response (AR2)

Dear Lousie, dear reviewers,

Thank you very much for reviewing our manuscript again and for your detailed and constructive feedback. We are very pleased that we were able to improve the readability of the manuscript in particular.

Please find below our responses to your comments and our explanations of further changes to the manuscript.

Best regards,
The authors

**Response to Report #1**

Reviewer comments are marked in italics.

**Minor comments:**
- *I believe that the abstract would be more robust if you could quantify some of your main findings.*

We fully agree and have revised the main findings of the abstract.

- *The firn model from Veldhuijsen uses RACMO/ERA5 and the firn model from Medley uses MERRA-2, why different firn models and different reanalysis datasets? does that introduce extra uncertainty?*

We now clarify in the data section: "We include different firn modelling data and altimetry products to test the sensitivity of our results to the choice of data sets and to assess uncertainties."

- *Consider merging sect. 3.5.3 with another section, as it is very short to stand alone*

We have considered this, but the content of Sect. 3.5.3 does not fit with any of the other sections. We therefore prefer it to stand on its own, in line with its own separate topic.

- *Consider merging sect. 4.4 with another section, as it is very short to stand alone*

Same as above.

- *R-squared vs R 2 you use both in the manuscript like in sect 4.2.4 lines 308 and 319. Also you use R-squared in the caption of Fig 9. and R 2 in the caption of Table 3. Choose one for consistency*

It is true that sometimes we write "R-squared" and sometimes we also use "$R^2$". We would prefer to continue using both, since "R-squared" is the name, while "$R^2$" is the corresponding mathematical symbol. This usage is also consistent with the way we treat other variables, such as the "altimetric residuals" (name) with "$r^A$" (mathematical symbol).

- *You write rmsm std, and psd with small letters but RCM, EOF, PCA, and SMB in capital letters, is there any reason for the difference?*

This is indeed inconsistent. We have changed the lower case of the abbreviations to upper case.

- *Please go through all the figures again and make sure that they have units, if they are unitless that should be indicated, I have given some examples below.*

All unitless variables presented in the figures are now indicated by (-), as proposed.

**All specific comments have been revised as suggested:**
- *L269. Replace Fig 5 with Figure 5, as it is the beginning of the sentence. As per the submission guidelines.*
- *L 323. Same as above*
- *L 395. Same as above*
- *L540-545. Might be useful with a reference here, maybe this one, https://www.nature.com/articles/s43017-023-00507-9*

- *L527, Is RACMO2.4 not a polar version? referred to as RACMO2.4p1*

In the meantime, a paper and a dataset of RACMO2.4p1 have also been submitted. Therefore we have changed the sentence in L527 and the reference in L529:
„An update of RACMO2.3p2 to RACMO2.4p1 with enhanced physics is now available for 2006 to 2015. This includes (…) (van Dalum et al., 2024).“

Previous reference:
van Dalum, C. and van de Berg, W.: First results of RACMO2.4: A new model version with updated surface and atmospheric processes, in: EGU General Assembly 2023, Vienna, Austria, 24–28 Apr 2023, EGU23-13907, https://doi.org/10.5194/egusphere-egu23-13907, 2023.

New reference (preprint):
van Dalum, C., van de Berg, W., Gadde, S., van Tiggelen, M., van der Drift, T., van Meijgaard, E., van Ulft, L., and van den Broeke, M. R.: First results of the polar regional climate model RACMO2.4, EGUsphere, pp. 1–36, https://doi.org/10.5194/egusphere-2024-895, 2024.

**All comments on the figures have been revised as suggested:**
- Figure 4 has no units on the y-axis nor does the colorbar

It is now indicated by (-).

- Figure 7d. what is the unit, if unitless it should be stated or indicated with (-)
- Figure 10b. what is the unit, if unitless it should be stated or indicated with (-)
- Figure 11d. what is the unit, if unitless it should be stated or indicated with (-)
- Figure 12d. what is the unit, if unitless it should be stated or indicated with (-)
- Figure 14. what is the unit, if unitless it should be stated or indicated with (-)
- Figures S6 to S9 have no units on the y-axis

It is now indicated by (-).

- Figure S24. what is the unit, if unitless it should be stated or indicated with (-)
- Figure S26. what is the unit, if unitless it should be stated or indicated with (-)

- Figure S30. what is the unit, if unitless it should be stated or indicated with (-), also can the scaling factor be less than zero, since the colourbar has a minimum extension?

The scaling factor *e* is estimated by the least squares method and it can be negative. The corresponding regression approach is presented in Eq. A2.
The interesting question might be: Does this scaling make sense for e < 0 or what does it mean? For the grid cells where the time series from the firn model must be negatively scaled to match the altimetry time series as best as possible, it can be concluded that altimetry and the firn model cannot be meaningfully combined at the grid cell level because their errors are too large. This is somewhat remedied by the approach finally adopted, which does not scale the firn model time series per grid cell, but rather its dominant PCs.

**Response to Report #2**

Reviewer comments are marked in italics.

**Minor issues:**
- *L192: "The stochastic model". I asked about the deterministic/stochastic issues last time and I did not understand your reply. You point me to read this part of the text, but I still cannot see that you describe what is stochastic in Eqn (1). You have a regression onto the PCs and then there is a residual. Where is the stochastic part in that?*

Yes, there is no stochastic part in Eq. 1. Eq. 1 is the deterministic or also called functional model of the least-squares adjustment. The unknown parameters in Eq. 1 are solved by the least-squares method taking into account the variance-covariance information of the altimetric variations. From L192 onwards, the stochastic model or variance-covariance matrix is described verbally rather than by equations. The different weighting for different time periods means that our stochastic model is only a diagonal matrix, so only variances and no covariances are considered.

We agree that the term "stochastic model" may be unnecessary and confusing at this stage, so we have changed the sentence in L192 to read:
"We use different weights for the observations from different time periods."

- *L247-254: This paragraph is still hard to follow. I know you have already worked with it but give it another round, please.*

We have revised the paragraph again and explain why the individual data sets/matrices are merged into a single data set/matrix for PCA. We hope that the reasoning and procedure is now a little clearer.

- *L262: "longer wavelength". Would it not be better with "lower frequency" to distinguish spatial and temporal characteristics?*

We have changed this.

- *278+279: In 278 you have 0.831 and in 279 you have 82%. Should this not be the same number?*

Indeed, this is a typo. We changed it to 83%.

- *Table 2: The line "A1a-A2a", shouldn't this be "different ... based on Ma" and "Mb" for "A1b-A2b"?*

This is correct. In the revision, the names and numbers were adjusted and rearranged, but we forgot to rearrange the corresponding descriptions. We have corrected this. Thanks a lot for pointing this out.

- *L418: "This lessens the actual firn compaction variability" This makes it sound like the real world variability is lessened, but that is not what you mean, right? Perhaps something like "This lessens the modelled firn compaction variability compared to the actual variability..."?*

We agree that the wording is misleading. We have reworded the sentence as suggested.

- *L575-577: Yes, but also to (correctly) modeled variability which just is not covered by the included PCs, right?*

That is correct. Although it is a minor part, it should be mentioned. We have added the following sentence: "A small part of these residual signals may be due to the limitation of our regression which neglects up to 10 % of the potentially correctly modelled variations. However, we attribute the larger part to (...)".